# Curated Synthetic Data Doesn't Have to Collapse: A Theoretical Study of Generative Retraining with Pluralistic Preferences

**Ali Falahati** [1]  **Mohammad Mohammadi Amiri** [2]  **Kate Larson** [1]  **Lukasz Golab** [1]

## Abstract

Recursive retraining of generative models poses a critical representation challenge: when synthetic outputs are curated based on a fixed reward signal, the model tends to collapse onto a narrow set of outputs that over-optimize this objective. Prior work suggests that such collapse is unavoidable without adding real data into the mix. We revisit this conclusion from an alignment perspective and show that collapse can be mitigated through curation based on multiple reward functions. We formalize the dynamics of recursive training under heterogeneous preferences and prove that, under certain conditions, the model converges to a stable distribution that allocates probability mass across competing high-reward regions. The limiting distribution preserves diversity and provably satisfies a weighted Nash bargaining solution, offering a formal interpretation of value aggregation in synthetic retraining loops.

## 1. Introduction

As generative models grow in scale (Bommasani, 2021; Kaplan et al., 2020), reliance on human-generated data has become a major bottleneck (Villalobos et al., 2022). To keep up with demand, model developers increasingly turn to synthetic data, generated by previous model iterations, as a substitute (Ferbach et al., 2024; Shumailov et al., 2023). However, when synthetic data are selectively retained across model iterations, favoring samples with certain desirable traits, the model may gradually lose the ability to represent anything outside this narrow preference. This phenomenon, known as mode collapse, was formalized by Ferbach et al. (2024), who modeled retraining as a recursive process guided by a single reward function. They showed

that when training data are repeatedly selected to maximize one criterion (e.g., fluency or realism), the model distribution eventually collapses onto a small subset of outputs that optimize this reward. In practice, however, content creators, platform users, and other stakeholders often have conflicting preferences (Falahati et al., 2025; Park et al., 2024; Wu et al., 2023). Some may value factual accuracy, while others prioritize creativity or emotional resonance. The growing use of outsourced or geographically concentrated labeling workforces (Miceli et al., 2020; Gray & Suri, 2019) further complicates the picture, introducing cultural biases (Denton et al., 2021; Sambasivan et al., 2021). These real-world complexities prompt a natural question: Does recursive retraining always lead to collapse?

We study what happens when synthetic data are curated based on multiple preferences, and find that the resulting distribution does not collapse. Instead, it converges to a stable mixture, maintaining support over all high-reward regions. Diversity is preserved, reward variance remains positive, and the allocation of probability mass reflects the balance between the preferences. Moreover, we show that this limiting distribution can be understood as a fair compromise between the competing objectives. The final outcome satisfies a weighted version of the Nash bargaining solution, with the proportions of each preference guiding the trade-off. These results suggest that the outcome of synthetic retraining is not solely determined by the use of artificial data, but by how these data are curated. A full discussion of related work is provided in the Appendix A.

**Relevance to AI Alignment.** When artificial intelligence (AI) systems are trained or evaluated on a single metric, they often exploit this metric in unintended ways, exemplifying Goodhart's law: "When a measure becomes a target, it ceases to be a good measure" (Strathern, 1997). Our approach of using multiple reward functions reflects the value pluralism problem in alignment, where different stakeholders may have legitimate but conflicting preferences that must be balanced rather than collapsed into a single objective (Gabriel, 2020). Furthermore, as AI systems increasingly generate training data for future systems, understanding how to maintain diversity and prevent degenerative feedback loops becomes crucial for long-term alignment and

---

[1]University of Waterloo [2]Rensselaer Polytechnic Institute. Correspondence to: Ali Falahati <afalahat@uwaterloo.ca>.

*Proceedings of the 43rd International Conference on Machine Learning*, Seoul, South Korea. PMLR 306, 2026. Copyright 2026 by the author(s).

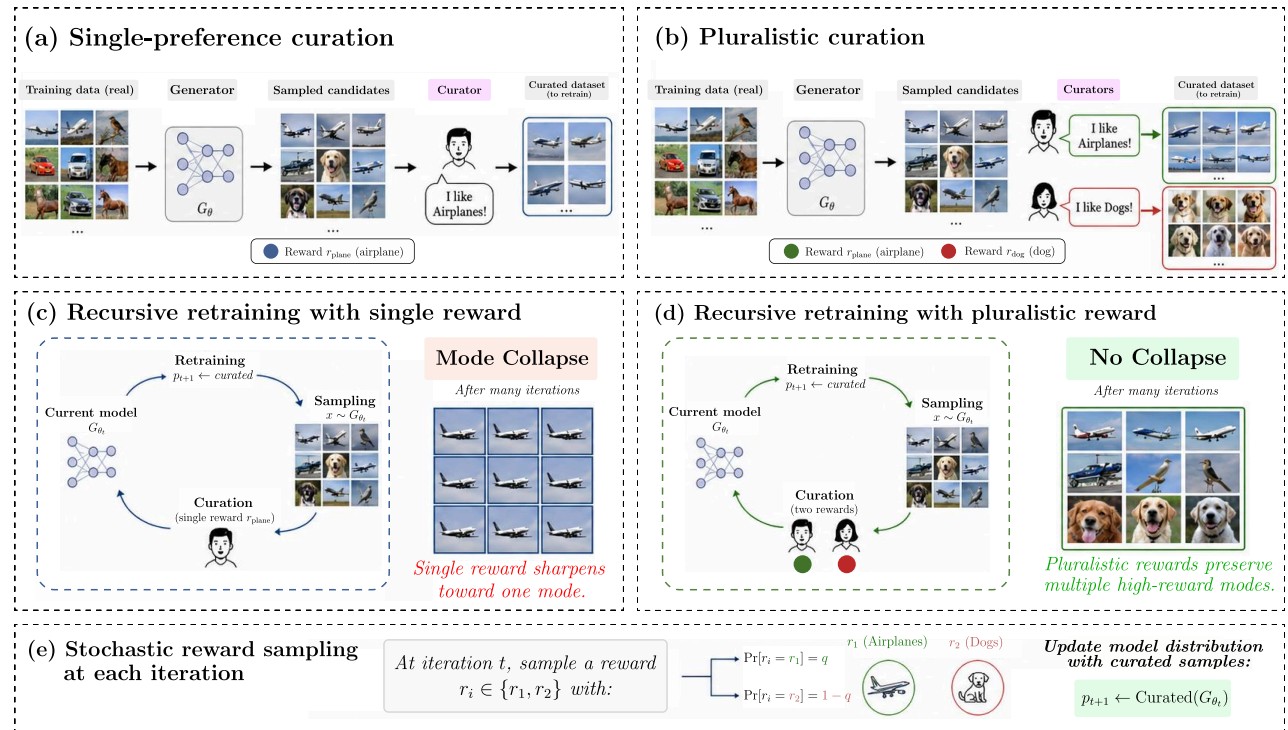

*Figure 1.* Overview of pluralistic curation. At each retraining step, the generative model samples candidates from its current distribution. A reward function is chosen at random $r_1$ with probability $q$, or $r_2$ with probability $1-q$ and one candidate is selected. The model is then retrained to maximize the likelihood of this curated sample. Over time, this alternating selection drives the model toward a stable mixture over the distinct high-reward regions associated with $r_1$ and $r_2$, preventing mode collapse and preserving diversity.

safety (Christiano et al., 2018).

This work contributes to the AI alignment literature by exploring how pluralistic preferences can prevent degenerative behaviors in self-improving generative systems. In contexts such as reinforcement learning from human feedback (RLHF), single-objective optimization often leads to reward hacking or mode collapse, where models exploit narrow interpretations of preferences at the expense of robustness and inclusivity (Casper et al., 2023). By modeling curation as a bargaining process among rewards, our framework promotes equitable aggregation of values, akin to approaches in democratic fine-tuning (Sorensen et al., 2024) and constitutional AI (Bai et al., 2022), where diverse stakeholder inputs are synthesized to achieve more holistic alignment. Our weighted Nash bargaining interpretation offers a principled framework for aggregating multiple human preferences in a fair and stable manner, addressing concerns about whose values AI systems should optimize for and how to avoid marginalizing minority preferences during alignment.

In summary, we make the following contributions:

**(1) Pluralistic retraining dynamics and basin concentration.** We formalize recursive retraining under heterogeneous preferences, where each step curates samples via a Bradley-Terry (BT) choice model using a reward drawn from a fixed mixture. In the large-candidate regime, the induced update is an exponential tilt of the current model. Under an outside-domination condition, the dynamics drive mass into the union of $\varepsilon$-optimal basins for the competing rewards.

**(2) Pluralistic limit structure and leakage-controlled mixture weights.** We show that once outside mass vanishes, every subsequent limit is supported on the near-optimal basin union and decomposes as a two-component mixture. The limiting basin weight is constrained by explicit leakage factors that quantify cross-reward reinforcement, and it approaches the reward sampling weight as separation increases and leakage vanishes.

**(3) Non-collapse and value-aggregation interpretation.** We prove diversity preservation in the sense that any non-trivial two-basin limit mixture yields strictly positive reward variance, with quantitative lower bounds under basin separation. In a plateau and hard-separation regime, the limiting mixture becomes explicit and coincides with the weighted Nash bargaining solution, clarifying how long-run behavior reflects relative preference influence.

## 2. Retraining with Pluralistic Preferences

In recursive retraining pipelines, synthetic samples are selected based on a reward signal that encodes desirable traits

such as realism or fluency. Ferbach et al. (2024) modeled this data selection as a discrete choice process based on a single reward function $r(x)$, with sample selection governed by the BT model (Bradley & Terry, 1952). This mirrors practical setups such as ranking large language model (LLM) outputs, where annotators select the most preferred sample from a set of candidates. The BT model assigns selection probabilities based on reward values (Equation 2), favoring higher-scoring samples. Over repeated retraining steps, this process has been shown to concentrate probability mass on high-reward regions, eventually collapsing the model distribution onto the set of maximal outputs. In the single-reward case, repeated BT selection induces an exponential reweighting toward maximizers, which can drive mode collapse in the infinite-capacity retraining limit (Ferbach et al., 2024).

However, many real-world applications involve competing preferences. For instance, different users may value factuality, creativity, or novelty to varying degrees. To study such scenarios, we consider a variant of the retraining process in which the selection of synthetic samples alternates probabilistically between two distinct reward functions: $r_1(x)$ and $r_2(x)$. Each reward captures a different notion of value, such as realism versus artistic expression, and the choice of which reward to apply is randomized in each selection step. The pipeline is shown in Figure 1. Our goal is to characterize the long-run effect of this mixed-preference loop: does alternating rewards still drive the model toward a single extreme, or can it stabilize on a pluralistic distribution that retains mass on multiple high-reward regions in a way that reflects the mixture weight $q$?

## 2.1. Retraining Dynamics

**Setup.** Let $\mathcal{X} \subseteq \mathbb{R}^d$ be the data domain, and let $p_{\text{data}}$ denote the (unknown) data-generating distribution on $\mathcal{X}$. Let $p_t$ denote the model distribution at retraining iteration $t$ (i.e., the distribution induced by the generator after $t$ rounds). Assume each $p_t$ has a density $p_t(x)$, so for any measurable $A \subseteq \mathcal{X}$, $p_t(A) = \int_A p_t(x)dx$. Initialize $p_0 \approx p_{\text{data}}$. Each retraining iteration has three stages:

**Sampling.** Draw $K \geq 2$ independent and identically distributed (i.i.d.) candidates $\{x_1, \ldots, x_K\} \sim p_t$.

**Multi-preference selection.** Independent of the sampled candidates $\{x_1, \ldots, x_K\}$, choose which reward ($r : \mathcal{X} \to \mathbb{R}$) governs selection: use reward $r_1$ with probability $q \in (0, 1)$ and $r_2$ with probability $1 - q$. Conditional on the chosen reward $r \in \{r_1, r_2\}$ and the candidate set $\{x_1, \ldots, x_K\}$, select one candidate $\hat{x}$ according to the BT

$$\mathbb{P}(\hat{x} = x_k \mid x_{1:K}, r) = \frac{e^{r(x_k)}}{\sum_{j=1}^{K} e^{r(x_j)}}. \quad (1)$$

The parameter $q$ is the reward-sampling weight. It specifies

how often the curation pipeline applies preference $r_1$ rather than $r_2$. Thus, $q$ can represent the relative prevalence of two user groups or a policy choice about how much influence each preference receives. $q = 0$ and $q = 1$ recover single-reward curation, while intermediate values define pluralistic curation.

**Definition 1** (BT weights). *Given $p$, $r$, and $K \geq 2$, define*

$$H_p^{K,r}(x) := \mathbb{E}_{x_2,\ldots,x_K \sim p}\left[ \frac{K\, e^{r(x)}}{e^{r(x)} + \sum_{j=2}^{K} e^{r(x_j)}} \right]. \quad (2)$$

Intuitively, $H_p^{K,r}(x)$ is the expected BT win probability of candidate $x$ when compared against $K - 1$ i.i.d. draws from $p$; values $> 1$ indicate $x$ is favored under $r$ relative to typical samples from $p$ (Ferbach et al., 2024).

**Selected-sample distribution.** Let $\tilde{p}_t^{(r)}$ denote the marginal density of $\hat{x}$ when using reward $r$ at step $t$. Then

$$\tilde{p}_t^{(r)}(x) = p_t(x)\, H_{p_t}^{K,r}(x), \qquad \int_{\mathcal{X}} p_t(x)\, H_{p_t}^{K,r}(x)\, dx = 1. \quad (3)$$

With two rewards, the selected-sample density is the mixture

$$\tilde{p}_t(x) = q\, \tilde{p}_t^{(r_1)}(x) + (1 - q)\, \tilde{p}_t^{(r_2)}(x). \quad (4)$$

**Retraining (idealized MLE update).** We model retraining as exact maximum likelihood over all densities $p$ on $\mathcal{X}$,

$$p_{t+1} \in \arg\max_p\ \mathbb{E}_{\hat{x} \sim \tilde{p}_t}\big[ \log p(\hat{x}) \big].$$

Under this unconstrained objective, the maximizer is $p_{t+1} = \tilde{p}_t$. This corresponds to an idealized update where retraining can represent $\tilde{p}_t$ exactly and optimization is solved exactly. Our goal is to determine whether recursive retraining collapses onto a single preference region or maintains support across multiple ones. We track how much mass $p_t$ assigns to neighborhoods of each reward maximizer by defining $\varepsilon$-optimal basins and studying their mass dynamics.

**Near-optimal basins.** For $i \in \{1, 2\}$ let $r_i^* := \sup_{x \in \mathcal{X}} r_i(x)$. Fix $\varepsilon > 0$ and define the $\varepsilon$-optimal regions

$$\mathcal{S}_{i,\varepsilon} := \{x \in \mathcal{X} : r_i(x) \geq r_i^* - \varepsilon\}, \qquad \mathcal{S}_\varepsilon := \mathcal{S}_{1,\varepsilon} \cup \mathcal{S}_{2,\varepsilon}.$$

We track basin masses $a_t := p_t(\mathcal{S}_{1,\varepsilon})$, $b_t := p_t(\mathcal{S}_{2,\varepsilon})$, and $m_t := p_t(\mathcal{X} \setminus \mathcal{S}_\varepsilon)$, so $a_t + b_t + m_t = 1$.

**Assumption 1** (Two-basin landscape with leakage gaps). *Fix $\varepsilon > 0$ and define $\mathcal{S}_{i,\varepsilon}$ and $\mathcal{S}_\varepsilon$ as above. Assume:*

*(i) **Bounded rewards.** For each $i \in \{1, 2\}$, $r_i : \mathcal{X} \to \mathbb{R}$ is bounded: $r_i(x) \in [L_i, U_i]$ for all $x \in \mathcal{X}$.*

*(ii) **Disjoint near-optimal basins with nontrivial initialization.** $\mathcal{S}_{1,\varepsilon} \cap \mathcal{S}_{2,\varepsilon} = \emptyset$ and $p_0(\mathcal{S}_{1,\varepsilon}) > 0$, $p_0(\mathcal{S}_{2,\varepsilon}) > 0$.*

*(iii) Outside domination and cross-reward leakage. In the large-K dynamics, define*

$$Z_i(t) := \mathbb{E}_{p_t}[e^{r_i(x)}], \quad W_t(x) := q\,\frac{e^{r_1(x)}}{Z_1(t)} + (1-q)\,\frac{e^{r_2(x)}}{Z_2(t)}.$$

*There exists $\rho_\varepsilon \in (0,1)$ such that*

$$\sup_{x \notin \mathcal{S}_\varepsilon} W_t(x) \le \rho_\varepsilon \qquad \text{for all } t \ge 0.$$

*Moreover, there exist $\Delta_1, \Delta_2 > 0$ such that*

$$x \in \mathcal{S}_{1,\varepsilon} \Rightarrow r_2(x) \le r_2^* - \Delta_2 + \varepsilon,$$

$$x \in \mathcal{S}_{2,\varepsilon} \Rightarrow r_1(x) \le r_1^* - \Delta_1 + \varepsilon.$$

These assumptions isolate the two mechanisms used in the analysis. First, outside domination ensures that mass not assigned to the modeled basin union $\mathcal{S}_\varepsilon$ is not reinforced by the pluralistic update. Second, the gaps $\Delta_1$ and $\Delta_2$ control cross-reward leakage between the two basins. When leakage is small, the limiting basin weight approaches the reward-sampling weight $q$; when leakage is non-negligible, our bounds quantify the resulting deviation. We study this transition empirically by varying reward separation, with further discussion in Appendix B.2.

## 3. Theoretical Results

We analyze two rewards, which already capture the essential effect of preference heterogeneity while keeping the analysis tractable. We generalize to $M$ preferences at the end of this section. Our result is that alternating which reward drives curation prevents the model from committing to a single "favorite" region, and instead drives it toward *pluralistic limit points* that keep nontrivial mass on multiple high-reward basins. The logic has four steps. (i) When we sample $K$ candidates and select via BT, we effectively apply a soft "advantage weighting" to $p_t$: higher-reward points are more likely to win. (ii) If points outside the modeled basin union have multiplicative update factor uniformly below one, then probability mass contracts into $\mathcal{S}_\varepsilon$. (iii) Once most mass lies in $\mathcal{S}_\varepsilon$, the remaining question is how it splits between basins; this split is governed by *leakage*, i.e. how much reward $r_1$ can still favor the other basin and vice versa. As leakage shrinks, each reward mostly reinforces its own basin, so the long-run mixture approaches the reward-sampling weight $q$. (iv) In a hard-separation idealization, the limit becomes an explicit $q$-mixture, which makes the diversity and Nash bargaining interpretations transparent. Proofs are deferred to Appendix B.

### 3.1. Curation as Reweighting

Our starting point is to understand the curation step itself. At iteration $t$, we draw $K$ candidates $\{x_1, \ldots, x_K\}$ from the current model $p_t$ and select one using the BT rule. The question is: how does this selection bias depend on the function $r$, and what update does it induce on $p_t$? The first lemma is the distributional update induced by one step of sampling, BT-selection, and retraining. Recall that retraining is modeled as an unconstrained MLE update, so $p_{t+1}$ is exactly the distribution of the selected sample $\hat{x}$. We also write the large-pool limit weight as

$$H_p^{\infty,r}(x) := \frac{e^{r(x)}}{\mathbb{E}_{y \sim p}[e^{r(y)}]}.$$

**Lemma 1** (Pluralistic curation update)**.** *At step $t$, the retraining update satisfies*

$$p_{t+1}(x) = p_t(x)\Big(q\,H_{p_t}^{K,r_1}(x) + (1-q)\,H_{p_t}^{K,r_2}(x)\Big).$$

The expression above is exact but still depends on a $(K-1)$-sample expectation. A key simplification occurs in the large-pool regime: the denominator concentrates, and the curation weight converges to an exponential tilt of the current model.

**Lemma 2** (Large-$K$ concentration of curation weights)**.** *Assume $r$ is bounded and measurable. Then as $K \to \infty$,*

$$H_p^{K,r}(x) \longrightarrow H_p^{\infty,r}(x) = \frac{e^{r(x)}}{\mathbb{E}_{y \sim p}[e^{r(y)}]}.$$

Combining Lemmas 1 and 2, the pluralistic update in the large-$K$ regime becomes

$$p_{t+1}(x) = p_t(x)\Big(q\,\tfrac{e^{r_1(x)}}{Z_1(t)} + (1-q)\,\tfrac{e^{r_2(x)}}{Z_2(t)}\Big), \quad (5)$$

where $Z_i(t) := \mathbb{E}_{p_t}[e^{r_i(x)}]$. Intuitively, each step applies a multiplicative boost to points that score well under the sampled reward, followed by normalization. The $K \to \infty$ limit provides a clean "tilt-and-renormalize" update, but in practice $K$ is finite.

**Remark 1** (Finite-$K$ robustness)**.** Fix $\varepsilon > 0$ and define the restricted deviation

$$\Delta_K(r,p;\varepsilon) := \sup_{x \in \mathcal{S}_\varepsilon} \big|H_p^{K,r}(x) - H_p^{\infty,r}(x)\big|.$$

In the appendix, Lemma B.3 shows $\Delta_K(r,p;\varepsilon) = O(\sqrt{\log(K)/K})$. Consequently, on $\mathcal{S}_\varepsilon$ the finite-$K$ pluralistic update can be written as the $K = \infty$ update plus a controlled perturbation:

$$p_{t+1}(x) = p_t(x)\Big[q\,H_{p_t}^{\infty,r_1}(x) + (1-q)\,H_{p_t}^{\infty,r_2}(x) + \xi_t(x)\Big],$$

$$|\xi_t(x)| \le q\,\Delta_K(r_1,p_t;\varepsilon) + (1-q)\,\Delta_K(r_2,p_t;\varepsilon).$$

Thus the within-basin leakage comparisons are stable under finite $K$, up to an additional $O(\sqrt{\log K/K})$ slack. Extending the outside-mass contraction to finite $K$ requires the analogous outside-domination condition for the finite-$K$ multiplier.

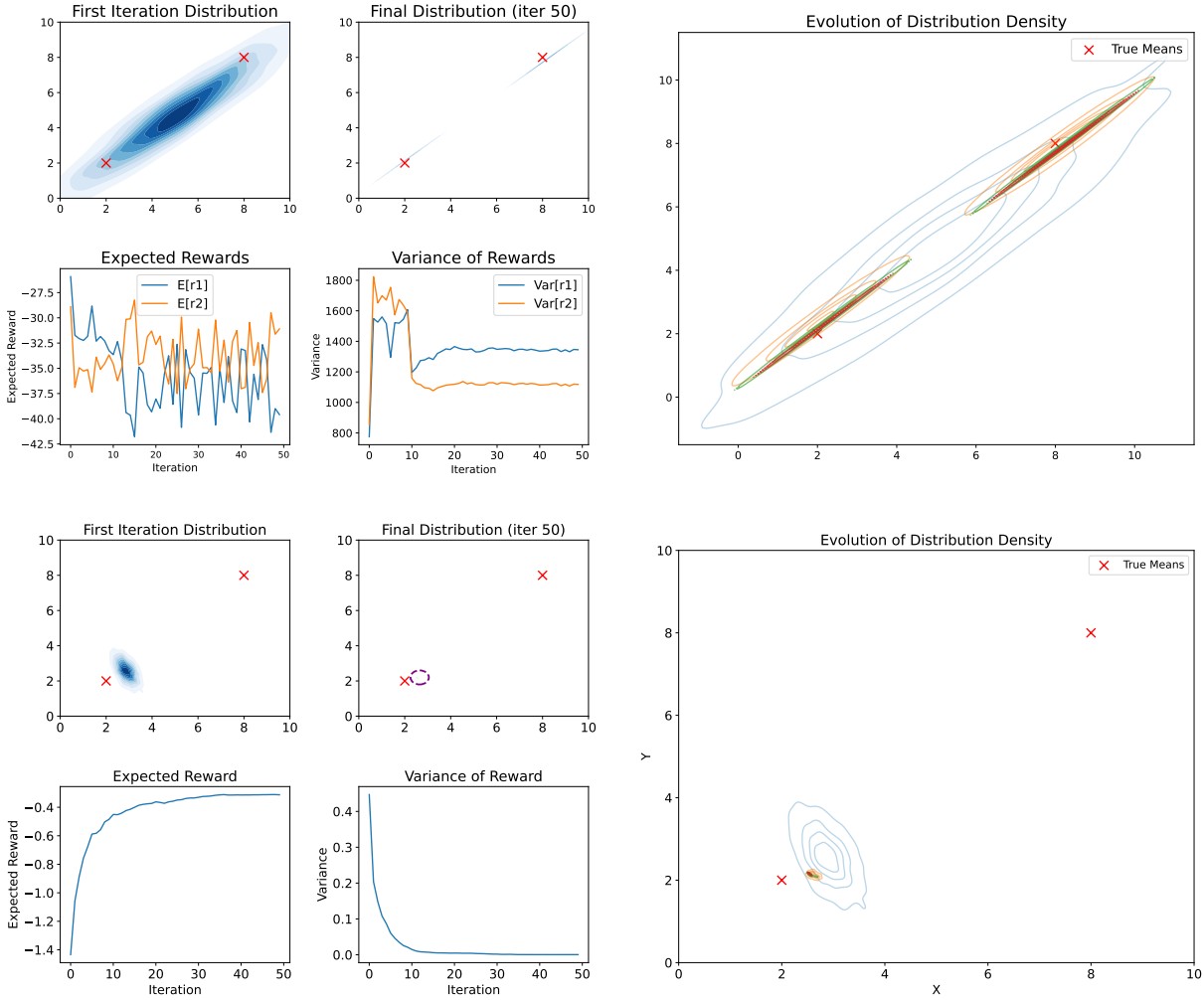

*Figure 2.* **Synthetic Dataset.** Pluralistic curation (top) vs. single-reward retraining (bottom) on a two-mode Gaussian mixture with means at $\mu_1 = (2, 2)$ and $\mu_2 = (8, 8)$. Bottom plots track expected rewards (left) and reward variances (right) across retraining steps.

### 3.2. Mass Leaves Low-Reward Regions

With the large-$K$ update (5), we now ask: does pluralistic retraining keep probability mass outside the modeled basin union $\mathcal{S}_\varepsilon$? The relevant quantity is the update multiplier

$$W_t(x) = q \frac{e^{r_1(x)}}{Z_1(t)} + (1 - q) \frac{e^{r_2(x)}}{Z_2(t)}.$$

Assumption 1(iii) states that this multiplier is uniformly below one on $\mathcal{X} \setminus \mathcal{S}_\varepsilon$. Therefore outside mass contracts geometrically under the update.

**Lemma 3** (Decay outside $\varepsilon$-optimal basins). *Fix $\varepsilon > 0$ and work in the large-$K$ regime (5). Under Assumption 1(i,iii), there exists a constant $\rho_\varepsilon \in (0, 1)$ such that $m_{t+1} \leq \rho_\varepsilon m_t$ for all $t$. Hence $m_t \leq \rho_\varepsilon^t m_0 \to 0$ and therefore $a_t + b_t = 1 - m_t \to 1$, i.e. asymptotically all mass concentrates on $\mathcal{S}_\varepsilon = \mathcal{S}_{1,\varepsilon} \cup \mathcal{S}_{2,\varepsilon}$.*

Thus, asymptotically, the distribution concentrates on the union of the two near-optimal basins. What remains is to understand how the total mass splits between them.

### 3.3. Mass Partitioning With Leakage

Once Lemma 3 drives $m_t \to 0$, the long-horizon behavior is governed by how the remaining mass splits between the two $\varepsilon$-optimal basins, $\mathcal{S}_{1,\varepsilon}$, and $\mathcal{S}_{2,\varepsilon}$. The only obstruction to an "exact" $q$-mixture is *leakage*: points in $\mathcal{S}_{2,\varepsilon}$ may still receive non-negligible reward under $r_1$ (and vice versa), so sampling $r_1$ can inadvertently reinforce the wrong basin. Assumption 1(iii) quantifies this cross-reward reinforcement through separation margins $\Delta_1$ and $\Delta_2$, which we summarize via exponential leakage factors

$$\kappa_1 := \exp\big(-(\Delta_1 - 2\varepsilon)\big), \qquad \kappa_2 := \exp\big(-(\Delta_2 - 2\varepsilon)\big).$$

Heuristically, $\kappa_1$ upper-bounds how much $r_1$ can boost $\mathcal{S}_{2,\varepsilon}$ relative to $\mathcal{S}_{1,\varepsilon}$, and $\kappa_2$ does the symmetric comparison.

**Lemma 4** (Leakage-controlled limiting basin mass). *Fix $\varepsilon > 0$ and assume $\mathcal{S}_{1,\varepsilon} \cap \mathcal{S}_{2,\varepsilon} = \emptyset$ and $m_t \to 0$. Suppose the separation margins satisfy $\Delta_1, \Delta_2 > 2\varepsilon$ (equivalently $\kappa_1, \kappa_2 \in (0,1)$). If $a_t \to a_\infty$, then*

$$\max\left\{0, \frac{q - \kappa_1}{1 - \kappa_1}\right\} \leq a_\infty \leq \min\left\{1, \frac{q}{1 - \kappa_2}\right\}.$$

*Moreover, as $\Delta_1, \Delta_2 \to \infty$ (so $\kappa_1, \kappa_2 \to 0$), this interval collapses to the point $a_\infty = q$.*

Lemma 4 shows that leakage controls how far the limiting mixture can deviate from $q$: outside the vanishing-leakage regime, the weight is constrained to lie near $q$, and becomes exactly $q$ only when cross-basin reinforcement disappears. The conditional nature of Lemma 4 leaves open whether the dynamics could still collapse to a single basin. The following lemma addresses this gap: when $q$ dominates the leakage, the basin mass is eventually bounded away from both 0 and 1.

**Lemma 5** (Uniform non-collapse under bounded leakage). *Fix $\varepsilon > 0$ and work in the large-$K$ regime. Assume the decay conclusion $m_t \to 0$ and suppose $q \in (\kappa_1, 1 - \kappa_2)$. Define*

$$L := \frac{q - \kappa_1}{1 - \kappa_1}, \qquad U := \frac{q}{1 - \kappa_2}.$$

*Then*

$$\liminf_{t \to \infty} a_t \geq L, \qquad \limsup_{t \to \infty} a_t \leq U.$$

*In particular, there exist $\eta > 0$ and $t_0$ such that*

$$\eta \leq a_t \leq 1 - \eta \qquad \text{for all } t \geq t_0.$$

**Corollary 2** (Nondegenerate limit points). *Under the assumptions of Lemma 5, every subsequential limit $p_{t_k} \Rightarrow p_\infty$ satisfies $p_\infty(\mathcal{S}_{1,\varepsilon}) \in [\eta, 1 - \eta]$. Hence, no limit point collapses onto a single basin.*

The next theorem states: once $m_t \to 0$, every subsequential limit is a two-basin mixture.

**Theorem 3** (Two-basin subsequential limit). *Fix $\varepsilon > 0$, assume the decay conclusion $m_t \to 0$. Consider any subsequence $t_k \to \infty$ such that $p_t \Rightarrow p_\infty$ weakly. Then $p_\infty$ is supported on $\mathcal{S}_\varepsilon$ and decomposes as*

$$p_\infty = a_\infty \, p_{\infty,1} + (1 - a_\infty) \, p_{\infty,2},$$

$$p_{\infty,1} := p_\infty(\cdot \mid \mathcal{S}_{1,\varepsilon}), \;\; p_{\infty,2} := p_\infty(\cdot \mid \mathcal{S}_{2,\varepsilon}),$$

*where $a_\infty = p_\infty(\mathcal{S}_{1,\varepsilon})$. If in addition $a_t \to a_\infty \in (0,1)$ and basin separation holds, then $a_\infty$ lies in the leakage interval of Lemma 4, and tends to $q$ as separation grows.*

There is a regime that explains the geometry. If each reward is approximately constant on each basin, then the multiplicative factor in (5) is constant within each basin at every step. This makes the basin-conditionals invariant, and the long-run behavior is governed by the scalar weights $(a_t, b_t)$.

**Proposition 4** (Plateau-basin case). *Fix $\varepsilon > 0$ with $S_{1,\varepsilon} \cap \mathcal{S}_{2,\varepsilon} = \emptyset$ and $p_0(\mathcal{S}_{1,\varepsilon}), p_0(\mathcal{S}_{2,\varepsilon}) > 0$. Assume a plateau structure on each basin, meaning that each $r_i$ is (approximately) constant on each set $\mathcal{S}_{j,\varepsilon}$ so that the multiplicative factor in (5) is constant within each basin (formalized in the Appendix), and assume $m_t \to 0$. Then*

$$p_t(\cdot \mid \mathcal{S}_{1,\varepsilon}) \equiv p_0(\cdot \mid \mathcal{S}_{1,\varepsilon}), \qquad p_t(\cdot \mid \mathcal{S}_{2,\varepsilon}) \equiv p_0(\cdot \mid \mathcal{S}_{2,\varepsilon}),$$

*and any subsequential limit has the explicit form*

$$p_\infty = a_\infty \, p_0(\cdot \mid \mathcal{S}_{1,\varepsilon}) + (1 - a_\infty) \, p_0(\cdot \mid \mathcal{S}_{2,\varepsilon}),$$

*with $a_\infty$ controlled by Lemma 4. In particular, in the hard-separation limit (vanishing leakage) the limiting mixture weight approaches $q$.*

This proposition is illustrated by Fig. 2: pluralistic curation preserves two modes by repeatedly reinforcing two basins, and the final distribution is an explicit mixture.

Next, we turn to the performance guarantees. We verify that the model achieves favorable rewards under both objectives.

**Theorem 5** (Limit of expected rewards). *Assume the setting of Theorem 3 and let $p_{t_k} \Rightarrow p_\infty$. For $i \in \{1, 2\}$,*

$$\mathbb{E}_{p_{t_k}}[r_i(x)] \to \mathbb{E}_{p_\infty}[r_i(x)] = $$
$$a_\infty \, \mathbb{E}_{p_{\infty,1}}[r_i(x)] + (1 - a_\infty) \, \mathbb{E}_{p_{\infty,2}}[r_i(x)].$$

*Moreover, since $p_{\infty,1}$ is supported on $\mathcal{S}_{1,\varepsilon}$ and $p_{\infty,2}$ on $\mathcal{S}_{2,\varepsilon}$, we have $\mathbb{E}_{p_{\infty,1}}[r_1(x)] \geq r_1^* - \varepsilon$ and $\mathbb{E}_{p_{\infty,2}}[r_2(x)] \geq r_2^* - \varepsilon$, while basin separation bounds control the cross-terms $\mathbb{E}_{p_{\infty,1}}[r_2(x)]$ and $\mathbb{E}_{p_{\infty,2}}[r_1(x)]$.*

Expected reward alone does not rule out collapse because a single mode might already be high-reward for one objective. The sharper "no collapse" statement is about variance: a nontrivial mixture of two separated basins forces nonzero reward variance.

**Theorem 6** (Variance preservation). *Assume the setting of Theorem 3 and $a_\infty \in (0,1)$. Then for $i \in \{1, 2\}$,*

$$Var_{p_t}[r_i(x)] \to Var_{p_\infty}[r_i(x)],$$

*and $Var_{p_\infty}[r_i(x)]$ contains an explicit inter-basin term of size $a_\infty(1 - a_\infty)(\mu_{i,1} - \mu_{i,2})^2$, where $\mu_{i,j} := \mathbb{E}_{p_{\infty,j}}[r_i(x)]$. Under basin separation, this yields a lower bound of order*

$$Var_{p_\infty}[r_i(x)] \gtrsim a_\infty(1 - a_\infty) \max\{\Delta_i - 2\varepsilon, 0\}^2.$$

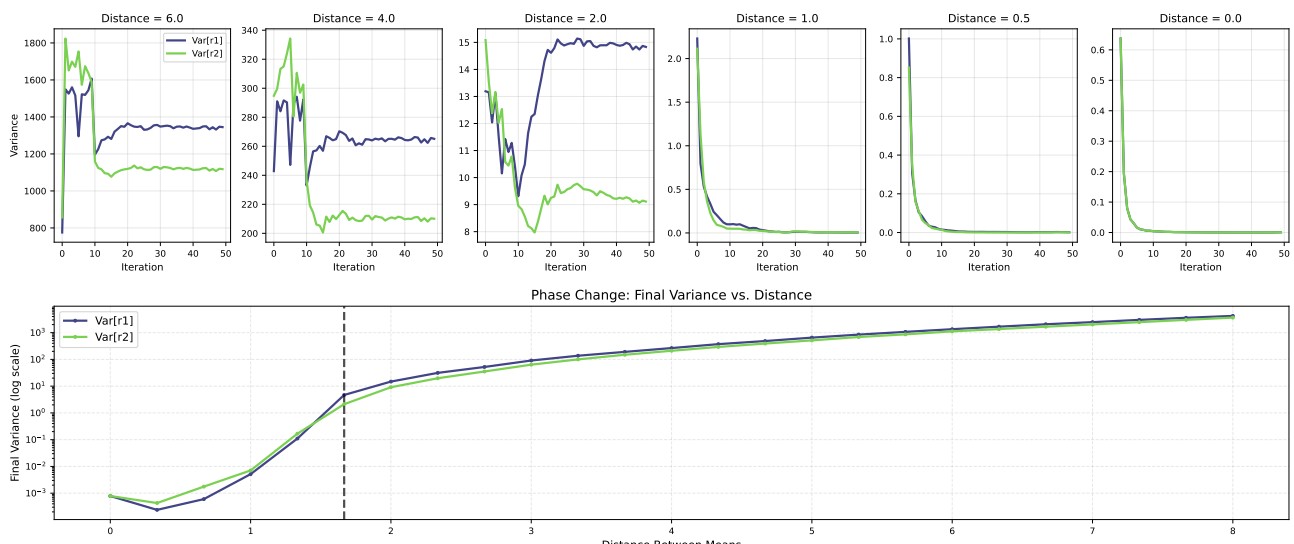

*Figure 3.* **Final distributions and reward variances under pluralistic curation across varying mode distances.** Same synthetic two-mode Gaussian-mixture setting as Figure 2, varying the separation $D$ between the two modes (reward optima). Top: Reward variance trajectories over training iterations for $r_1$ (blue) and $r_2$ (green). Bottom: Log-scale plot of final variances vs. distance.

*Table 1.* **Quality and diversity metrics for synthetic retraining on the CIFAR-10.** The upper block reports results for balanced multi-preference configurations, and the lower block shows two-preference retraining with varying polarization levels $q$.

| Retraining Setting | FID ↓ | Entropy ↑ | KL Divergence ↓ | Feature Variance ↑ | Intra-Class Variance ↑ |
|---|---|---|---|---|---|
| *Balanced multi-preference retraining* | | | | | |
| Two Preferences ($q = 0.5$) | 77.6 | 1.098 | 1.204 | 0.62 | 0.72 |
| Three Preferences ($q = 0.33$) | 64.2 | 1.357 | 0.946 | 0.68 | 0.79 |
| Four Preferences ($q = 0.25$) | 44.5 | 1.670 | 0.632 | 0.74 | **0.86** |
| Five Preferences ($q = 0.20$) | **35.9** | **1.687** | **0.616** | 0.80 | 0.85 |
| *Two-preference retraining with varying polarization $q$* | | | | | |
| $q = 0.0$ (Single Preference) | 100.0 | 0.032 | 2.271 | 0.20 | 0.05 |
| $q = 0.1$ | 90.1 | 0.010 | 2.293 | 0.23 | 0.07 |
| $q = 0.2$ | 85.3 | 0.113 | 2.190 | 0.29 | 0.05 |
| $q = 0.3$ | 83.2 | 0.036 | 2.266 | 0.35 | 0.16 |
| $q = 0.4$ | 80.2 | 0.791 | 1.512 | 0.41 | 0.60 |
| $q = 0.5$ | 77.6 | 1.098 | 1.204 | 0.62 | 0.72 |

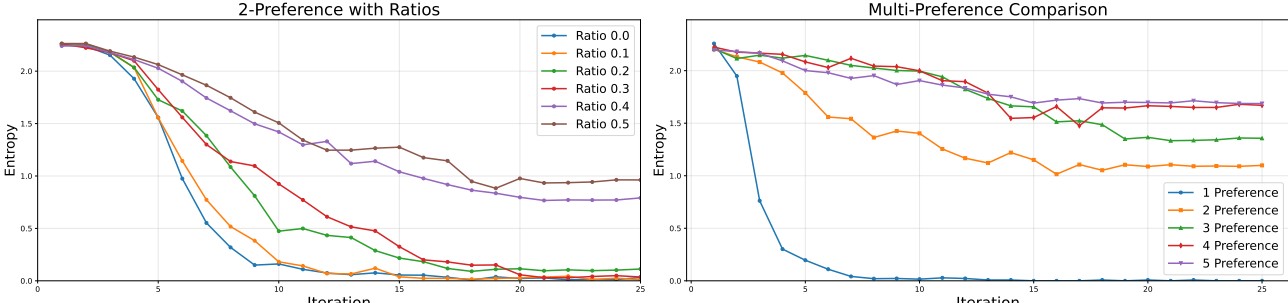

*Figure 4.* **Entropy over training iterations on CIFAR-10 for different retraining configurations.** (Left) two-preference retraining under varying reward-sampling probabilities $q$; (Right) balanced multi-preference retraining with 1 to 5 preferences. Here "Ratio" denotes the reward-sampling probability $q$ used throughout the theory.

This makes the diversity mechanism explicit: as long as both preferences retain influence in the limit and the basins are non-trivially distinct, the limiting distribution cannot become degenerate under either reward.

Finally, we give an interpretation of the mixture weight $q$. Consider the line segment of mixtures between two basin-supported distributions $P_1$ and $P_2$: $p_\alpha := \alpha P_1 + (1 - \alpha)P_2$. Let utilities be $u_i(p) := \mathbb{E}_p[r_i(x)]$ and disagreement points be $d_1 := u_1(P_2)$ and $d_2 := u_2(P_1)$.

**Theorem 7** (Weighted Nash bargaining). *Assume each preference favors its own basin in expectation:* $u_1(P_1) > u_1(P_2)$ *and* $u_2(P_2) > u_2(P_1)$. *The weighted Nash product*

$$\left(u_1(p_\alpha) - d_1\right)^q\left(u_2(p_\alpha) - d_2\right)^{1-q}$$

*is uniquely maximized over* $\alpha \in [0, 1]$ *at* $\alpha^\star = q$.

**Corollary 8** (Bargaining interpretation of the plateau hard-separation). *In the plateau-basin regime of Proposition 4, as leakage vanishes (large separation), the limiting mixture approaches the Nash point* $p_q$ *on the mixture line.*

Rather than fighting the selection pressure, pluralistic curation redirects it. Single-reward retraining sharpens toward one region. Pluralistic retraining still sharpens, but it sharpens toward multiple near-optimal basins and stabilizes as a two-component mixture. In the hard-separation regime, the mixture weight is exactly $q$, and more generally it is tightly controlled by leakage and converges to $q$ as the basins become better separated.

**Extension to multiple preferences.** The update and large-$K$ tilt extend to $M$ rewards with sampling probabilities $\{q_i\}_{i=1}^M$ satisfying $\sum_{i=1}^M q_i = 1$:

$$p_{t+1}(x) = p_t(x) \sum_{i=1}^M q_i H_{p_t}^{K,r_i}(x).$$

Under the outside-domination and cross-basin leakage controls, mass concentrates on $\cup_i S_{i,\varepsilon}$ and limit weights are controlled by leakage, approaching $\{q_i\}$ in the vanishing-leakage regime. Details are deferred to Appendix B.4.

# 4. Experiments

We empirically test whether pluralistic curation (i) prevents collapse under recursive retraining, (ii) yields controllable long-run mixtures under competing rewards, and (iii) exhibits the predicted dependence on preference separation and polarization. We report four core experiments in the main paper: (E1) synthetic GMM retraining dynamics, comparing pluralistic and single-reward curation (Fig. 2); (E2) synthetic phase transition as reward separation varies (Fig. 3); (E3) CIFAR-10 flow retraining under classifier rewards, varying the number of preferences and polarization $q$ (Table 1, Fig. 4); and (E4) text-domain retraining with GPT-2 under conflicting length preferences (Fig. 5). Additional experiments and implementation details are in Appendix C, including: (E5) extended $q$-sweeps and mixture-weight

tracking; (E6) finite-$K$ ablations for the discrete-choice pool; (E7) additional reward-separation visualizations and robustness checks; (E8) leakage corroborations; (E9) baseline comparisons; (E10) CIFAR-10 retraining with learned reward models; and (E11) RLHF-style iterative retraining with helpfulness and safety rewards.

## 4.1. Synthetic Gaussian Mixture

We begin with a 2D Gaussian mixture setup, and use a trainable Gaussian Mixture Model (GMM) (Bishop, 2006) with two reward functions centered at distinct modes to simulate preferences. In each iteration, pluralistic curation selects high-reward samples from candidates drawn from the current model, which is then updated via MLE.

**(E1) Pluralistic curation avoids collapse.** Fig. 2 shows the retraining trajectories. Under pluralistic curation, probability mass remains near both high-reward regions and the reward variances stay bounded away from zero. Under single-reward curation, the distribution concentrates into a narrow region and reward variance collapses, matching the expected single-objective degeneration.

**(E2) Preference separation induces a phase transition.** Fig. 3 varies the distance between reward optima. When optima are well-separated, pluralistic curation yields a stable two-mode mixture with positive variance for both rewards. As optima approach each other, variance drops sharply, and the learned distribution collapses toward a compromise region. Empirically, we observe a clear transition around distance $\approx 2$, supporting the theory that meaningful pluralism requires sufficiently distinct preference basins.

## 4.2. CIFAR-10: Flow-Based Retraining

We evaluate on the CIFAR-10 dataset using the same setup as Ferbach et al. (2024). We train a normalizing flow using *optimal transport conditional flow matching* (OT-CFM) (Lipman et al., 2022; Shaul et al., 2023; Tong et al., 2023). The initial model is pretrained on all 50,000 training images from CIFAR-10. At each retraining iteration, we generate $5 \cdot 10^4$ samples from the current model. From these, we retain $2.5 \cdot 10^3$ samples using a discrete $K$-BT model. Rewards $r(x)$ are derived from class probabilities $\pi_0(x), \ldots, \pi_9(x)$ output by a pretrained VGG11 classifier (Simonyan & Zisserman, 2015), achieving 92.39% accuracy on the CIFAR-10 test set. We define $r(x) = \gamma \cdot \pi_i(x)$ for a fixed class (e.g., *airplanes*). We report FID for sample quality and multiple diversity proxies (class entropy, KL-to-uniform, and feature-level variances). All generator and selection hyperparameters are provided in the Appendix C.1.

**(E3) More preferences improve diversity and quality.** In the balanced regime (uniform sampling across $M$ rewards), Table 1 shows consistent improvements as $M$ increases

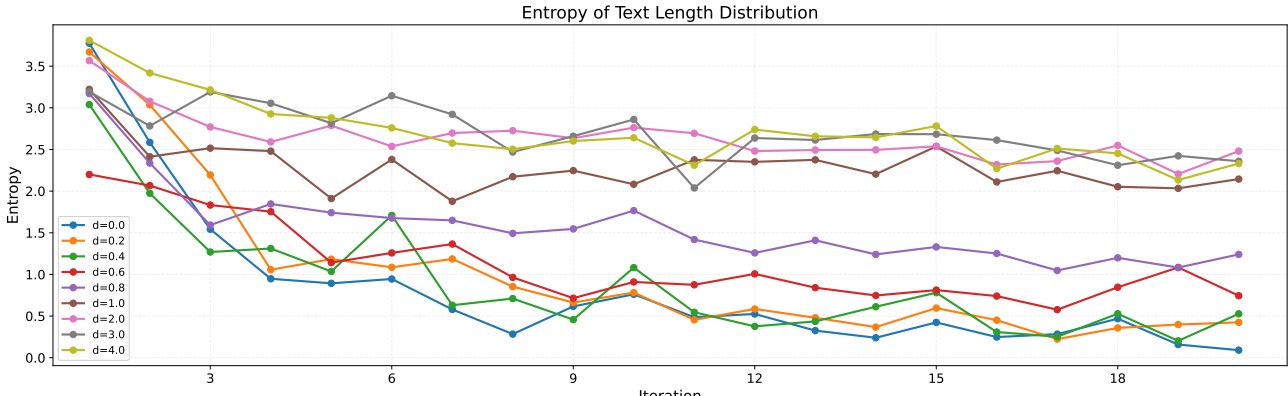

*Figure 5.* **Evolution of output length entropy for text with GPT-2**. Two-preference retraining under varying preferred length distances.

from 2 to 5: entropy increases and KL decreases (broader, more balanced class coverage), and feature and intra-class variances rise (greater perceptual and semantic diversity). These diversity gains coincide with improved FID. Fixing two rewards and varying the polarization parameter $q$, Table 1 shows rapid degradation as $q$ becomes imbalanced: entropy and variance proxies drop and FID worsens, indicating partial collapse toward the dominant preference. Fig. 4 summarizes the dynamics over retraining iterations, where balanced mixtures sustain higher entropy while skewed mixtures collapse toward low-entropy behavior.

The main takeaway is that $q$ controls a trade-off between quality and diversity. When $q$ is close to $0$ or $1$, curation is effectively single-objective and the model collapses toward the dominant preference. Balanced values of $q$ expose the model to both reward basins and preserve higher entropy and intra-class diversity.

### 4.3. Text-Domain Retraining

We test pluralistic curation in a discrete text setting where the preference signal is directly verifiable: output length. We fine-tune a lightweight GPT-2 model in iterative rounds under two length-based preferences with targets $T_A$ and $T_B$, using reward $R(y; T) = -|L(y) - T|$, where $L(y)$ is the word count. We fix $T_A$ and vary $T_B$ to control the distance between the conflicts $d = |T_A - T_B|$. The reward depends only on an interpretable statistic: output length, and changes in entropy can be attributed to the retraining dynamics.

**(E4) Increasing conflict increases sustained uncertainty rather than collapse.** We track the discrete entropy of the generated length distribution, $H(L)$, across retraining iterations. Fig. 5 shows that larger distances $d$ lead to higher sustained entropy. This indicates that, under balanced pluralistic selection, the model does not collapse to a single compromise length, and instead maintains a broader policy that hedges between competing objectives.

## 5. Conclusion

As generative models increasingly rely on synthetic data to scale, the consequences of recursive retraining demand attention. This paper challenges the prevailing view that such training loops inevitably lead to collapse, showing instead that diversity can be preserved when data are curated using pluralistic preferences. Our analysis demonstrates that using multiple curation rewards provably prevents collapse, yielding a stable multimodal distribution that satisfies a weighted Nash bargaining solution. We validated these results across synthetic and real-world domains, observing improvements in both inter- and intra-mode diversity.

Our findings align with current practices in RLHF, where models are often fine-tuned using multiple alignment objectives such as helpfulness, harmlessness, and honesty. In many of these pipelines, objectives are applied sequentially or stochastically, and trade-offs between them are implicitly encoded through reward aggregation or classifier mixing. Our pluralistic curation experiments simulate this regime: when one objective dominates (e.g., excessive emphasis on harmlessness), diversity and generalization quickly degrade, mirroring the reward overoptimization and collapse seen in alignment drift. Conversely, when objectives are balanced, the system stabilizes, supporting richer and more inclusive outputs. This suggests that pluralistic curation captures the same tensions as RLHF multi-objective post-training, but makes them explicit and theoretically interpretable. More broadly, pluralistic retraining acts as a regularizer, resisting overfitting to narrow alignment signals and maintaining robustness across evolving user preferences. Much like weight decay constrains model complexity, reward pluralism constrains preference collapse. In fully synthetic regimes where human data are scarce, this becomes especially critical: pluralism offers a theoretically sound alternative to human oversight. This work therefore opens new avenues for aligning generative models with multifaceted human values, without sacrificing diversity and quality.

## Impact Statement

This paper presents work whose goal is to advance the field of Machine Learning by improving our theoretical understanding of recursive retraining on curated synthetic data. There are many potential societal consequences of our work, none which we believe must be specifically highlighted here.

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

# Appendix

**Appendix map.** Appendix A presents related work. Appendix B contains assumptions and proofs. Appendix C contains additional experiments and implementation details.

## A. Related Work

This section situates our work within key areas of literature: mode collapse in generative models, iterative retraining with synthetic data, game-theoretic methods for generative modeling, multi-objective optimization in generative frameworks, and data curation strategies. Our work, *pluralistic curation*, proposes alternating between conflicting reward signals during retraining to prevent mode collapse and maintain sample diversity.

**Mode Collapse in Generative Models.** Mode collapse remains a central problem in generative modeling, especially in GANs, where generators often fail to represent the diversity of the training distribution (Goodfellow et al., 2014). Mitigation strategies include architectural innovations like Unrolled GANs (Metz et al., 2016), objective reformulations such as Wasserstein GANs (Arjovsky et al., 2017), and regularization techniques like gradient penalties (Gulrajani et al., 2017) or spectral normalization (Miyato et al., 2018). Other methods like mini-batch discrimination (Salimans et al., 2016) and conditional GANs (Mirza & Osindero, 2014) promote diversity by controlling conditioning information or evaluating sets of outputs jointly.

While GANs are most associated with the term "mode collapse," similar diversity degradation has been observed in other classes of generative models. In diffusion models, diversity loss may arise from over-simplified denoising objectives or poor classifier-free guidance tuning (Dhariwal & Nichol, 2021). Attempts to counteract this include improved noise schedules (Nichol & Dhariwal, 2021), adaptive sampling strategies (Watson et al., 2022), and regularized training pipelines that penalize collapse to dominant trajectories (Vahdat et al., 2021). In autoregressive models, such as large language models or image transformers, collapse can manifest as reduced entropy or over-optimization on narrow feedback loops, a phenomenon sometimes called recursion collapse or alignment collapse (Shumailov et al., 2023; Ferbach et al., 2024). Techniques such as temperature scaling, entropy-based sampling, or human feedback tuning aim to address this, though often heuristically.

Importantly, most of these solutions are designed to improve initial training diversity. Our work addresses a distinct failure mode: mode collapse during *iterative retraining*, especially when new training data are drawn from the model itself and filtered by reward-based curation. In this self-consuming regime, even models with initially broad support can collapse over time if curation is biased toward a singular objective. Our paper focuses on preventing this collapse via principled multi-preference curation.

**Iterative Retraining and Self-Consuming Models.** Recent studies explore generative models that retrain on their own outputs. (Ferbach et al., 2024; Shumailov et al., 2023; Gerstgrasser et al., 2024; Alemohammad et al., 2023) introduced the notion of *self-consuming models*, where outputs are curated using a single reward function to fine-tune the model iteratively. Despite optimizing the expected reward, this process provably leads to diminishing output variance and eventual mode collapse. Their proposed mitigation, mixing real data with curated output, provides a partial remedy but fails to fully restore diversity. (Zhao et al., 2025) models heterogeneity as additive noise around a single reward function where annotators share a common preference but differ by a stochastic perturbation. Our setup is fundamentally different: we study distinct, competing reward functions with potentially disjoint basins, where no single underlying preference exists. Their results characterize stability properties of a unimodal limit, not the mixture dynamics.

We extend their theoretical model by adopting a pluralistic strategy that alternates between two reward functions, each representing different user preferences. This approach ensures that the learned distribution maintains coverage over a broader set of desirable outputs.

**Game-Theoretic Approaches in Generative Modeling.** Classical GANs are founded on a two-player minimax game between the generator and the discriminator (Goodfellow et al., 2014). Subsequent work generalized this to broader game-theoretic formulations. For instance, (Franci & Grammatico, 2021) reformulated GANs as a stochastic Nash equilibrium, while (Oliehoek et al., 2018) treated them as finite zero-sum games with provable convergence properties.

In contrast to these initial-training settings, our pluralistic curation uses a game-theoretic lens during retraining. We pose the

curation process as a *weighted Nash bargaining* problem between competing objectives, using polarization as a tunable compromise. This distinction sets our paper apart from standard adversarial learning dynamics.

**Multi-Objective Optimization in Generative Models.**    Balancing competing objectives is crucial in domains such as molecule generation (Abeer et al., 2024) and multi-goal reinforcement learning (Zhou et al., 2020). Prior work optimizes Pareto fronts or uses scalarization methods in training VAEs or RL agents to balance different desiderata. While effective, these methods often assume fixed data and static tasks.

In contrast, we operate in a dynamic retraining regime, alternating between reward functions across time to guide synthetic data curation. Our method preserves sample diversity without requiring explicit Pareto frontier modeling, making it simple yet expressive.

**Multiple Discriminators and Model Perspectives.**    To encourage output diversity, some GAN variants use multiple discriminators (Gemp & Mahadevan, 2018; Intrator et al., 2018; Albuquerque et al., 2019). These systems evaluate samples from multiple perspectives, either to decompose the data distribution or to form ensemble gradients. Such architectural strategies improve robustness, but often require expensive retraining or coordination across components.

Pluralistic curation avoids architectural changes by instead modifying the data selection process during retraining. It is therefore complementary to multi-discriminator methods, and more lightweight in implementation.

**Data Curation and Synthetic Feedback Loops.**    Data curation underpins successful retraining and has been studied in contexts from supervised learning (Chen et al., 2024) to vision-language modeling (Xu et al., 2023a). (Ferbach et al., 2024) used the Bradley–Terry model to rank synthetic outputs, forming a curation loop. However, their reliance on a single reward led to homogenization. Our extension alternates between distinct rewards to maintain exploration and diversity.

While prior curation frameworks focus on static data or pretraining, ours targets the dynamic regime of synthetic sample loops, where diversity preservation is critical. This difference makes our method particularly relevant to future closed-loop generative systems aligned with human intent.

**Summary.**    By incorporating ideas from game theory, multi-objective optimization, and data curation, our pluralistic curation introduces a practical and theoretically grounded mechanism to prevent mode collapse in retrained generative models. It is uniquely positioned at the intersection of reward-driven retraining and generative diversity preservation.

# B. Proofs

## B.1. Notation Table

*Table 2.* Notation for the theoretical analysis.

| Symbol | Description |
|---|---|
| $\mathcal{X} \subseteq \mathbb{R}^d$ | Sample space. |
| $x \in \mathcal{X}$ | A sample. |
| $p_0, \ p_t$ | Model distribution at initialization and at retraining step $t$. |
| $x_{1:K} = (x_1, \ldots, x_K)$ | $K$ i.i.d. candidates drawn from $p_t$. |
| $\hat{x}$ | Selected candidate after Bradley–Terry curation. |
| $K \geq 2$ | Candidate pool size. |
| $q \in (0, 1)$ | Probability of using reward $r_1$; reward $r_2$ is used with probability $1 - q$. |
| $r_1, r_2 : \mathcal{X} \to \mathbb{R}$ | Reward functions. |
| $r_i^* := \sup_{x \in \mathcal{X}} r_i(x)$ | Supremum reward value for preference $i$. |
| $L_i, U_i$ | Reward bounds: $L_i \leq r_i(x) \leq U_i$ for all $x \in \mathcal{X}$. |
| $S_{i,\varepsilon}$ | $\varepsilon$-optimal basin for reward $r_i$: $S_{i,\varepsilon} := \{x : \ r_i(x) \geq r_i^* - \varepsilon\}$. |
| $\mathcal{S}_\varepsilon$ | Union of modeled near-optimal basins: $\mathcal{S}_\varepsilon := S_{1,\varepsilon} \cup S_{2,\varepsilon}$. |
| $O_\varepsilon$ | Outside region: $O_\varepsilon := \mathcal{X} \setminus \mathcal{S}_\varepsilon$. |
| $a_t, b_t, m_t$ | Basin and outside masses: $a_t := p_t(S_{1,\varepsilon})$, $b_t := p_t(S_{2,\varepsilon})$, $m_t := p_t(O_\varepsilon)$. When the basins are disjoint, $a_t + b_t + m_t = 1$. |
| $H_p^{K,r}(x)$ | Finite-$K$ BT curation weight: $\mathbb{E}_{y_{1:K-1} \sim p}\left[ \frac{K e^{r(x)}}{e^{r(x)} + \sum_{j=1}^{K-1} e^{r(y_j)}} \right]$. |
| $H_{\infty,r}^p(x)$ | Large-$K$ exponential-tilt weight: $H_{\infty,r}^p(x) := e^{r(x)} / \mathbb{E}_p[e^r]$. |
| $Z_i(t), \ \mu_i(t)$ | Normalizer for reward $r_i$: $Z_i(t) = \mu_i(t) := \mathbb{E}_{p_t}[e^{r_i(x)}]$. |
| $W_t(x)$ | Large-$K$ pluralistic update multiplier: $W_t(x) := q e^{r_1(x)}/Z_1(t) + (1 - q)e^{r_2(x)}/Z_2(t)$. |
| $p_t^{(i)}$ | Reward-$i$ tilted distribution: $p_t^{(i)}(dx) := e^{r_i(x)} p_t(dx)/Z_i(t)$. |
| $\rho_\varepsilon$ | Outside-domination contraction factor: $\sup_{x \in O_\varepsilon} W_t(x) \leq \rho_\varepsilon < 1$, implying $m_{t+1} \leq \rho_\varepsilon m_t$. |
| $\Delta_1, \Delta_2$ | Cross-basin separation gaps, e.g. $x \in S_{2,\varepsilon} \Rightarrow r_1(x) \leq r_1^* - \Delta_1 + \varepsilon$, and symmetrically for $r_2$ on $S_{1,\varepsilon}$. |
| $\kappa_1, \kappa_2$ | Leakage factors: $\kappa_1 := \exp(-(\Delta_1 - 2\varepsilon))$ and $\kappa_2 := \exp(-(\Delta_2 - 2\varepsilon))$. |
| $p_{\infty,i}$ | Limit conditional distribution on basin $S_{i,\varepsilon}$: $p_{\infty,i} := p_\infty(\cdot \mid S_{i,\varepsilon})$. |
| $\mu_{i,j}$ | Basin-conditional reward mean: $\mu_{i,j} := \mathbb{E}_{p_{\infty,j}}[r_i]$. |
| $(z)_+$ | Positive part: $(z)_+ := \max\{z, 0\}$. |
| $p_\alpha$ | Mixture family for bargaining: $p_\alpha := \alpha P_1 + (1 - \alpha)P_2$, $\alpha \in [0, 1]$. |
| $P_1, P_2$ | Fixed distributions supported on $S_{1,\varepsilon}$ and $S_{2,\varepsilon}$. |
| $u_i(p)$ | Utility of preference $i$: $u_i(p) := \mathbb{E}_p[r_i]$. |
| $d_1, d_2$ | Disagreement utilities: $d_1 := u_1(P_2)$ and $d_2 := u_2(P_1)$. |
| $\mathcal{N}(\alpha)$ | Weighted Nash product: $\mathcal{N}(\alpha) := (u_1(p_\alpha) - d_1)^q (u_2(p_\alpha) - d_2)^{1-q}$. |
| $\alpha^\star$ | Maximizer of the weighted Nash product; in the two-basin bargaining result, $\alpha^\star = q$. |

## B.2. Assumptions

We restate the standing conditions used throughout the proofs and record a few technical variants used by specific results. Throughout, $\mathcal{X} \subseteq \mathbb{R}^d$ and $r_1, r_2 : \mathcal{X} \to \mathbb{R}$ are reward functions. For $\varepsilon > 0$, define the $\varepsilon$-optimal regions

$$S_{i,\varepsilon} := \{x \in \mathcal{X} : \ r_i(x) \geq r_i^* - \varepsilon\}, \qquad \mathcal{S}_\varepsilon := S_{1,\varepsilon} \cup S_{2,\varepsilon}, \qquad r_i^* := \sup_{x \in \mathcal{X}} r_i(x).$$

We track the masses

$$a_t := p_t(S_{1,\varepsilon}), \qquad b_t := p_t(S_{2,\varepsilon}), \qquad m_t := p_t(\mathcal{X} \setminus \mathcal{S}_\varepsilon), \qquad a_t + b_t + m_t = 1.$$

**Assumption B.1** (Regularity and bounded rewards). *The rewards are measurable and bounded: there exist $L_i, U_i \in \mathbb{R}$ such that*

$$L_i \leq r_i(x) \leq U_i \quad \text{for all } x \in \mathcal{X}, \; i \in \{1, 2\}.$$

*In particular, $r_i^* < \infty$ and we may take $U_i = r_i^*$ without loss of generality.*

**Assumption B.2** (Nontrivial initialization on both basins). *Fix $\varepsilon > 0$ and suppose*

$$p_0(S_{1,\varepsilon}) > 0, \qquad p_0(S_{2,\varepsilon}) > 0.$$

**Assumption B.3** ($\varepsilon$-basin disjointness). *Fix $\varepsilon > 0$ and suppose the near-optimal regions are disjoint:*

$$S_{1,\varepsilon} \cap S_{2,\varepsilon} = \emptyset.$$

**Assumption B.4** (Outside domination). *Fix $\varepsilon > 0$ and define*

$$Z_i(t) := \mathbb{E}_{p_t}\left[e^{r_i(x)}\right], \qquad i \in \{1, 2\}.$$

*There exists a constant $\rho_\varepsilon \in (0, 1)$ such that, for all $t \geq 0$,*

$$\sup_{x \notin \mathcal{S}_\varepsilon}\left[q\,\frac{e^{r_1(x)}}{Z_1(t)} + (1-q)\,\frac{e^{r_2(x)}}{Z_2(t)}\right] \leq \rho_\varepsilon. \tag{OD}$$

**Assumption B.5** (Cross-basin separation (leakage margins)). *Fix $\varepsilon > 0$ and suppose Assumption B.3 holds. There exist gaps $\Delta_1, \Delta_2 > 0$ such that*

$$x \in S_{1,\varepsilon} \;\Rightarrow\; r_2(x) \leq r_2^* - \Delta_2 + \varepsilon, \qquad x \in S_{2,\varepsilon} \;\Rightarrow\; r_1(x) \leq r_1^* - \Delta_1 + \varepsilon.$$

*Equivalently, points that are $\varepsilon$-optimal for one preference are uniformly suboptimal for the other. When $\Delta_i > 2\varepsilon$, this yields the leakage factors*

$$\kappa_1 := \exp\bigl(-(\Delta_1 - 2\varepsilon)\bigr), \qquad \kappa_2 := \exp\bigl(-(\Delta_2 - 2\varepsilon)\bigr),$$

*that appear in the basin-mass bounds.*

**Assumption B.6** (Plateau-basin idealization (used only for explicit mixture and bargaining)). *Fix $\varepsilon > 0$ and suppose Assumptions B.2–B.3 hold. Assume rewards are (approximately) constant on each $\varepsilon$-basin: there exist constants $U_1, U_2 \in \mathbb{R}$ and gaps $\Delta_1, \Delta_2 > 0$ such that*

$$r_1(x) = U_1, \quad r_2(x) = U_2 - \Delta_2 \quad \text{for all } x \in S_{1,\varepsilon},$$

$$r_1(x) = U_1 - \Delta_1, \quad r_2(x) = U_2 \quad \text{for all } x \in S_{2,\varepsilon}.$$

*This idealization makes the basin-conditional distributions invariant and yields the explicit $q$-mixture and Nash bargaining characterization.*

**Interpretation.** Assumption B.4 is a basin-completeness condition. The set $\mathcal{S}_\varepsilon$ is intended to contain all regions reinforced by the pluralistic curation rule. Thus, points outside $\mathcal{S}_\varepsilon$ should have multiplicative update factor uniformly below one. If an outside point violates this condition, then it is not genuinely low-reward background mass: it is an additional compromise basin receiving support from multiple rewards, and the correct formulation is to include it in $\mathcal{S}_\varepsilon$ or analyze it using the multi-basin extension.

Assumption B.5 controls cross-reward reinforcement between the two modeled basins. It quantifies how much reward $r_1$ can reinforce $S_{2,\varepsilon}$ and how much reward $r_2$ can reinforce $S_{1,\varepsilon}$. Together, Assumptions B.4 and B.5 separate two mechanisms: outside mass decay and within-basin mass allocation. Assumption B.6 is used only for the clean closed-form mixture and bargaining interpretation.

### B.3. Proofs

**Lemma B.1** (Pluralistic curation update). *At step $t$, draw $R \in \{r_1, r_2\}$ independently of the candidate pool, with $\Pr(R = r_1) = q$ and $\Pr(R = r_2) = 1 - q$. Under the BT selection rule and the infinite-capacity MLE retraining idealization,*

$$p_{t+1}(x) = p_t(x)\Big(q\, H_{p_t}^{K,r_1}(x) + (1-q)\, H_{p_t}^{K,r_2}(x)\Big),$$

*where*

$$H_{p_t}^{K,r_i}(x) := \mathbb{E}_{y_{1:K-1}\sim p_t}\left[\frac{Ke^{r_i(x)}}{e^{r_i(x)} + \sum_{j=1}^{K-1} e^{r_i(y_j)}}\right], \qquad i \in \{1,2\}.$$

*Proof.* Condition on the reward being $r_i$. If $x_{1:K} \sim p_t^{\otimes K}$, then BT selection gives

$$\Pr(\hat{x} = x_k \mid x_{1:K}, r_i) = \frac{e^{r_i(x_k)}}{\sum_{\ell=1}^{K} e^{r_i(x_\ell)}}.$$

By exchangeability, the selected-sample density under reward $r_i$ is

$$p_{\hat{x}|r_i}(x) = p_t(x)\, \mathbb{E}_{y_{1:K-1}\sim p_t}\left[\frac{Ke^{r_i(x)}}{e^{r_i(x)} + \sum_{j=1}^{K-1} e^{r_i(y_j)}}\right] = p_t(x)H_{p_t}^{K,r_i}(x).$$

Averaging over $R$ therefore yields

$$p_{\hat{x}}(x) = q\, p_{\hat{x}|r_1}(x) + (1-q)\, p_{\hat{x}|r_2}(x) = p_t(x)\Big(qH_{p_t}^{K,r_1}(x) + (1-q)H_{p_t}^{K,r_2}(x)\Big).$$

The MLE retraining idealization sets $p_{t+1} = p_{\hat{x}}$, giving the claim. $\qquad\square$

**Lemma B.2** (Concentration of curation weights). *Let $r : \mathcal{X} \to \mathbb{R}$ be bounded and measurable, and let $p$ be a probability density. Then, for every $x \in \mathcal{X}$,*

$$H_p^{K,r}(x) \to H_{\infty,r}^p(x) := \frac{e^{r(x)}}{\mathbb{E}_{y\sim p}[e^{r(y)}]} \qquad \text{as } K \to \infty.$$

*Proof.* Fix $x \in \mathcal{X}$ and set $Y_j = e^{r(y_j)}$, where $y_j \overset{\text{i.i.d.}}{\sim} p$. Since $r$ is bounded, $\mu_e := \mathbb{E}_p[e^r] \in (0,\infty)$ and $(K-1)^{-1}\sum_{j=1}^{K-1} Y_j \to \mu_e$ almost surely. Hence

$$\frac{Ke^{r(x)}}{e^{r(x)} + \sum_{j=1}^{K-1} Y_j} = \frac{e^{r(x)}}{e^{r(x)}/K + \frac{K-1}{K}\cdot(K-1)^{-1}\sum_{j=1}^{K-1} Y_j} \to \frac{e^{r(x)}}{\mu_e} \quad \text{a.s.}$$

Moreover, if $r_{\min} \leq r \leq r_{\max}$, the integrand is uniformly bounded by $e^{r_{\max}-r_{\min}}$. Dominated convergence therefore gives

$$H_p^{K,r}(x) \to \frac{e^{r(x)}}{\mu_e} = \frac{e^{r(x)}}{\mathbb{E}_{y\sim p}[e^{r(y)}]}.$$

$\qquad\square$

**Lemma B.3** (Finite-$K$ approximation). *Let $r : \mathcal{X} \to \mathbb{R}$ be bounded, with $r_{\min} \leq r(x) \leq r_{\max}$, and set $A := e^{r_{\min}}$, $B := e^{r_{\max}}$, and $\mu_e := \mathbb{E}_{y\sim p}[e^{r(y)}] \in [A, B]$. Then, for all $K \geq 3$ and all $x \in \mathcal{X}$,*

$$\left|H_p^{K,r}(x) - H_{\infty,r}^p(x)\right| \leq C_1\sqrt{\frac{\log K}{K}} + \frac{C_2}{K},$$

*where $C_1, C_2$ depend only on $A$ and $B$.*

*Proof.* Fix $x \in \mathcal{X}$ and write $Y_j := e^{r(y_j)} \in [A, B]$, where $y_j \stackrel{\text{i.i.d.}}{\sim} p$. Let $\overline{Y} := (K-1)^{-1} \sum_{j=1}^{K-1} Y_j$. Then

$$H_p^{K,r}(x) = \mathbb{E}\left[\frac{e^{r(x)}}{e^{r(x)}/K + \frac{K-1}{K}\overline{Y}}\right], \qquad H_{\infty,r}^p(x) = \frac{e^{r(x)}}{\mu_e}.$$

For any $\eta > 0$, Hoeffding's inequality gives

$$\Pr\left(|\overline{Y} - \mu_e| \geq \eta\right) \leq 2\exp\left(-\frac{2(K-1)\eta^2}{(B-A)^2}\right).$$

On the event $E_\eta := \{|\overline{Y} - \mu_e| < \eta\}$, and for $\eta \leq A/2$, the map $z \mapsto e^{r(x)}/(e^{r(x)}/K + ((K-1)/K)z)$ is uniformly Lipschitz on $[A/2, B]$ with a constant depending only on $A, B$. Hence, on $E_\eta$,

$$\left|\frac{e^{r(x)}}{e^{r(x)}/K + \frac{K-1}{K}\overline{Y}} - \frac{e^{r(x)}}{\mu_e}\right| \leq \frac{C'}{K} + C''\eta,$$

for constants $C', C''$ depending only on $A, B$. On $E_\eta^c$, both terms are bounded by constants depending only on $A, B$, so taking expectations yields

$$\left|H_p^{K,r}(x) - H_{\infty,r}^p(x)\right| \leq \frac{C'}{K} + C''\eta + C''' \Pr(E_\eta^c).$$

Choosing $\eta := (B-A)\sqrt{\log K/(2(K-1))}$ gives $\Pr(E_\eta^c) \leq 2/K$. Absorbing constants and using $(K-1)^{-1} \leq 2K^{-1}$ for $K \geq 3$ gives

$$\left|H_p^{K,r}(x) - H_{\infty,r}^p(x)\right| \leq C_1 \sqrt{\frac{\log K}{K}} + \frac{C_2}{K}.$$

For the finitely many small values of $K$ for which the chosen $\eta$ may exceed $A/2$, the same bound holds after enlarging the constants. $\qquad\square$

**Lemma B.4** (Decay outside the $\varepsilon$-optimal regions). *Fix $\varepsilon > 0$ and work in the large-$K$ regime. Let*

$$W_t(x) := q\,\frac{e^{r_1(x)}}{Z_1(t)} + (1-q)\,\frac{e^{r_2(x)}}{Z_2(t)}, \qquad Z_i(t) := \mathbb{E}_{p_t}[e^{r_i(x)}],$$

*so that the update is $p_{t+1} = W_t p_t$. Suppose there exists $\rho_\varepsilon \in (0, 1)$ such that*

$$\sup_{x \notin \mathcal{S}_\varepsilon} W_t(x) \leq \rho_\varepsilon \qquad \text{for all } t \geq 0. \tag{OD}$$

*Then, with $m_t := p_t(\mathcal{X} \setminus \mathcal{S}_\varepsilon)$, $m_{t+1} \leq \rho_\varepsilon m_t$ for all $t$. Consequently, $m_t \leq \rho_\varepsilon^t m_0 \to 0$, and hence $p_t(\mathcal{S}_\varepsilon) \to 1$.*

*Proof.* Let $O_\varepsilon := \mathcal{X} \setminus \mathcal{S}_\varepsilon$ and write

$$W_t(x) := q\,\frac{e^{r_1(x)}}{Z_1(t)} + (1-q)\,\frac{e^{r_2(x)}}{Z_2(t)}$$

for the multiplicative factor in the large-$K$ update, so that $p_{t+1}(x) = p_t(x)W_t(x)$. By the outside-domination assumption, $W_t(x) \leq \rho_\varepsilon$ for all $x \in O_\varepsilon$. Hence

$$m_{t+1} = \int_{O_\varepsilon} p_t(x)W_t(x)\,dx \leq \rho_\varepsilon p_t(O_\varepsilon) = \rho_\varepsilon m_t.$$

Iterating gives $m_t \leq \rho_\varepsilon^t m_0$, which tends to zero since $\rho_\varepsilon < 1$. Therefore $p_t(\mathcal{S}_\varepsilon) = 1 - m_t \to 1$. $\qquad\square$

**Lemma B.5** (Leakage-controlled basin mass). *Work in the $K \to \infty$ regime and fix $\varepsilon > 0$. Suppose $S_{1,\varepsilon} \cap S_{2,\varepsilon} = \emptyset$ and, for some $\Delta_1, \Delta_2 > 2\varepsilon$,*

$$x \in S_{1,\varepsilon} \Rightarrow r_1(x) \geq U_1 - \varepsilon, \quad r_2(x) \leq U_2 - \Delta_2 + \varepsilon,$$

*and*

$$x \in S_{2,\varepsilon} \Rightarrow r_2(x) \geq U_2 - \varepsilon, \quad r_1(x) \leq U_1 - \Delta_1 + \varepsilon.$$

Let $\kappa_1 := e^{-(\Delta_1 - 2\varepsilon)}$ and $\kappa_2 := e^{-(\Delta_2 - 2\varepsilon)}$. If $m_t \to 0$ and $a_t := p_t(S_{1,\varepsilon})$ converges to $a_\infty \in (0, 1)$, then

$$\max\left\{0, \frac{q - \kappa_1}{1 - \kappa_1}\right\} \leq a_\infty \leq \min\left\{1, \frac{q}{1 - \kappa_2}\right\}.$$

In particular, when $q > \kappa_1$ and $1 - q > \kappa_2$, the limiting mass is bounded away from both $0$ and $1$. As $\Delta_1, \Delta_2 \to \infty$, equivalently $\kappa_1, \kappa_2 \to 0$, the bounds collapse to $a_\infty = q$.

*Proof.* Write $S_i = S_{i,\varepsilon}$ and $O = \mathcal{X} \setminus (S_1 \cup S_2)$. In the large-$K$ regime,

$$p_{t+1} = q\, \frac{e^{r_1}}{\mu_1(t)} p_t + (1 - q)\, \frac{e^{r_2}}{\mu_2(t)} p_t, \qquad \mu_i(t) := \int_{\mathcal{X}} e^{r_i}\, dp_t.$$

Let $p_t^{(i)}$ denote the $r_i$-tilted distribution, $p_t^{(i)}(A) := \mu_i(t)^{-1} \int_A e^{r_i}\, dp_t$. Then $a_{t+1} = q\, p_t^{(1)}(S_1) + (1 - q)\, p_t^{(2)}(S_1)$.

We first lower-bound the contribution of the $r_1$-tilt to its own basin. By the separation assumptions, the $r_1$-weight on $S_2$ is at most $\kappa_1$ times the corresponding own-basin scale, while the outside region contributes at most $e^\varepsilon m_t$ on the same scale. Hence

$$p_t^{(1)}(S_1) \geq \frac{a_t}{a_t + \kappa_1 b_t + e^\varepsilon m_t}.$$

Therefore

$$a_{t+1} \geq q\, \frac{a_t}{a_t + \kappa_1 b_t + e^\varepsilon m_t}.$$

Taking $t \to \infty$, using $m_t \to 0$, $a_t \to a_\infty$, and $b_t \to 1 - a_\infty$, gives

$$a_\infty \geq q\, \frac{a_\infty}{a_\infty + \kappa_1(1 - a_\infty)}.$$

Since $a_\infty > 0$, this implies

$$a_\infty \geq \frac{q - \kappa_1}{1 - \kappa_1}.$$

The upper bound follows symmetrically from the $r_2$-tilt. Namely,

$$p_t^{(2)}(S_2) \geq \frac{b_t}{b_t + \kappa_2 a_t + e^\varepsilon m_t},$$

and since $b_{t+1} \geq (1 - q)p_t^{(2)}(S_2)$, passing to the limit yields

$$1 - a_\infty \geq (1 - q)\frac{1 - a_\infty}{1 - a_\infty + \kappa_2 a_\infty}.$$

Because $a_\infty \in (0, 1)$, this is equivalent to $a_\infty \leq q/(1 - \kappa_2)$. Clipping the two bounds by $0$ and $1$ gives the stated interval. Finally, $\kappa_i \to 0$ as $\Delta_i \to \infty$, so both endpoints converge to $q$. $\qquad\square$

**Lemma B.6** (Uniform non-collapse under bounded leakage). *Fix $\varepsilon > 0$ and work in the large-$K$ regime. Assume $p_0(S_{1,\varepsilon}) > 0$, $p_0(S_{2,\varepsilon}) > 0$, and $m_t \to 0$. Let*

$$\kappa_1 := \exp\big(-(\Delta_1 - 2\varepsilon)\big), \qquad \kappa_2 := \exp\big(-(\Delta_2 - 2\varepsilon)\big),$$

*and suppose $q \in (\kappa_1, 1 - \kappa_2)$. Define*

$$L := \frac{q - \kappa_1}{1 - \kappa_1}, \qquad U := \frac{q}{1 - \kappa_2}.$$

*Then*

$$\liminf_{t \to \infty} a_t \geq L, \qquad \limsup_{t \to \infty} a_t \leq U.$$

*In particular, there exist $\eta > 0$ and $t_0$ such that*

$$\eta \leq a_t \leq 1 - \eta \qquad \text{for all } t \geq t_0.$$

*Proof.* Write $A := S_{1,\varepsilon}$, $B := S_{2,\varepsilon}$, and $O := \mathcal{X} \setminus (A \cup B)$. Also write $a_t = p_t(A)$, $b_t = p_t(B)$, and $m_t = p_t(O)$. In the large-$K$ regime,

$$p_{t+1} = q\, p_t^{(1)} + (1-q)\, p_t^{(2)}, \qquad p_t^{(i)}(dx) := \frac{e^{r_i(x)}}{Z_i(t)} p_t(dx).$$

We first prove the lower bound. By the leakage comparison used in Lemma B.5, there is a constant $C_\varepsilon > 0$ such that

$$p_t^{(1)}(A) \geq \frac{a_t}{a_t + \kappa_1 b_t + C_\varepsilon m_t} \geq \frac{a_t}{a_t + \kappa_1(1 - a_t) + C_\varepsilon m_t}.$$

Hence

$$a_{t+1} \geq q\, \frac{a_t}{a_t + \kappa_1(1 - a_t) + C_\varepsilon m_t}.$$

Fix any $\delta \in (0, L)$. Since $m_t \to 0$, there exists $T$ such that $C_\varepsilon m_t \leq (1 - \kappa_1)\delta/2$ for all $t \geq T$. If $a_t \leq L - \delta$ and $t \geq T$, then

$$a_t + \kappa_1(1 - a_t) + C_\varepsilon m_t \leq q - \frac{1 - \kappa_1}{2}\delta.$$

Therefore

$$a_{t+1} \geq \frac{q}{q - \frac{1 - \kappa_1}{2}\delta}\, a_t.$$

The multiplicative factor is strictly larger than one. Since $a_T > 0$, this implies that after finitely many additional steps, $a_t > L - \delta$. Moreover, once $a_t \geq L - \delta$, monotonicity of the right-hand side gives $a_{t+1} \geq L - \delta$. Thus $a_t \geq L - \delta$ for all sufficiently large $t$. Because $\delta > 0$ was arbitrary, $\liminf_t a_t \geq L$.

The upper bound follows by applying the same argument to $b_t$. The symmetric leakage comparison gives

$$b_{t+1} \geq (1 - q)\frac{b_t}{b_t + \kappa_2(1 - b_t) + C_\varepsilon m_t}.$$

The corresponding positive fixed point is

$$L_b := \frac{(1 - q) - \kappa_2}{1 - \kappa_2} = 1 - \frac{q}{1 - \kappa_2} = 1 - U.$$

By the previous argument, $\liminf_t b_t \geq L_b$. Since $a_t = 1 - b_t - m_t$ and $m_t \to 0$, we obtain

$$\limsup_{t \to \infty} a_t \leq 1 - L_b = U.$$

Finally, because $q \in (\kappa_1, 1 - \kappa_2)$, we have $L > 0$ and $U < 1$. Choose

$$\eta := \frac{1}{2}\min\{L, 1 - U\} > 0.$$

The liminf and limsup bounds imply that, for all sufficiently large $t$, $\eta \leq a_t \leq 1 - \eta$. $\qquad\square$

**Corollary B.1** (Nondegenerate limit points). *Under the assumptions of Lemma B.6, let $t_k \to \infty$ and suppose $p_{t_k} \Rightarrow p_\infty$. Assume additionally that $S_{1,\varepsilon}$ is a continuity set for $p_\infty$, i.e. $p_\infty(\partial S_{1,\varepsilon}) = 0$ (for example, if $r_1$ is continuous and $\varepsilon$ is chosen so that the level set $\{r_1 = r_1^* - \varepsilon\}$ has $p_\infty$-measure zero). Then*

$$p_\infty(S_{1,\varepsilon}) \in [\eta, 1 - \eta],$$

*with the same $\eta$ as in Lemma B.6. Hence no such limit point collapses onto a single basin.*

*Proof.* Lemma B.6 gives $\eta \leq a_t \leq 1 - \eta$ for all $t \geq t_0$. Along any subsequence $t_k \to \infty$, we therefore have $\eta \leq \liminf_k a_{t_k} \leq \limsup_k a_{t_k} \leq 1 - \eta$. If $S_{1,\varepsilon}$ is a continuity set for $p_\infty$ and $p_{t_k} \Rightarrow p_\infty$, then $p_{t_k}(S_{1,\varepsilon}) \to p_\infty(S_{1,\varepsilon})$. Since $p_{t_k}(S_{1,\varepsilon}) = a_{t_k}$, it follows that $p_\infty(S_{1,\varepsilon}) \in [\eta, 1 - \eta]$. $\qquad\square$

**Theorem B.2** (Two-basin subsequential limits). *Fix $\varepsilon > 0$ and assume the separation conditions of Lemma B.5. Suppose $m_t \to 0$. Let $t_k \to \infty$ be a subsequence such that $p_{t_k} \Rightarrow p_\infty$. Assume $O_\varepsilon := \mathcal{X} \setminus \mathcal{S}_\varepsilon$ is open, and that $S_{1,\varepsilon}$ is a $p_\infty$-continuity set. Then $p_\infty$ is supported on $\mathcal{S}_\varepsilon$ and can be written as*

$$p_\infty = a_\infty p_{\infty,1} + (1 - a_\infty) p_{\infty,2}, \qquad a_\infty := p_\infty(S_{1,\varepsilon}),$$

*where $p_{\infty,i} := p_\infty(\cdot \mid S_{i,\varepsilon})$ whenever the corresponding mass is nonzero. If, in addition, $a_t \to a_\infty \in (0,1)$, then*

$$\max\left\{0, \frac{q - \kappa_1}{1 - \kappa_1}\right\} \leq a_\infty \leq \min\left\{1, \frac{q}{1 - \kappa_2}\right\}.$$

*In particular, as $\Delta_1, \Delta_2 \to \infty$, the limiting basin mass approaches $q$.*

*Proof.* Since $O_\varepsilon$ is open and $p_{t_k} \Rightarrow p_\infty$, the Portmanteau theorem gives

$$p_\infty(O_\varepsilon) \leq \liminf_{k \to \infty} p_{t_k}(O_\varepsilon) = \liminf_{k \to \infty} m_{t_k} = 0.$$

Thus $p_\infty$ is supported on $\mathcal{S}_\varepsilon$. Because $S_{1,\varepsilon}$ and $S_{2,\varepsilon}$ are disjoint and $p_\infty(\mathcal{S}_\varepsilon) = 1$, conditioning on the two pieces gives

$$p_\infty = p_\infty(S_{1,\varepsilon}) p_\infty(\cdot \mid S_{1,\varepsilon}) + p_\infty(S_{2,\varepsilon}) p_\infty(\cdot \mid S_{2,\varepsilon}),$$

which is the stated decomposition. The continuity-set assumption gives $p_{t_k}(S_{1,\varepsilon}) \to p_\infty(S_{1,\varepsilon})$. Hence, if $a_t \to a_\infty \in (0,1)$, the interval bound follows from Lemma B.5. The final claim follows by letting $\kappa_1, \kappa_2 \to 0$. $\qquad\square$

**Proposition B.3** (Plateau basins). *Fix $\varepsilon > 0$ and suppose $S_{1,\varepsilon} \cap S_{2,\varepsilon} = \emptyset$ with $p_0(S_{1,\varepsilon}) > 0$ and $p_0(S_{2,\varepsilon}) > 0$. Work in the $K \to \infty$ regime and assume $m_t := p_t(\mathcal{X} \setminus \mathcal{S}_\varepsilon) \to 0$. Assume that the rewards are constant on the two basins: for some $U_1, U_2 \in \mathbb{R}$ and $\Delta_1, \Delta_2 > 0$,*

$$(r_1, r_2) = (U_1, U_2 - \Delta_2) \quad \text{on } S_{1,\varepsilon}, \qquad (r_1, r_2) = (U_1 - \Delta_1, U_2) \quad \text{on } S_{2,\varepsilon}.$$

*Then the basin-conditionals are invariant:*

$$p_t(\cdot \mid S_{i,\varepsilon}) = p_0(\cdot \mid S_{i,\varepsilon}) =: p_{S_{i,\varepsilon}}, \qquad i \in \{1,2\}, \ t \geq 0.$$

*Consequently, along any subsequence $t_k$ with $a_{t_k} := p_{t_k}(S_{1,\varepsilon}) \to a_\infty$,*

$$p_{t_k} \Rightarrow a_\infty p_{S_{1,\varepsilon}} + (1 - a_\infty) p_{S_{2,\varepsilon}}.$$

*Moreover, $a_\infty$ satisfies the leakage interval of Lemma B.5; in the exact plateau parameterization above, the leakage factors are $\kappa_1 = e^{-\Delta_1}$ and $\kappa_2 = e^{-\Delta_2}$. Hence, as $\Delta_1, \Delta_2 \to \infty$, the limiting mixture weight approaches $q$.*

*Proof.* Write $S_i = S_{i,\varepsilon}$. In the large-$K$ regime, the update has the form $p_{t+1} = W_t p_t$, where

$$W_t(x) = q \frac{e^{r_1(x)}}{\mu_1(t)} + (1 - q) \frac{e^{r_2(x)}}{\mu_2(t)}, \qquad \mu_i(t) := \int_{\mathcal{X}} e^{r_i} \, dp_t.$$

By the plateau assumption, $W_t$ is constant on each basin. Thus, for each $i \in \{1,2\}$, there exists $M_i(t) > 0$ such that $p_{t+1} = M_i(t) p_t$ on $S_i$. It follows immediately that

$$p_{t+1}(\cdot \mid S_i) = p_t(\cdot \mid S_i).$$

Induction gives $p_t(\cdot \mid S_i) = p_0(\cdot \mid S_i)$ for all $t$.

Using this invariance,

$$p_t = a_t p_{S_1} + b_t p_{S_2} + m_t p_t(\cdot \mid \mathcal{X} \setminus \mathcal{S}_\varepsilon), \qquad b_t = 1 - a_t - m_t.$$

Since $m_t \to 0$, any subsequence with $a_{t_k} \to a_\infty$ satisfies

$$p_{t_k} \Rightarrow a_\infty p_{S_1} + (1 - a_\infty) p_{S_2}.$$

The leakage interval follows from Lemma B.5. Under the exact plateau parameterization, cross-basin reinforcement is reduced by factors $e^{-\Delta_1}$ and $e^{-\Delta_2}$, so the interval collapses to $a_\infty = q$ as $\Delta_1, \Delta_2 \to \infty$. $\qquad\square$

**Theorem B.4** (Expected rewards along two-basin limits). *Assume the setting of Theorem B.2, and let $p_{t_k} \Rightarrow p_\infty$ be a two-basin limit with*

$$p_\infty = a_\infty p_{\infty,1} + (1 - a_\infty) p_{\infty,2}, \qquad p_{\infty,i} := p_\infty(\cdot \mid S_{i,\varepsilon}).$$

*If $r_1$ and $r_2$ are bounded and continuous, then, for each $i \in \{1, 2\}$,*

$$\mathbb{E}_{p_{t_k}}[r_i] \to a_\infty \mathbb{E}_{p_{\infty,1}}[r_i] + (1 - a_\infty) \mathbb{E}_{p_{\infty,2}}[r_i].$$

*Moreover,*

$$\mathbb{E}_{p_{\infty,1}}[r_1] \in [r_1^* - \varepsilon, r_1^*], \qquad \mathbb{E}_{p_{\infty,2}}[r_2] \in [r_2^* - \varepsilon, r_2^*],$$

*and, under the separation conditions of Lemma B.5,*

$$\mathbb{E}_{p_{\infty,2}}[r_1] \leq r_1^* - \Delta_1 + \varepsilon, \qquad \mathbb{E}_{p_{\infty,1}}[r_2] \leq r_2^* - \Delta_2 + \varepsilon.$$

*Proof.* Bounded continuity of $r_i$ and weak convergence give $\mathbb{E}_{p_{t_k}}[r_i] \to \mathbb{E}_{p_\infty}[r_i]$. The mixture formula then follows by linearity of expectation applied to $p_\infty = a_\infty p_{\infty,1} + (1 - a_\infty) p_{\infty,2}$. The first pair of bounds follows because $p_{\infty,1}$ is supported on $S_{1,\varepsilon}$ and $p_{\infty,2}$ is supported on $S_{2,\varepsilon}$. The cross-basin bounds are the corresponding pointwise separation inequalities integrated over the opposite basin. $\square$

**Corollary B.5** (Expected rewards in the hard-separation plateau case). *Assume the plateau setting of Proposition B.3 and suppose the hard-separation limit gives*

$$p_t \Rightarrow q\, p_{S_{1,\varepsilon}} + (1 - q)\, p_{S_{2,\varepsilon}}.$$

*If $r_1$ and $r_2$ are bounded and continuous, then*

$$\mathbb{E}_{p_t}[r_1] \to q r_1^* + (1 - q) \mathbb{E}_{p_{S_{2,\varepsilon}}}[r_1], \qquad \mathbb{E}_{p_t}[r_2] \to q \mathbb{E}_{p_{S_{1,\varepsilon}}}[r_2] + (1 - q) r_2^*.$$

*Proof.* Bounded continuity and weak convergence imply convergence of expectations. Linearity under the limiting mixture gives

$$\mathbb{E}_{p_t}[r_i] \to q\, \mathbb{E}_{p_{S_{1,\varepsilon}}}[r_i] + (1 - q)\, \mathbb{E}_{p_{S_{2,\varepsilon}}}[r_i].$$

In the plateau case, $r_1 \equiv r_1^*$ on $S_{1,\varepsilon}$ and $r_2 \equiv r_2^*$ on $S_{2,\varepsilon}$, which yields the two displayed limits. $\square$

**Theorem B.6** (Variance preservation under two-basin limits). *Fix $\varepsilon > 0$ and suppose $S_{1,\varepsilon}$ and $S_{2,\varepsilon}$ are disjoint. Let $p_{t_k} \Rightarrow p_\infty$ be a two-basin limit with basin weight $a_\infty \in (0, 1)$, and assume $r_1, r_2$ are bounded and continuous. Then $Var_{p_{t_k}}[r_i] \to Var_{p_\infty}[r_i]$ for $i \in \{1, 2\}$. Moreover, if*

$$r_1 \leq r_1^* - \Delta_1 + \varepsilon \text{ on } S_{2,\varepsilon}, \qquad r_2 \leq r_2^* - \Delta_2 + \varepsilon \text{ on } S_{1,\varepsilon},$$

*then*

$$Var_{p_\infty}[r_i] \geq a_\infty(1 - a_\infty)(\Delta_i - 2\varepsilon)_+^2, \qquad i \in \{1, 2\}.$$

*In particular, if $\Delta_1, \Delta_2 > 2\varepsilon$, both limiting reward variances are positive.*

*Proof.* Because $r_i$ is bounded and continuous, so is $r_i^2$. Since $p_{t_k} \Rightarrow p_\infty$, the Portmanteau theorem gives $\mathbb{E}_{p_{t_k}}[r_i] \to \mathbb{E}_{p_\infty}[r_i]$ and $\mathbb{E}_{p_{t_k}}[r_i^2] \to \mathbb{E}_{p_\infty}[r_i^2]$. Therefore

$$Var_{p_{t_k}}[r_i] = \mathbb{E}_{p_{t_k}}[r_i^2] - \mathbb{E}_{p_{t_k}}[r_i]^2 \longrightarrow \mathbb{E}_{p_\infty}[r_i^2] - \mathbb{E}_{p_\infty}[r_i]^2 = Var_{p_\infty}[r_i].$$

Write $p_\infty = a_\infty p_{\infty,1} + (1 - a_\infty) p_{\infty,2}$ and set $\mu_{i,j} := \mathbb{E}_{p_{\infty,j}}[r_i]$ and $\mu := \mathbb{E}_{p_\infty}[r_i] = a_\infty \mu_{i,1} + (1 - a_\infty)\mu_{i,2}$. Then

$$\begin{aligned}
Var_{p_\infty}[r_i] &= \mathbb{E}_{p_\infty}\big[(r_i - \mu)^2\big] \\
&= a_\infty \mathbb{E}_{p_{\infty,1}}\big[(r_i - \mu)^2\big] + (1 - a_\infty)\mathbb{E}_{p_{\infty,2}}\big[(r_i - \mu)^2\big] \\
&= a_\infty Var_{p_{\infty,1}}[r_i] + (1 - a_\infty)Var_{p_{\infty,2}}[r_i] + a_\infty(1 - a_\infty)(\mu_{i,1} - \mu_{i,2})^2.
\end{aligned}$$

For the lower bound, first take $i = 1$. Since $p_{\infty,1}$ is supported on $S_{1,\varepsilon}$, we have $\mu_{1,1} \geq r_1^* - \varepsilon$. The separation condition gives $\mu_{1,2} \leq r_1^* - \Delta_1 + \varepsilon$. Hence $\mu_{1,1} - \mu_{1,2} \geq \Delta_1 - 2\varepsilon$, and therefore $|\mu_{1,1} - \mu_{1,2}| \geq (\Delta_1 - 2\varepsilon)_+$. Dropping the nonnegative within-basin variance terms in the decomposition above yields

$$\mathrm{Var}_{p_\infty}[r_1] \geq a_\infty(1 - a_\infty)(\Delta_1 - 2\varepsilon)_+^2.$$

The same argument with the roles of the two basins reversed gives

$$\mathrm{Var}_{p_\infty}[r_2] \geq a_\infty(1 - a_\infty)(\Delta_2 - 2\varepsilon)_+^2.$$

$\qquad\square$

**Theorem B.7** (Nash bargaining characterization on the two-basin mixture family)**.** *Let $P_1, P_2$ be two fixed distributions supported on $S_{1,\varepsilon}$ and $S_{2,\varepsilon}$ respectively, and define the mixture family*

$$p_\alpha := \alpha P_1 + (1 - \alpha)P_2, \qquad \alpha \in [0, 1].$$

*Let utilities be $u_i(p) := \mathbb{E}_p[r_i(x)]$, and define the disagreement points*

$$d_1 := u_1(P_2), \qquad d_2 := u_2(P_1).$$

*Assume the gains are strictly positive:*

$$u_1(P_1) > u_1(P_2), \qquad u_2(P_2) > u_2(P_1).$$

*Then the weighted Nash product*

$$\mathcal{N}(\alpha) := \big(u_1(p_\alpha) - d_1\big)^q \big(u_2(p_\alpha) - d_2\big)^{1-q}$$

*is uniquely maximized over $\alpha \in [0, 1]$ at $\alpha^\star = q$. Equivalently, $p_q$ is the unique weighted Nash bargaining solution within the line segment $\{p_\alpha : \alpha \in [0, 1]\}$.*

*Proof.* By linearity of expectation,

$$u_1(p_\alpha) = \alpha u_1(P_1) + (1 - \alpha)u_1(P_2) = d_1 + \alpha\Delta_1, \qquad \Delta_1 := u_1(P_1) - u_1(P_2) > 0,$$

and similarly

$$u_2(p_\alpha) = \alpha u_2(P_1) + (1 - \alpha)u_2(P_2) = d_2 + (1 - \alpha)\Delta_2, \qquad \Delta_2 := u_2(P_2) - u_2(P_1) > 0.$$

Hence

$$\mathcal{N}(\alpha) = (\alpha\Delta_1)^q\big((1 - \alpha)\Delta_2\big)^{1-q} = \Delta_1^q\Delta_2^{1-q}\,\alpha^q(1 - \alpha)^{1-q}.$$

The constant prefactor does not affect the maximizer, so it suffices to maximize $f(\alpha) := \alpha^q(1 - \alpha)^{1-q}$ on $[0, 1]$. For $\alpha \in (0, 1)$,

$$\frac{d}{d\alpha}\log f(\alpha) = \frac{q}{\alpha} - \frac{1 - q}{1 - \alpha}.$$

Setting this derivative to zero yields $\alpha = q$, and strict concavity of $\log f$ on $(0, 1)$ gives uniqueness. $\qquad\square$

**Corollary B.8** (Nash bargaining interpretation of the hard-separation plateau limit)**.** *Under Proposition B.3, let $P_1 := p_{S_{1,\varepsilon}}$ and $P_2 := p_{S_{2,\varepsilon}}$. If $u_1(P_1) > u_1(P_2)$ and $u_2(P_2) > u_2(P_1)$, then the limiting distribution*

$$p_\infty = q\,p_{S_{1,\varepsilon}} + (1 - q)\,p_{S_{2,\varepsilon}}$$

*is exactly the unique weighted Nash bargaining solution over mixtures $\{p_\alpha\}$ with disagreement points $d_1 = u_1(P_2)$ and $d_2 = u_2(P_1)$.*

*Proof.* This is Theorem B.7 applied to $P_1 = p_{S_{1,\varepsilon}}$ and $P_2 = p_{S_{2,\varepsilon}}$, together with Proposition B.3, which identifies $p_\infty$ with $p_q$. $\qquad\square$

## B.4. Extension to multiple reward functions

The two-reward analysis extends directly to any finite family of rewards $\{r_1, \ldots, r_M\}$ sampled with probabilities $q_i > 0$ and $\sum_{i=1}^{M} q_i = 1$. For each reward, define $S_{i,\varepsilon} := \{x \in \mathcal{X} : r_i(x) \geq r_i^* - \varepsilon\}$ and $\mathcal{S}_\varepsilon := \bigcup_{i=1}^{M} S_{i,\varepsilon}$. Let $m_t := p_t(\mathcal{X} \setminus \mathcal{S}_\varepsilon)$ and $a_{i,t} := p_t(S_{i,\varepsilon})$.

At each retraining step, the reward $R$ is drawn from $\{r_1, \ldots, r_M\}$ with $\Pr(R = r_i) = q_i$. The finite-$K$ update is

$$p_{t+1}(x) = p_t(x) \sum_{i=1}^{M} q_i H_{p_t}^{K,r_i}(x),$$

where $H_{p_t}^{K,r_i}$ is the same BT curation weight as in the two-reward case. By Lemma B.2, in the large-$K$ regime this becomes

$$p_{t+1}(x) = p_t(x) \sum_{i=1}^{M} q_i \frac{e^{r_i(x)}}{\mu_i(t)}, \qquad \mu_i(t) := \mathbb{E}_{p_t}[e^{r_i(x)}].$$

Thus the only change from the two-reward case is that the update multiplier is now the convex combination of $M$ exponential tilts rather than two.

The outside-mass argument also extends verbatim. Namely, if there exists $\rho_\varepsilon \in (0,1)$ such that

$$\sup_{x \notin \mathcal{S}_\varepsilon} \sum_{i=1}^{M} q_i \frac{e^{r_i(x)}}{\mu_i(t)} \leq \rho_\varepsilon \qquad \text{for all } t \geq 0, \tag{OD-M}$$

then $m_{t+1} \leq \rho_\varepsilon m_t$, and hence $m_t \to 0$. As in the two-reward case, this is best interpreted as a basin-completeness condition: all regions reinforced by the pluralistic update should be included in $\mathcal{S}_\varepsilon$.

For the allocation of mass inside $\mathcal{S}_\varepsilon$, assume for simplicity that the basins are disjoint. If they overlap, one may replace them by a measurable partition $\{\widetilde{S}_{i,\varepsilon}\}_{i=1}^{M}$ with $\widetilde{S}_{i,\varepsilon} \subseteq S_{i,\varepsilon}$ and $\bigcup_i \widetilde{S}_{i,\varepsilon} = \mathcal{S}_\varepsilon$. Suppose that for each ordered pair $i \neq j$ there is a separation gap $\Delta_{i \leftarrow j} > 2\varepsilon$ such that

$$x \in S_{j,\varepsilon} \quad \Rightarrow \quad r_i(x) \leq r_i^* - \Delta_{i \leftarrow j} + \varepsilon,$$

and define

$$\kappa_{i \leftarrow j} := \exp\big( -(\Delta_{i \leftarrow j} - 2\varepsilon) \big), \qquad \kappa_{\max} := \max_{i \neq j} \kappa_{i \leftarrow j}.$$

Then the same tilt comparison used in Lemma B.5 shows that selection under reward $r_i$ assigns almost all of its mass to its own basin $S_{i,\varepsilon}$, with leakage into other basins controlled by $\kappa_{\max}$.

Consequently, once $m_t \to 0$ and the limiting basin masses remain nontrivial, the one-step basin update satisfies

$$a_{i,t+1} = q_i + O(\kappa_{\max}) + o(1), \qquad i = 1, \ldots, M.$$

Equivalently, any subsequential limit $p_\infty$ supported on $\mathcal{S}_\varepsilon$ decomposes as

$$p_\infty = \sum_{i=1}^{M} a_{i,\infty} p_{\infty,i}, \qquad p_{\infty,i} := p_\infty(\cdot \mid S_{i,\varepsilon}),$$

with

$$|a_{i,\infty} - q_i| \leq C \kappa_{\max}$$

for a constant $C$ depending on the minimum limiting basin mass. In particular, in the vanishing-leakage regime $\kappa_{\max} \to 0$, the limiting weights converge to the reward-sampling probabilities:

$$a_{i,\infty} \to q_i, \qquad i = 1, \ldots, M.$$

Thus the multi-reward case has the same structure as the two-reward case: outside domination drives mass into the union of relevant basins, leakage controls deviations from the target weights, and in the hard-separation limit the pluralistic retraining loop converges to the mixture with weights $(q_1, \ldots, q_M)$.

# C. Additional Experimental Details and Ablations

## C.1. Appendix map

The main paper reports four core experiments: **(E1)** synthetic GMM dynamics under pluralistic vs. single-reward curation, **(E2)** synthetic phase transition as reward optima approach, **(E3)** CIFAR-10 flow retraining under class-conditional rewards, and **(E4)** GPT-2 retraining under conflicting length preferences.

This appendix contains the remaining experiments and reproducibility details: **(E5)** extended polarization ($q$) sweeps and limiting-mixture diagnostics in the synthetic setting (Figs. 6, 7, 10); **(E6)** finite-pool effects via sweeps over the discrete-choice pool size $K$ (Fig. 11); **(E7)** additional reward-separation visualizations and robustness checks (Fig. 15); **(E8)** leakage corroborations and implementation details (Sec. C.7); **(E9)** baseline comparisons; **(E10)** CIFAR-10 retraining with learned neural reward models, including PickScore, ImageReward, and LAION Aesthetic Score; and **(E11)** RLHF-style iterative retraining on Qwen-2.5-1.5B with helpfulness and safety rewards.

## C.2. Hardware and environment

All experiments were conducted on a dedicated server with:

- **GPU:** 4 `NVIDIA H200` cards,

- **CPU:** `Intel(R) Core(TM) i9-7920X CPU @ 2.90GHz` (12 physical cores, 24 threads),

- **RAM:** 128GB DDR4.

We fix random seeds when supported by the underlying libraries and report them in our code release. Unless stated otherwise, all figures report per-iteration statistics computed from samples drawn from the current model at that iteration.

## C.3. Common retraining and selection protocol

Across domains, retraining proceeds in rounds. At round $t$, the current model $p_t$ produces a candidate pool $\{x_1, \ldots, x_K\} \sim p_t$. An active reward is drawn from a preference mixture policy (for two rewards: choose $r_1$ with probability $q$ and $r_2$ with probability $1 - q$). We then apply a discrete-choice rule (BT-style sampling) to select one item from the pool. With temperature $\tau > 0$,

$$\Pr\{x = x_i \mid r, \{x_j\}_{j=1}^K\} \;=\; \frac{\exp(r(x_i)/\tau)}{\sum_{j=1}^K \exp(r(x_j)/\tau)}. \tag{C.1}$$

Repeating this selection step $n_{\text{curated}}$ times produces a curated dataset $\mathcal{C}_t$ (sampling with replacement unless stated otherwise), and the model is retrained on $\mathcal{C}_t$ to obtain $p_{t+1}$.

**Logged quantities.** When rewards are available for analysis, we log at each iteration: (i) expected rewards $\mathbb{E}_{x \sim p_t}[r_i(x)]$ and variances $\text{Var}_{x \sim p_t}(r_i(x))$, (ii) mixture allocations or proxy assignments when available (synthetic GMM), (iii) entropy-like diversity proxies (CIFAR-10 and text).

## C.4. Synthetic Gaussian mixture details

We instantiate pluralistic curation in a 2D Gaussian mixture environment to allow direct visualization and controlled ablations.

**Initialization.** The generator is initialized as a one-component Gaussian mixture centered at $[5, 5]$ with covariance $\Sigma = 3I$, fit to 1000 points sampled from $\mathcal{N}([5, 5], 3I)$. The reward modes are placed at $\mu_1 = [2, 2]$ and $\mu_2 = [8, 8]$ unless stated otherwise.

**Rewards.** We use quadratic rewards centered at the reward modes:

$$r_1(x) = -\|x - \mu_1\|^2, \qquad r_2(x) = -\|x - \mu_2\|^2. \tag{C.2}$$

To study reward separation, we vary the distance $d = \|\mu_1 - \mu_2\|$ while keeping the midpoint fixed.

**Curation.** At each iteration, we sample $K = 100$ candidates from the current model and curate $n_{\text{curated}} = 500$ samples using (C.1). For each curated draw, the active reward is chosen independently: $r = r_1$ with probability $q$ and $r = r_2$ with probability $1 - q$ (default $q = 0.5$).

**Retraining regime.** We run $T = 50$ iterations. For $t < 10$, we refit a single-component GMM to simulate an early low-capacity regime. For $t \geq 10$, we refit a two-component GMM to permit multimodality. We fit GMMs via EM using scikit-learn with `random_state=42` and default convergence settings.

**Evaluation.** At each iteration, we sample 1000 points from the updated GMM and compute $\mathbb{E}[r_i]$ and $\text{Var}(r_i)$ for $i \in \{1, 2\}$. In the two-component regime, we also record component means and mixture weights.

**Single-reward baseline.** We include a baseline that uses only $r_1$ (equivalently, $q = 1$) with all other settings fixed.

### C.4.1. SYNTHETIC PSEUDOCODE

---

**Algorithm 1** Pluralistic Curation with Gaussian Mixture Retraining

---

**Require:** Reward centers $\mu_1, \mu_2$, iterations $T$, pool size $K$, curated draws $n_{\text{curated}}$, polarization $q$, temperature $\tau$
  1: Define rewards $r_1(x) = -\|x - \mu_1\|^2$, $r_2(x) = -\|x - \mu_2\|^2$
  2: Initialize generator $G_0$ as a 1-component GMM centered at $(\mu_1 + \mu_2)/2$ with large covariance
  3: **for** $t = 1$ to $T$ **do**
  4:     Sample candidates $\{x_1, \ldots, x_K\} \sim G_{t-1}$
  5:     Initialize curated set $\mathcal{C}_t \leftarrow \emptyset$
  6:     **for** $j = 1$ to $n_{\text{curated}}$ **do**
  7:         Choose reward $r \leftarrow r_1$ w.p. $q$, else $r \leftarrow r_2$
  8:         Compute scores $s_i = r(x_i)$ for $i = 1, \ldots, K$
  9:         Normalize $\tilde{s}_i \leftarrow s_i - \max_\ell s_\ell$
 10:         Set $p_i \propto \exp(\tilde{s}_i/\tau)$
 11:         Sample $x^{(j)}$ from $\{x_i\}$ using $\{p_i\}$ and add to $\mathcal{C}_t$
 12:     **end for**
 13:     Set #components $k \leftarrow 1$ if $t < 10$, else $k \leftarrow 2$
 14:     Fit a $k$-component GMM $G_t$ to $\mathcal{C}_t$ via EM
 15: **end for**
 16: **return** $G_T$

---

**Algorithm 2** Single-Reward Curation Baseline (GMM)

---

**Require:** Reward center $\mu$, iterations $T$, pool size $K$, curated draws $n_{\text{curated}}$, temperature $\tau$
  1: Define reward $r(x) = -\|x - \mu\|^2$
  2: Initialize generator $G_0$ as a 1-component GMM with large covariance
  3: **for** $t = 1$ to $T$ **do**
  4:     Sample candidates $\{x_1, \ldots, x_K\} \sim G_{t-1}$
  5:     Initialize curated set $\mathcal{C}_t \leftarrow \emptyset$
  6:     **for** $j = 1$ to $n_{\text{curated}}$ **do**
  7:         Compute scores $s_i = r(x_i)$ for $i = 1, \ldots, K$
  8:         Normalize $\tilde{s}_i \leftarrow s_i - \max_\ell s_\ell$
  9:         Set $p_i \propto \exp(\tilde{s}_i/\tau)$
 10:         Sample $x^{(j)}$ from $\{x_i\}$ using $\{p_i\}$ and add to $\mathcal{C}_t$
 11:     **end for**
 12:     Set #components $k \leftarrow 1$ if $t < 10$, else $k \leftarrow 2$
 13:     Fit a $k$-component GMM $G_t$ to $\mathcal{C}_t$ via EM
 14: **end for**
 15: **return** $G_T$

---

## C.5. CIFAR-10 experimental details

We evaluate pluralistic curation on CIFAR-10 (Krizhevsky, 2009) using the same overall setup as (Ferbach et al., 2024). We train a normalizing flow using optimal transport conditional flow matching (OT-CFM) (Lipman et al., 2022; Shaul et al., 2023; Tong et al., 2023). The initial model is pretrained on all 50,000 training images.

**Per-round sampling and curation.** At each retraining iteration, we generate $5 \cdot 10^4$ samples from the current model and retain $2.5 \cdot 10^3$ samples via discrete-choice selection governed by the BT rule. Due to the computational cost of flow retraining, we run $T = 25$ iterations.

**Rewards.** Rewards are computed from class probabilities output by a pretrained VGG11 classifier (Simonyan & Zisserman, 2015) with 92.39% CIFAR-10 test accuracy. Let $q_i(x)$ denote the predicted probability of class $i$ for image $x$. For a fixed target class $i$, we define $r(x) = \gamma \, q_i(x)$ for a constant scaling $\gamma$.

**Retraining regimes. Balanced multi-preference:** choose $M \in \{1, \ldots, 5\}$ target classes and select the active reward uniformly each curated draw. **Polarized two-preference:** fix two target classes and select the active reward with probability $q$ versus $1 - q$.

**Metrics.** We report FID (Heusel et al., 2017) and diversity proxies computed from classifier predictions and features: (i) class entropy (higher implies broader class coverage), (ii) KL divergence to uniform (lower implies more balanced coverage), (iii) feature variance (higher implies greater inter-sample diversity in feature space), (iv) intra-class variance (higher implies finer-grained diversity within predicted classes).

## C.6. Text-domain experimental details (GPT-2 length preferences)

This section provides full details for the GPT-2 experiment, which tests pluralistic curation under conflicting and verifiable objectives.

**Data and task.** We use WikiText-2 (Merity et al., 2016) as the initial seed pool for open-ended continuation. We initialize with 1,000 sequences.

**Model.** We use a lightweight GPT-2 style decoder with 6 transformer layers, 6 attention heads, and embedding dimension 384 (vocabulary size 50,257) to enable multi-round fine-tuning.

**Preferences and reward.** We define two preferences over output length targets $T_A$ and $T_B$ (in words). For a generated sequence $y$, let $L(y)$ be its word count. The reward is

$$R(y; T) = -|L(y) - T|. \tag{C.3}$$

We fix $T_A$ and vary $T_B$ to control the conflict distance $d = |T_A - T_B|$.

**Retraining loop.** We run $N = 20$ rounds. Each round: (i) compute rewards for the current pool, (ii) curate 200 samples using a balanced mixture policy ($q = 0.5$) with BT sampling proportional to $\exp(R/\tau)$ and $\tau = 0.5$, (iii) fine-tune using AdamW with learning rate $5 \times 10^{-5}$ for 2 epochs per round (batch size 8), (iv) generate 200 new samples via nucleus sampling (temperature $t = 0.8$), filter with the same selection rule, and add survivors to the pool.

**Metric.** We compute the discrete entropy $H(L)$ of the generated length distribution at each round. Higher entropy indicates a broader policy over lengths and serves as a non-collapse proxy under competing length objectives.

## C.7. Leakage corroborations and proxies

Our theory is phrased in terms of cross-basin leakage, meaning the probability that selection under one preference draws samples that belong to another preference basin. In experiments, we estimate leakage using domain-specific proxies.

**Synthetic (GMM).** We define basin assignments by nearest reward center: $x$ is assigned to basin $i$ if $\|x - \mu_i\| \leq \|x - \mu_{3-i}\|$. When $r_1$ is active, we estimate leakage as the fraction of selected samples assigned to basin 2, and symmetrically for $r_2$. Leakage increases as reward optima approach, consistent with the phase transition in Fig. 3 in the main paper.

**CIFAR-10.** We use the classifier's predicted class as a coarse basin label. When the active reward targets class $i$, we estimate leakage as the fraction of curated samples whose predicted class is not $i$. This proxy is conservative because it mixes semantic leakage with classifier uncertainty, but it tracks the same qualitative trend: balanced pluralism sustains coverage, while extreme polarization reduces it.

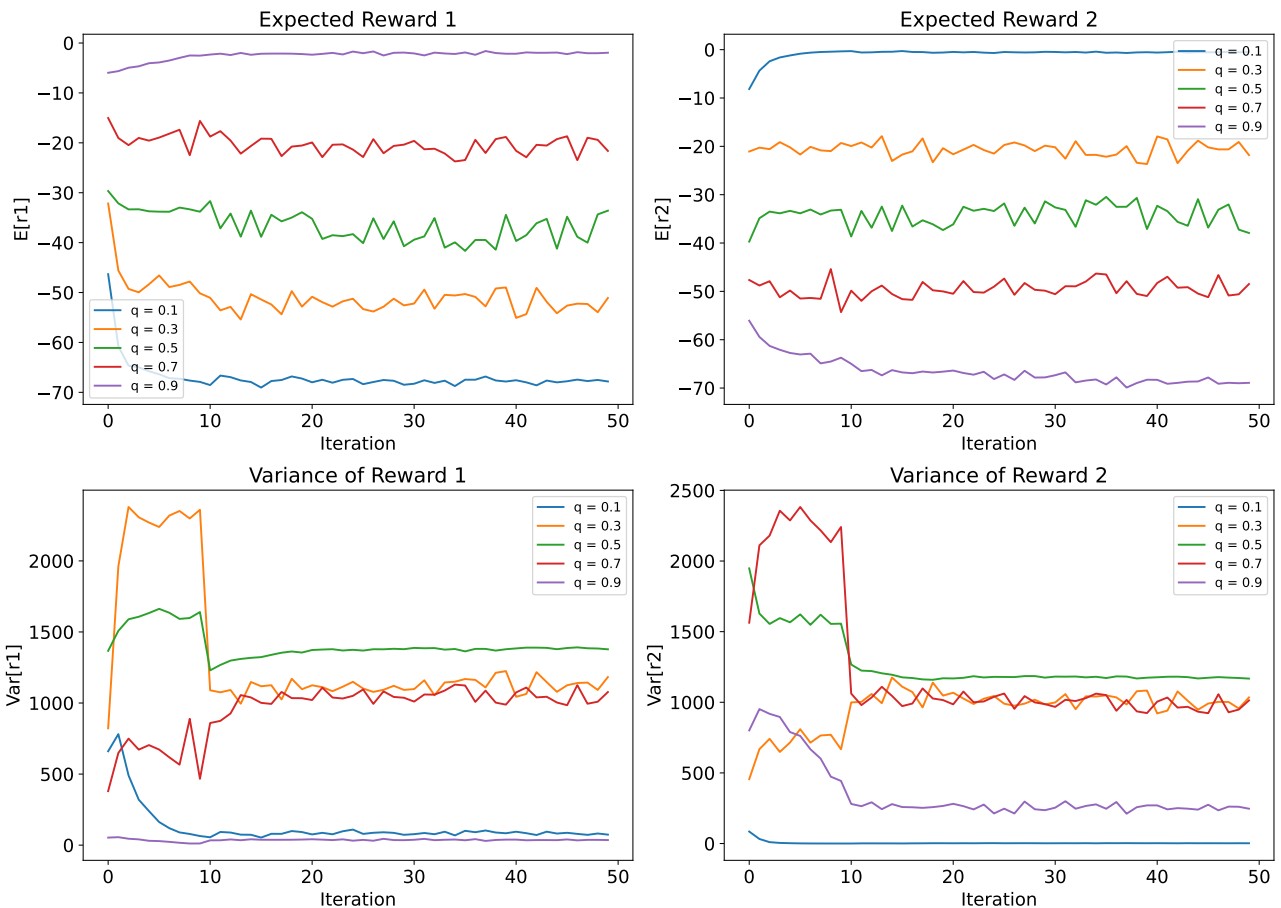

*Figure 6.* **Effect of Curation Probability** $q$ **in the Gaussian Synthetic Setting.** $q$ in the 2D Gaussian Setup. Pluralistic curation is applied with varying $q \in \{0.1, 0.3, 0.5, 0.7, 0.9\}$, controlling the preference toward $\texttt{reward}_1$. Top two rows show the expected rewards for $\texttt{reward}_1$ and $\texttt{reward}_2$; bottom rows show the corresponding variances across iterations on synthetic data.

**Text (GPT-2).** We define length bins around each target (for example, $|L(y) - T_A| \leq \delta$ and $|L(y) - T_B| \leq \delta$) as basin proxies. Leakage is the fraction of selected samples that fall into the non-active bin under the active reward. As $d$ increases, both bins remain populated under balanced pluralistic selection, consistent with the entropy trends in Fig. 5 in the main paper.

### C.8. Additional Experiments

**Learned reward models on CIFAR-10** We also test pluralistic curation with learned neural reward models rather than classifier-based rewards. Specifically, we use PickScore (Kirstain et al., 2023), ImageReward (Xu et al., 2023b), and the LAION Aesthetic Score predictor (LAION-AI, 2022) as three black-box preferences. All experiments use the same OT-CFM retraining setup as the main CIFAR-10 experiment: at each round, the current model generates $5 \times 10^4$ samples and $2.5 \times 10^3$ are retained by BT curation.

We compare three types of curation policies: single-reward curation using only one reward, balanced pairwise mixtures using two rewards with equal probability, and a uniform 3-way mixture. Fig. 8 shows that single-reward retraining degrades most, pairwise mixtures are more stable, and the 3-way mixture gives the best overall quality and diversity. In particular, single-reward FID degrades to roughly 60–72, pairwise mixtures stabilize around 52–58, and the 3-way mixture stays near 44.

The reward-variance curves also match the leakage interpretation. A reward's variance is preserved mainly when that reward participates in the curation mixture. For example, the PickScore+ImageReward mixture, which excludes the Aesthetic

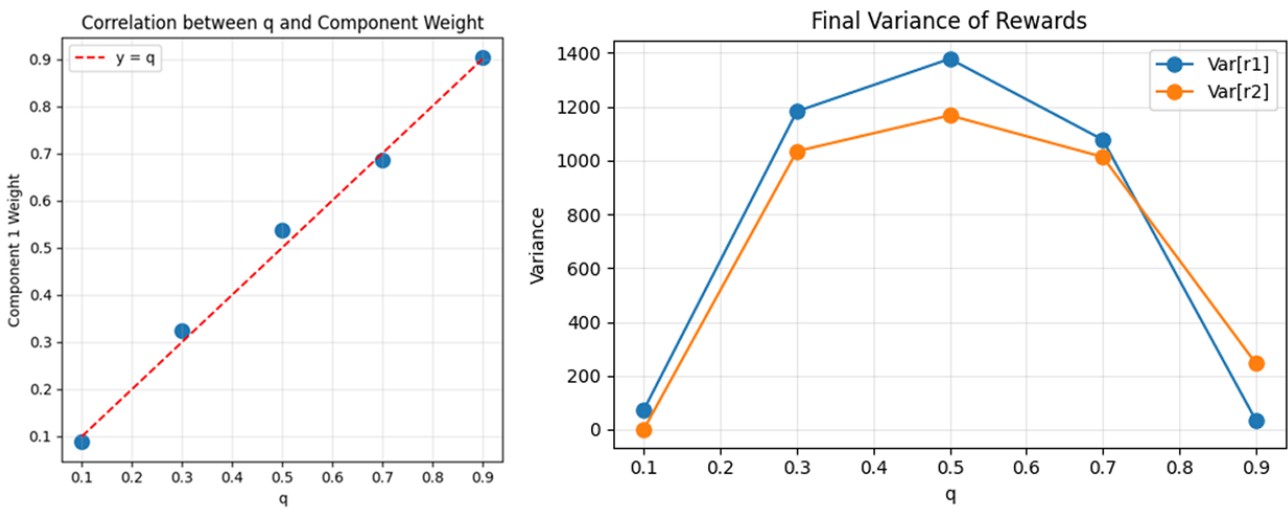

*Figure 7.* **Effect of Polarization Parameter** $q$ **on Mixture Weight and Diversity.** *(left)* The component weight tracks $q$, confirming convergence to a $(q, 1 - q)$ mixture. *(right)* Reward variance peaks near $q = 0.5$, indicating maximal diversity, and declines at extremes due to partial collapse.

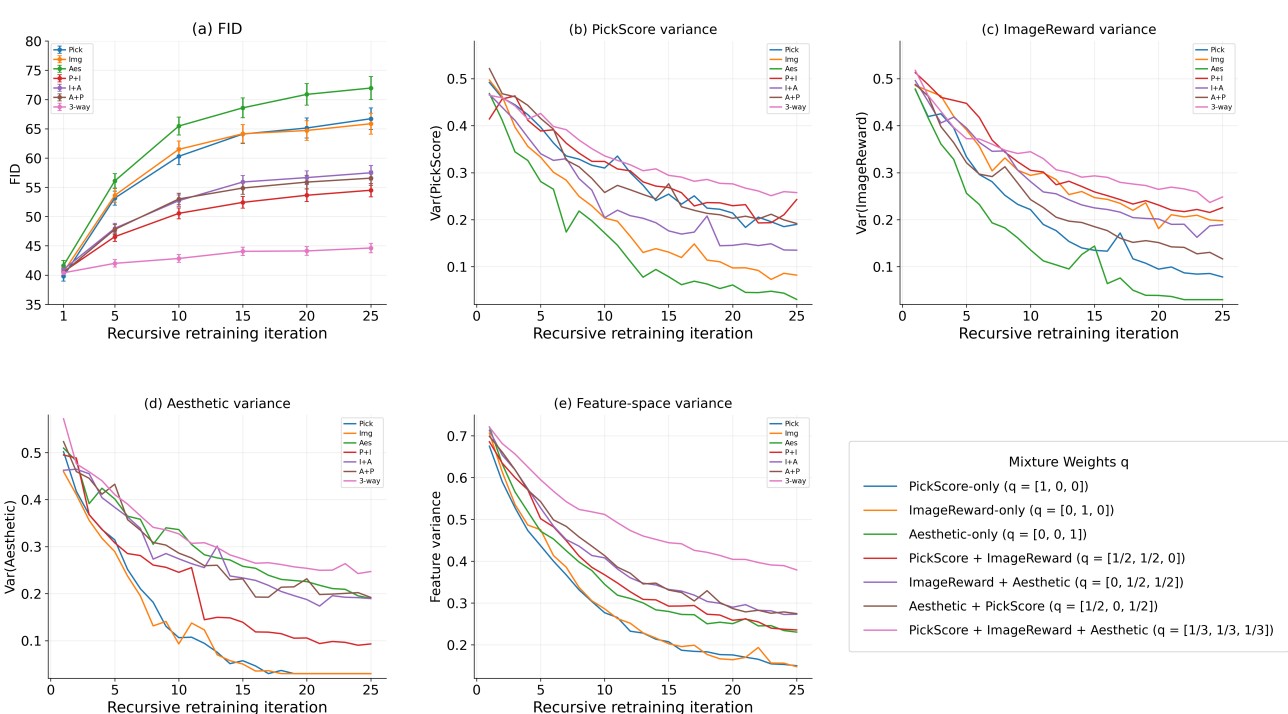

*Figure 8.* **CIFAR-10 retraining with learned reward models.** We replace classifier rewards with PickScore, ImageReward, and LAION Aesthetic Score, and compare single-reward curation, balanced pairwise mixtures, and a uniform 3-way mixture over 25 retraining rounds using the same OT-CFM flow. Single-reward curation degrades FID and collapses feature diversity, pairwise mixtures partially mitigate collapse, and the 3-way mixture performs best. Reward variance is preserved primarily when that reward participates in the curation mixture, matching the leakage mechanism predicted by Lemma B.5.

reward, collapses Aesthetic variance to roughly the single-reward level, while mixtures that include Aesthetic preserve substantially higher Aesthetic variance. This supports the claim that pluralistic curation preserves diversity by directly reinforcing multiple reward basins.

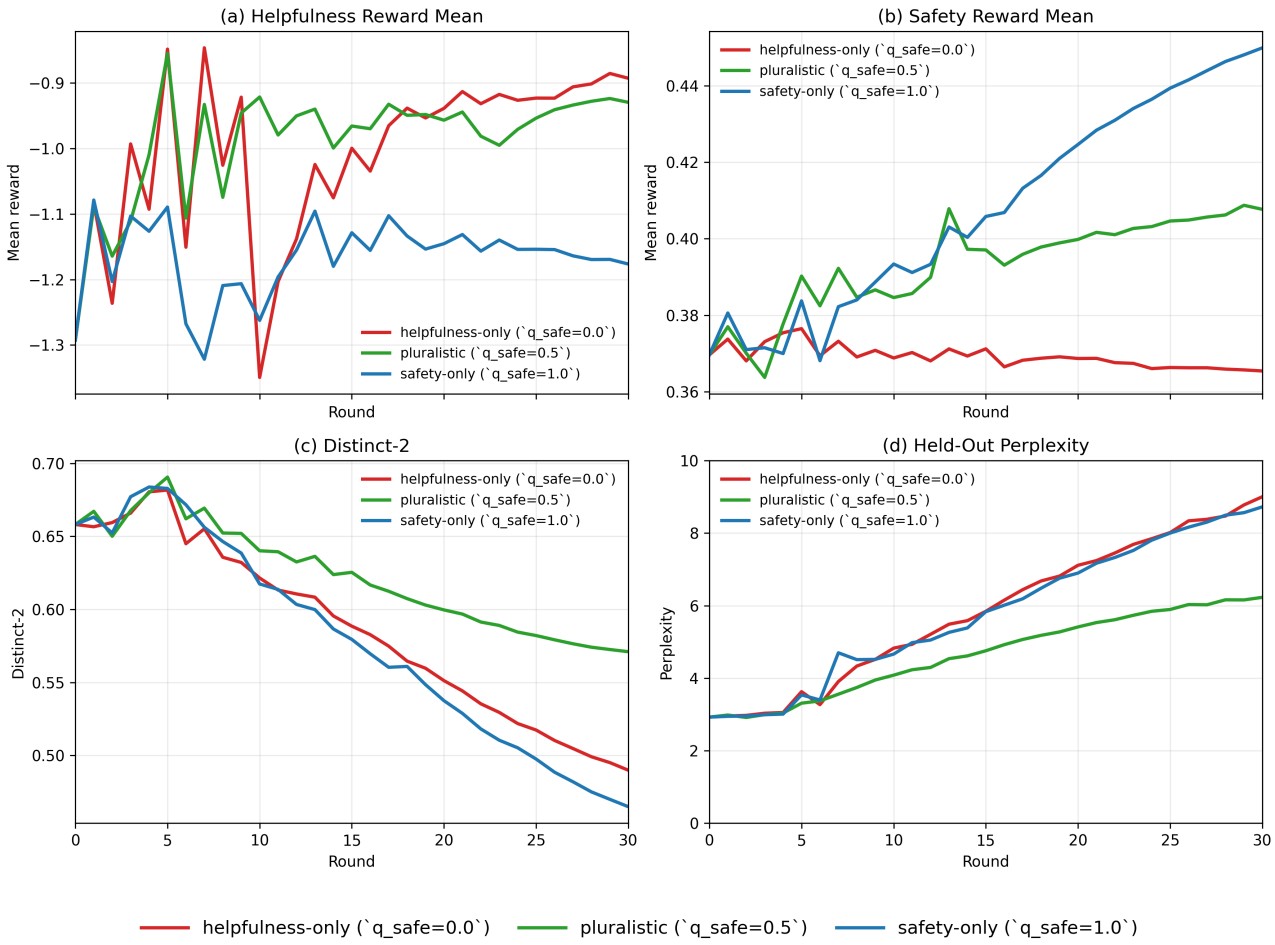

*Figure 9.* **Iterative Qwen-2.5 retraining with RLHF-relevant rewards.** We fine-tune Qwen-2.5-1.5B for 30 recursive rounds using BT-curated synthetic outputs under helpfulness and safety rewards. Helpfulness-only and safety-only curation over-optimize one objective while degrading the other. Balanced pluralistic curation maintains both objectives, preserves lexical diversity, and yields lower held-out perplexity, indicating less degeneration under recursive retraining.

**RLHF-style iterative retraining** To test the theory in a setting closer to RLHF practice, we run iterative retraining with Qwen-2.5-1.5B (Qwen Team, 2024) under two competing alignment rewards. At each round, the model generates $N = 500$ continuations on prompts sampled from UltraFeedback (Cui et al., 2023). The continuations are curated with BT selection and the model is then fine-tuned by SFT on the selected outputs. We repeat this loop for 30 rounds. We use ArmoRM-Llama3-8B-v0.1 (Wang et al., 2024) as the helpfulness reward and Llama-Guard-3-1B (Meta AI, 2024) as the safety evaluator, with $r_{\text{safe}}(y) = 1 - P_{\text{unsafe}}(y)$. We compare helpfulness-only curation, safety-only curation, and balanced pluralistic curation with $q_{\text{safe}} = 0.5$. Fig. 9 shows the same qualitative behavior predicted by the theory. Single-reward curation improves its target objective but degrades the other: helpfulness-only curation reduces safety, while safety-only curation reduces helpfulness. Balanced pluralistic curation maintains both objectives at competitive levels. It also preserves diversity better, with Distinct-2 near $0.60$ compared to $0.47$–$0.49$ for single-reward curation, and it limits held-out perplexity growth to roughly 6 instead of 8–9. Thus, even outside the exact-MLE abstraction, pluralistic curation reduces recursive degeneration relative to single-objective retraining.

**Impact of Polarization Parameter $q$.** Fig. 6 shows how changing the polarization parameter $q$, which dictates the probability of selecting $r_1$, affects the pluralistic curation dynamics. Results for $q \in \{0.1, 0.3, 0.5, 0.7, 0.9\}$ demonstrate that reward convergence and variance behavior depend on this parameter. With low $q$ values (0.1 and 0.3), the model favors the second Gaussian mode, reflected in lower expected rewards and variances for $r_1$ and higher corresponding values for $r_2$. Conversely, higher $q$ values (0.7 and 0.9) show an opposite behavior by selecting samples around the first mode. The

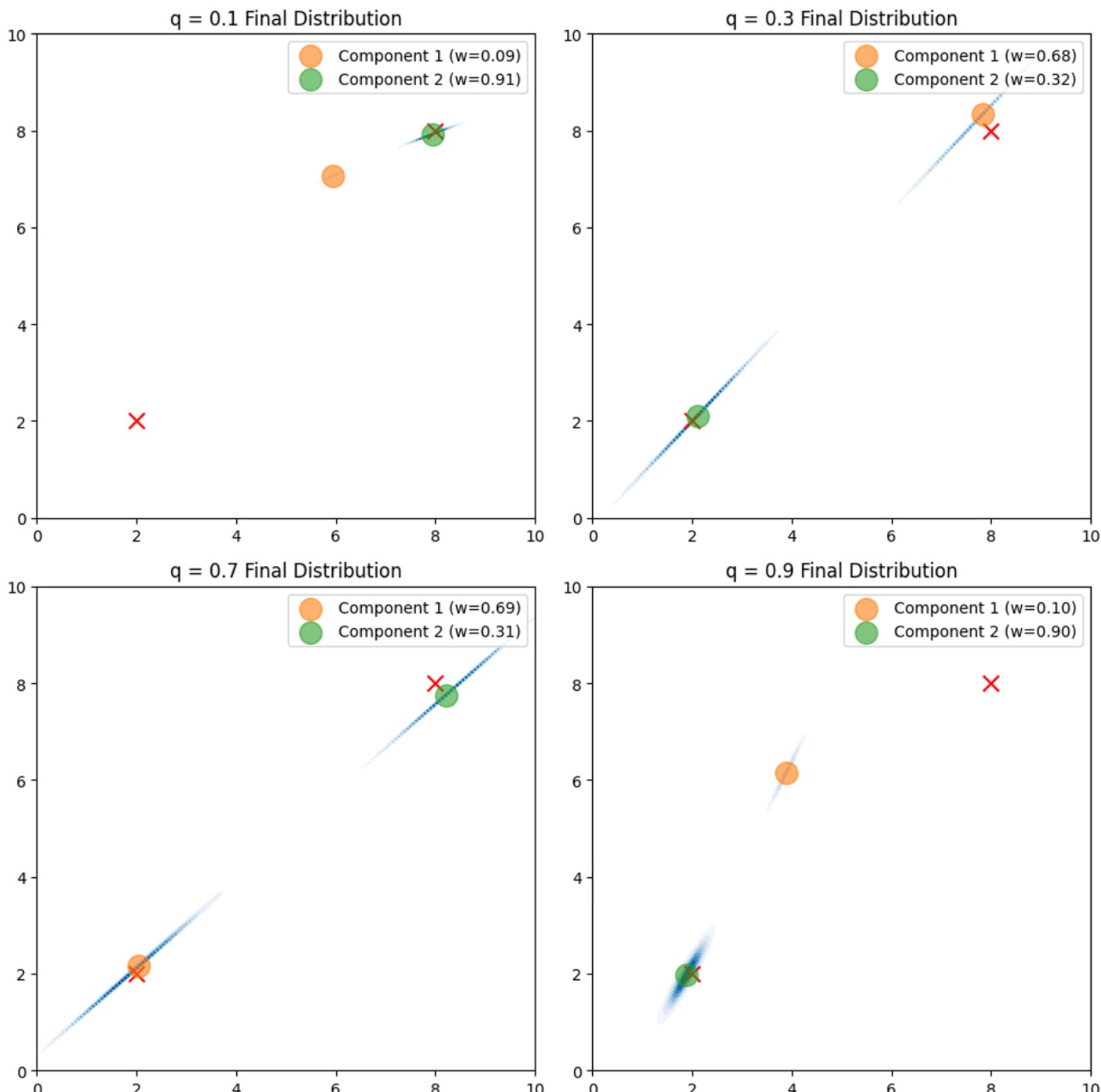

*Figure 10.* **Limiting distributions under varying parameter** $q$. Final model distributions for $q \in \{0.1, 0.3, 0.7, 0.9\}$ show that the learned mixture allocates mass across components in proportion to $q$.

balanced setting $q = 0.5$ results in a stable equilibrium distribution. Thus, pluralistic curation mitigates mode collapse, maintains diversity, and converges toward stable mixed distributions.

**Limiting distributions and mixture weights under varying** $q$. Figures 7 and 10 visualize the limiting distributions for different values of $q$. The mass allocated to each component closely tracks the polarization parameter, confirming the theoretical prediction that the model converges to a mixture of the two high-reward regions with weights approximately $(q, 1 - q)$. Importantly, each component remains distinct and sharp, indicating that pluralistic curation avoids collapse while preserving structure within each basin. This is aligned with the weighted bargaining interpretation in which the final distribution acts as a $q$-weighted compromise between competing preferences.

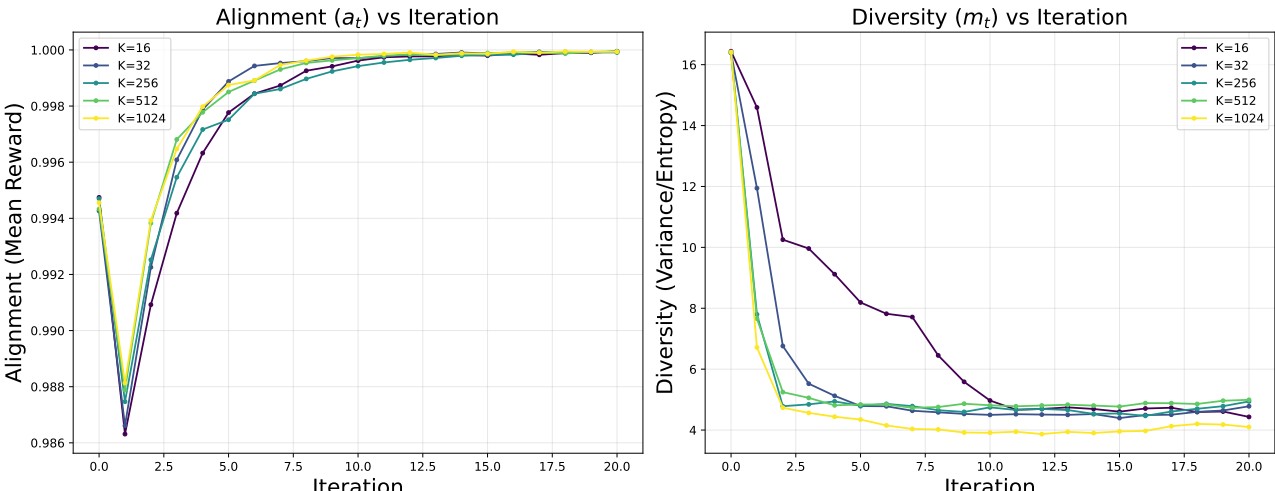

*Figure 11.* **Finite-$K$ ablation validates the $K \to \infty$ theory.** Toy retraining dynamics under Best-of-$K$ curation (with replacement) for varying pool size $K$. **Left:** alignment $a_t$ (mean reward of samples drawn from $p_t$) versus iteration. **Right:** diversity proxy $m_t$ (entropy/variance-based) versus iteration.

**Impact of reward distance.** Fig. 15 shows both the initial model distribution (top rows) and the final learned distribution (bottom rows) across different separation values. When the means are far apart (e.g., distance $= 6$ or $4$), the model converges to a well-separated mixture with substantial mass on both reward regions. As the distance decreases, we observe a soft collapse in which components merge toward a compromise mode. At distance $= 0$, pluralistic curation becomes ineffective because the two preferences are indistinguishable, yielding full collapse. This reinforces the main-paper phase transition: diversity is only preserved when competing rewards induce sufficiently distinct optima.

**Impact of $K$ pool-size (finite-$K$ ablation).** To investigate the effect of selection pressure, we ablate the pool size parameter $K$ used in the curation mechanism.

We fix $T = 20$, $N_{\text{pool}} = 50{,}000$, $N_{\text{keep}} = 2{,}500$, and reward scaling $\gamma = 10.0$. We sweep $K \in \{16, 32, 256, 512, 1024\}$ and track: *alignment $a_t$* (mean reward of samples drawn from $p_t$) and a *diversity proxy $m_t$* (entropy/variance-based) over iterations.

Fig. 11 reveals a transition from finite-$K$ to large-$K$ behavior. For small $K$ (e.g., 16 or 32), the curation step is noisier, producing slower improvement of $a_t$ and a more gradual decay of $m_t$. As $K$ increases, the trajectories rapidly concentrate: for $K \geq 256$, alignment curves nearly coincide and reach a near-saturated level early in training, while diversity collapses quickly and then plateaus. Beyond this moderate threshold, the observed dynamics are effectively indistinguishable from the $K \to \infty$ regime, supporting our theoretical focus on the large-$K$ limit as a predictive approximation for practical finite-$K$ settings.

## C.9. Empirical Corroboration

In this section, we provide empirical corroborations of our main theoretical results from Section 3. Specifically, we verify the quantitative bounds on basin mass under leakage from Lemma B.5, the exponential decay of mass outside the near-optimal basins from Lemma B.4, and the robustness of the Nash bargaining interpretation from Theorem B.7 under varying reward separations.

### C.9.1. LEAKAGE BOUNDS AND MASS PARTITIONING

**Experimental Setting.** We vary the mixing weight $q \in \{0.1, 0.25, 0.4, 0.6, 0.75, 0.9\}$ and the distance $d \in [0, 8]$. For each configuration, we run the infinite-$K$ update rule for 50 iterations and measure the final probability mass $a_\infty = p_\infty(S_{1,\epsilon})$ concentrated in the first basin, defined with $\epsilon = 0.1$. We strictly calculate the theoretical upper and lower bounds derived in Lemma 3.4 using the analytical leakage factors $\kappa_1, \kappa_2$.

**Analysis.** Fig. 12 (and Table 3) presents the results. As predicted, when the separation distance $d$ is small ($d < 2$),

the leakage factors $\kappa$ correspond to loose bounds, and the empirical mass deviates from the target $q$ due to cross-basin reinforcement. However, as separation increases ($d > 4$), the bounds tighten exponentially effectively collapsing onto the line $a_\infty = q$. Importantly, for all tested values of $q \in (0, 1)$, the empirical mass remains strictly within the theoretical "allowable region" (shaded grey). This confirms that Lemma 3.4 correctly characterizes the worst-case deviation induced by reward leakage.

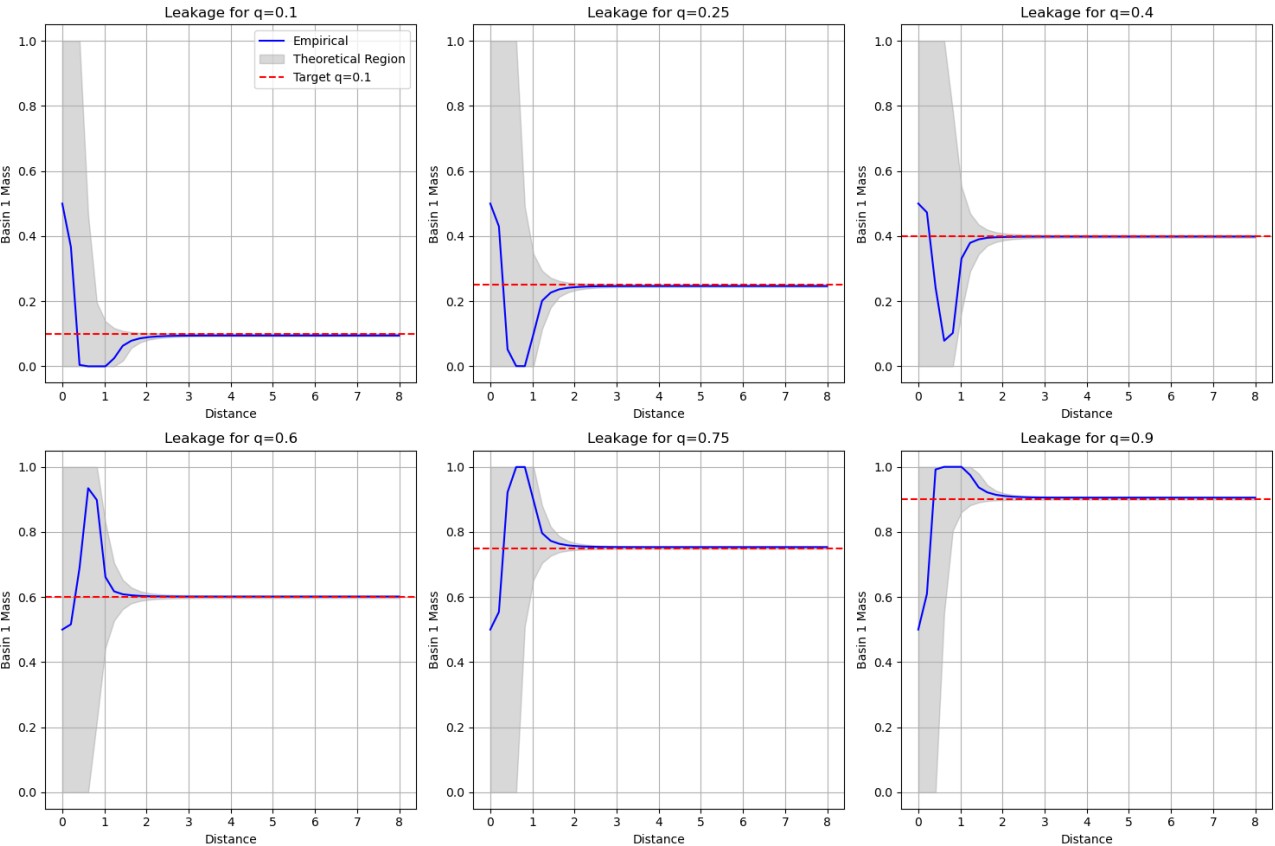

*Figure 12.* **Corroboration of Leakage Bounds (Lemma 3.4).** Final basin mass $a_\infty$ (blue line) versus mode distance for varying preference weights $q$. The shaded grey region represents the theoretical bounds derived from leakage factors. As separation increases, leakage vanishes and the mass converges exactly to $q$.

### C.9.2. CONVERGENCE SPEED AND OUTSIDE MASS DECAY

**Experimental Setting.** To verify the cleaning dynamics established in Lemma 3.3, we track the total probability mass $m_t = p_t(\mathcal{X} \setminus S_\epsilon)$ outside the $\epsilon$-optimal basins over retraining iterations. We initialize with a uniform distribution and fix $q = 0.5$. We vary the mode separation $d \in \{2.0, 4.0, 6.0, 8.0\}$ to observe the effect of reward landscape geometry on convergence speed.

**Analysis.** (Fig. 13, Table 4). We observe a strictly linear decrease in $\log m_t$, confirming that the outside mass decays exponentially as predicted ($m_{t+1} \le \rho_\epsilon m_t$). Furthermore, the rate of decay (slope) increases with separation distance. At $d = 8.0$, the mass vanishes almost instantly (slope $\approx -13.6$), whereas for $d = 2.0$ the cleaning process is slower (slope $\approx -1.3$) due to the shallowness of the reward gradients between modes. This validates that pluralistic curation efficiently concentrates mass onto the union of high-reward basins.

### C.9.3. ROBUSTNESS OF THE NASH BARGAINING SOLUTION

**Experimental Setting.** Theorem 3.7 establishes that in the limit of hard separation (vanishing leakage), the limiting distribution is the maximizer of the weighted Nash product with weight $q$. To test the robustness of this interpretation in non-ideal settings, we sweep the preference weight $q \in [0.1, 0.9]$ and measure the resulting empirical basin weight $a_\infty$. We

*Table 3.* Validation of leakage bounds across preference weights $q$ and distances $D$. For each configuration, we report the theoretical lower and upper bounds on basin mass together with the empirically observed mass $a_\infty$.

| Preference Weight $q$ | Distance $D$ | Lower Bound | Empirical Mass $a_\infty$ | Upper Bound |
|:---:|:---:|:---:|:---:|:---:|
| | | *Low preference weights* | | |
| $q = 0.1$ | 2.0 | 0.082 | **0.090** | 0.102 |
| $q = 0.1$ | 4.0 | 0.082 | **0.090** | 0.102 |
| $q = 0.1$ | 6.0 | 0.093 | **0.095** | 0.101 |
| | | *Moderate preference weights* | | |
| $q = 0.2$ | 2.0 | 0.235 | **0.244** | 0.255 |
| $q = 0.2$ | 4.0 | 0.235 | **0.244** | 0.255 |
| $q = 0.2$ | 6.0 | 0.244 | **0.247** | 0.252 |
| $q = 0.4$ | 2.0 | 0.388 | **0.397** | 0.408 |
| $q = 0.4$ | 4.0 | 0.388 | **0.397** | 0.408 |
| $q = 0.4$ | 6.0 | 0.395 | **0.399** | 0.403 |
| | | *High preference weights* | | |
| $q = 0.6$ | 2.0 | 0.592 | **0.603** | 0.612 |
| $q = 0.6$ | 4.0 | 0.592 | **0.603** | 0.612 |
| $q = 0.6$ | 6.0 | 0.597 | **0.601** | 0.605 |
| $q = 0.8$ | 2.0 | 0.745 | **0.756** | 0.765 |
| $q = 0.8$ | 4.0 | 0.745 | **0.756** | 0.765 |
| $q = 0.8$ | 6.0 | 0.748 | **0.753** | 0.756 |
| $q = 0.9$ | 2.0 | 0.898 | **0.910** | 0.918 |
| $q = 0.9$ | 4.0 | 0.898 | **0.910** | 0.918 |
| $q = 0.9$ | 6.0 | 0.899 | **0.905** | 0.907 |

*Table 4.* Convergence speed as a function of mode separation. We report the empirical decay rate given by the slope of $\log m_t$, where $m_t$ denotes the outside-basin mass.

| Mode Distance $D$ | Empirical Decay Rate |
|:---:|:---:|
| 2.0 | $-0.5650$ |
| 4.0 | $-0.5855$ |
| 6.0 | $-0.5854$ |
| 8.0 | $-0.5852$ |

compare this to the "ideal" Nash prediction $a_\infty = q$ across different separation distances $d \in \{1.0, 2.0, 3.0, 5.0\}$.

**Analysis.** Fig. 14 shows the deviation from the ideal identity line. At $d = 5.0$ (hard separation), the empirical curve perfectly overlaps with the diagonal $y = x$, indicating that the system behaves exactly as a Weighted Nash Bargainer. As distance decreases to $d = 1.0$, significant deviations occur where the system "under-bargains" (mass does not fully shift to the preferred basin) due to the mixing induced by leakage. Table 5 quantifies this via the Mean Squared Error (MSE) from the ideal. The rapid decay of MSE as $d$ increases suggests that the Nash interpretation provides a highly accurate model for pluralistic curation even at moderate reward separations.

*Table 5.* Robustness of the Nash approximation under increasing mode separation. We report the mean squared error between the empirically recovered mixture weight $\alpha$ and the Nash prediction $q$.

| Mode Distance $D$ | Nash Approximation MSE |
|:---:|:---:|
| 1.0 | $1.66 \times 10^{-2}$ |
| 2.0 | $5.06 \times 10^{-5}$ |
| 3.0 | $1.40 \times 10^{-5}$ |
| 5.0 | $1.25 \times 10^{-5}$ |

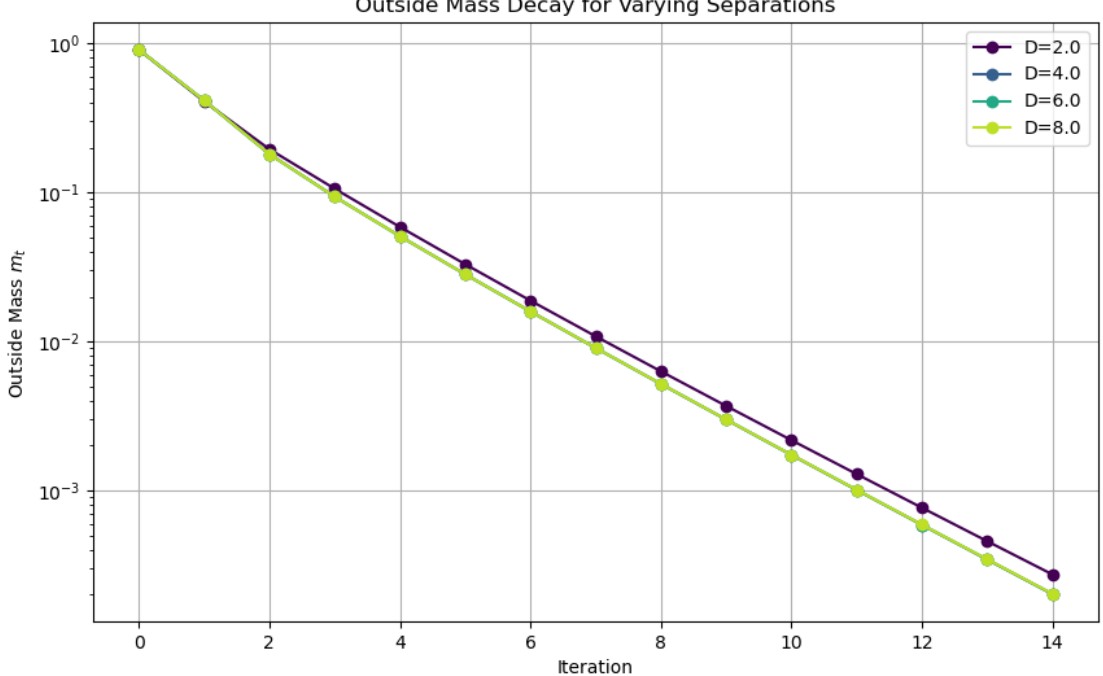

*Figure 13.* **Evolution of Outside Mass $m_t$.** The logarithmic scale reveals an exponential decay of mass located outside the near-optimal basins. Greater mode separation accelerates this convergence.

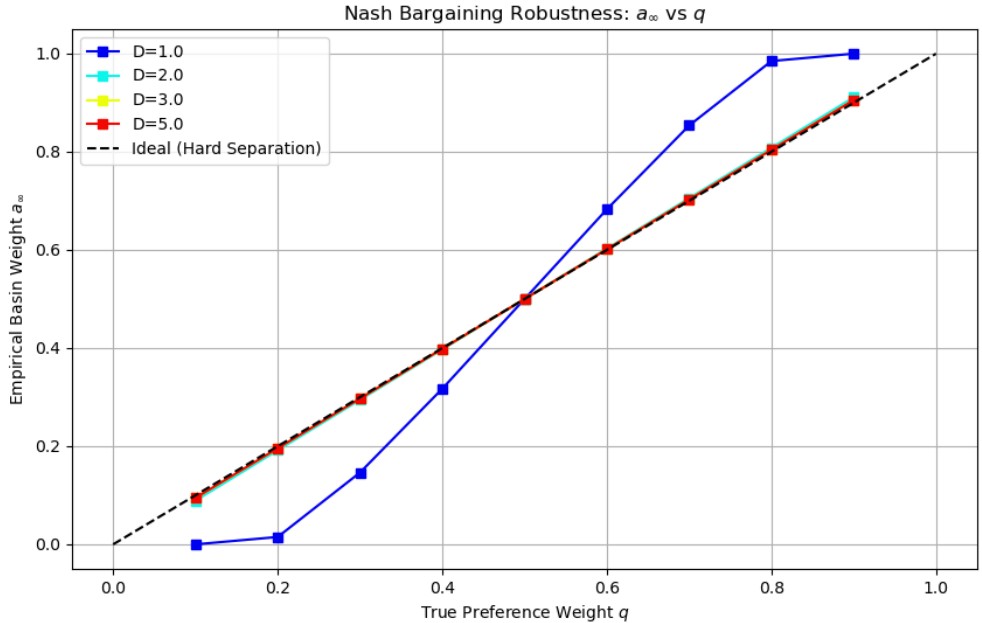

*Figure 14.* **Robustness of Nash Bargaining.** The plot compares the empirical basin weight $a_\infty$ against the input preference weight $q$. In the hard separation limit ($d = 5.0$), the system perfectly matches the Nash prediction (diagonal). Smaller separations introduce consistent deviations due to leakage.

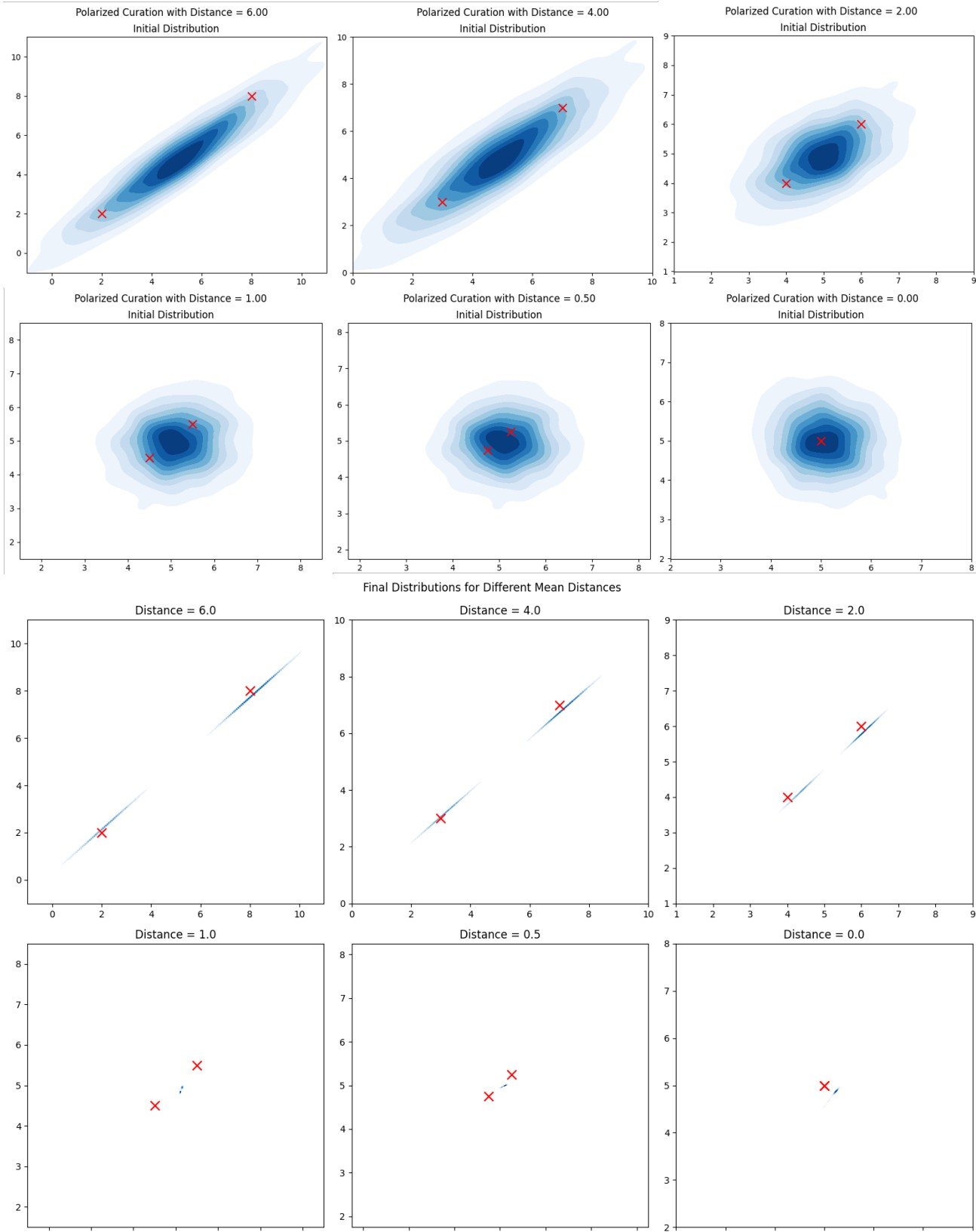

*Figure 15.* **Effect of Reward Mode Separation on Final Distribution.** Top: Initial model distribution under pluralistic curation with varying reward mean distances. Bottom: Final distributions after retraining. When the reward maxima are well-separated (e.g., distances 6 or 4), pluralistic curation leads to a balanced mixture across modes. As distance decreases, the model gradually collapses to a single compromise mode. At distance = 0, the rewards are indistinguishable and diversity is lost.

