$$S_{1,\varepsilon} \cap S_{2,\varepsilon} = \emptyset.$$

**Assumption B.4** (Outside-basin suboptimality). *Fix $\varepsilon > 0$. There exists $\delta_{c,\varepsilon} > 0$ such that for all $x \notin \mathcal{S}_\varepsilon$,*

$$r_1(x) \leq r_1^* - \delta_{c,\varepsilon}, \qquad r_2(x) \leq r_2^* - \delta_{c,\varepsilon}.$$

**Assumption B.5** (Cross-basin separation (leakage margins)). *Fix $\varepsilon > 0$ and suppose Assumption B.3 holds. There exist gaps $\Delta_1, \Delta_2 > 0$ such that*

$$x \in S_{1,\varepsilon} \; \Rightarrow \; r_2(x) \leq r_2^* - \Delta_2 + \varepsilon, \qquad x \in S_{2,\varepsilon} \; \Rightarrow \; r_1(x) \leq r_1^* - \Delta_1 + \varepsilon.$$

*Equivalently, points that are $\varepsilon$-optimal for one preference are uniformly suboptimal for the other. When $\Delta_i > 2\varepsilon$, this yields the leakage factors*

$$\kappa_1 := \exp\big(-(\Delta_1 - 2\varepsilon)\big), \qquad \kappa_2 := \exp\big(-(\Delta_2 - 2\varepsilon)\big),$$

*that appear in the basin-mass bounds.*

**Assumption B.6** (Plateau-basin idealization (used only for explicit mixture and bargaining)). *Fix $\varepsilon > 0$ and suppose Assumptions B.2–B.3 hold. Assume rewards are (approximately) constant on each $\varepsilon$-basin: there exist constants $U_1, U_2 \in \mathbb{R}$ and gaps $\Delta_1, \Delta_2 > 0$ such that*

$$r_1(x) = U_1, \quad r_2(x) = U_2 - \Delta_2 \quad \text{for all } x \in S_{1,\varepsilon},$$

$$r_1(x) = U_1 - \Delta_1, \quad r_2(x) = U_2 \quad \text{for all } x \in S_{2,\varepsilon}.$$

*This idealization makes the basin-conditional distributions invariant and yields the explicit q-mixture and Nash bargaining characterization.*

**Interpretation.** Assumptions B.4 and B.5 formalize a two-basin regime: (i) points outside $\mathcal{S}_\varepsilon$ are jointly suboptimal and their total mass decays over retraining rounds; (ii) within $\mathcal{S}_\varepsilon$, cross-basin rewards are controlled, producing explicit leakage factors that quantify how close the limiting mixture weight is to $q$. Assumption B.6 is used only for the clean closed-form mixture and bargaining interpretation, and can be viewed as an analytically transparent approximation to the general case.

## B.3. Proofs

**Lemma B.1** (Pluralistic Curation Update). *At retraining step $t$, the reward is chosen at random, $r_1$ with probability $q$ and $r_2$ with probability $1 - q$. The model update obeys*

$$p_{t+1}(x) = p_t(x)\Big(q\, H_{p_t}^{K,r_1}(x) + (1-q)\, H_{p_t}^{K,r_2}(x)\Big), \tag{1}$$

*where for $i \in \{1, 2\}$*

$$H_{p_t}^{K,r_i}(x) := \mathbb{E}_{y_1,\ldots,y_{K-1} \sim p_t}\left[\frac{K\, e^{r_i(x)}}{e^{r_i(x)} + \sum_{j=1}^{K-1} e^{r_i(y_j)}}\right] \tag{2}$$

*is the discrete choice curation weight under reward $r_i$.*

*Proof.* Condition on a fixed reward $r_i$. Given $x_{1:K} \sim p_t^{\otimes K}$, the BT selection rule is

$$\Pr(\hat{x} = x_k \mid x_{1:K}, r_i) = \frac{e^{r_i(x_k)}}{\sum_{j=1}^{K} e^{r_i(x_j)}}. \tag{3}$$

By exchangeability, the marginal density of $\hat{x}$ under $r_i$ is

$$p_{\hat{x}|r_i}(x) = p_t(x)\, \mathbb{E}_{y_{1:K-1} \sim p_t}\left[\frac{K e^{r_i(x)}}{e^{r_i(x)} + \sum_{j=1}^{K-1} e^{r_i(y_j)}}\right] = p_t(x)\, H_{p_t}^{K,r_i}(x). \tag{4}$$

Mixing over the random reward $R \in \{r_1, r_2\}$ with $\Pr(R = r_1) = q$ gives

$$p_{\hat{x}}(x) = q\, p_{\hat{x}|r_1}(x) + (1-q)\, p_{\hat{x}|r_2}(x) = p_t(x)\big(q H_{p_t}^{K,r_1}(x) + (1-q)H_{p_t}^{K,r_2}(x)\big). \tag{5}$$

Under the infinite-capacity MLE idealization, retraining matches the law of selected samples, hence $p_{t+1} = p_{\hat{x}}$, which is the desired update. $\square$

**Lemma B.2** (Concentration of curation weights). *Let $r : \mathcal{X} \to \mathbb{R}$ be bounded and measurable, and let $p$ be any probability density. Then, as $K \to \infty$,*

$$H_p^{K,r}(x) \longrightarrow H_{\infty,r}^p(x) := \frac{e^{r(x)}}{\mathbb{E}_{y \sim p}[e^{r(y)}]} \qquad \textit{for every } x \in \mathcal{X}. \tag{6}$$

*Proof.* Let $Y_j := e^{r(y_j)}$ with $y_j \overset{\text{i.i.d.}}{\sim} p$, and set $Z_{K-1} := \sum_{j=1}^{K-1} Y_j$ and

$$X_K(x) := K \frac{e^{r(x)}}{e^{r(x)} + Z_{K-1}}, \qquad H_p^{K,r}(x) = \mathbb{E}[X_K(x)]. \tag{7}$$

Since $r$ is bounded, $Y_j$ is integrable and bounded, and $\mu_e := \mathbb{E}[Y_1] \in (0, \infty)$. By the strong law of large numbers, $\frac{Z_{K-1}}{K-1} \to \mu_e$ almost surely, hence

$$X_K(x) = \frac{e^{r(x)}}{\frac{e^{r(x)}}{K} + \frac{K-1}{K} \cdot \frac{Z_{K-1}}{K-1}} \xrightarrow[K \to \infty]{\text{a.s.}} \frac{e^{r(x)}}{\mu_e}. \tag{8}$$

Also, boundedness of $r$ implies $0 \le X_K(x) \le e^{r_{\max} - r_{\min}}$ for all $K \ge 2$. By dominated convergence,

$$H_p^{K,r}(x) = \mathbb{E}[X_K(x)] \to \frac{e^{r(x)}}{\mu_e} = \frac{e^{r(x)}}{\mathbb{E}_{y \sim p}[e^{r(y)}]}. \tag{9}$$

$\square$

**Lemma B.3** (Finite-$K$ approximation (soft bound)). *Let $r$ be bounded with $r_{\min} \le r(x) \le r_{\max}$ and set*

$$A := e^{r_{\min}}, \qquad B := e^{r_{\max}}, \qquad \mu_e := \mathbb{E}_{y \sim p}[e^{r(y)}] \in [A, B]. \tag{10}$$

*Then for all $K \ge 3$ and all $x \in \mathcal{X}$,*

$$\left| H_p^{K,r}(x) - H_{\infty,r}^p(x) \right| \le C_1 \sqrt{\frac{\log K}{K}} + \frac{C_2}{K}, \tag{11}$$

*where $C_1, C_2$ depend only on $A, B$ (equivalently on $r_{\max} - r_{\min}$).*

*Proof.* Let $Y_j := e^{r(y_j)} \in [A, B]$ and $\overline{Y} := \frac{1}{K-1} \sum_{j=1}^{K-1} Y_j$, so $Z_{K-1} = (K-1)\overline{Y}$. Hoeffding implies that for any $\eta > 0$,

$$\Pr(|\overline{Y} - \mu_e| \ge \eta) \le 2 \exp\left( -\frac{2(K-1)\eta^2}{(B-A)^2} \right). \tag{12}$$

On the event $E_\eta := \{|\overline{Y} - \mu_e| < \eta\}$ with $\eta \le A/2$, we have $\overline{Y} \ge A/2$ and

$$X_K(x) = \frac{e^{r(x)}}{\frac{e^{r(x)}}{K} + \frac{K-1}{K}\overline{Y}}. \tag{13}$$

A direct Lipschitz bound of the map $z \mapsto \frac{e^{r(x)}}{\frac{e^{r(x)}}{K} + \frac{K-1}{K}z}$ on $z \in [A/2, B]$ yields constants $C', C''$ depending only on $A, B$ such that on $E_\eta$,

$$\left| X_K(x) - \frac{e^{r(x)}}{\mu_e} \right| \le \frac{C'}{K} + C'' \eta. \tag{14}$$

On $E_\eta^c$ we use the trivial bound $0 \le X_K(x) \le B/A$. Taking expectations gives

$$\left| H_p^{K,r}(x) - H_{\infty,r}^p(x) \right| = \left| \mathbb{E}[X_K(x)] - \frac{e^{r(x)}}{\mu_e} \right| \le \frac{C'}{K} + C'' \eta + \frac{B}{A} \Pr(E_\eta^c). \tag{15}$$

Choose $\eta := (B-A)\sqrt{\frac{\log K}{2(K-1)}}$ so that $\Pr(E_\eta^c) \le 2/K$. Absorbing constants yields

$$\left| H_p^{K,r}(x) - H_{\infty,r}^p(x) \right| \le C_1 \sqrt{\frac{\log K}{K}} + \frac{C_2}{K}. \tag{16}$$

with $C_1, C_2$ depending only on $A, B$. $\square$

**Lemma B.4** (Decay outside $\varepsilon$-optimal regions). *Fix $\varepsilon > 0$ and define*

$$S_{1,\varepsilon} := \{x \in \mathcal{X} : r_1(x) \geq r_1^* - \varepsilon\}, \qquad S_{2,\varepsilon} := \{x \in \mathcal{X} : r_2(x) \geq r_2^* - \varepsilon\}, \qquad \mathcal{S}_\varepsilon := S_{1,\varepsilon} \cup S_{2,\varepsilon}.$$

*Let*

$$m_t := \int_{\mathcal{X} \setminus \mathcal{S}_\varepsilon} p_t(x) \, dx.$$

*Assume bounded rewards $r_i \in [L_i, U_i]$ and work in the $K \to \infty$ regime of Lemma B.2. Assume there exists a uniform gap $\delta_{c,\varepsilon} > 0$ such that for all $x \notin \mathcal{S}_\varepsilon$,*

$$r_1(x) \leq U_1 - \delta_{c,\varepsilon}, \qquad r_2(x) \leq U_2 - \delta_{c,\varepsilon}. \tag{17}$$

*If*

$$\delta_{c,\varepsilon} > \log\left(q \, e^{U_1 - L_1} + (1 - q) \, e^{U_2 - L_2}\right), \tag{18}$$

*then there exists $\rho_\varepsilon \in (0, 1)$ such that*

$$m_{t+1} \leq \rho_\varepsilon \, m_t \quad \text{for all } t, \qquad \text{hence} \qquad m_t \leq \rho_\varepsilon^t \, m_0 \to 0.$$

*In particular, with $a_t := \int_{S_{1,\varepsilon}} p_t$ and $b_t := \int_{S_{2,\varepsilon}} p_t$, we have*

$$a_t + b_t = 1 - m_t \to 1.$$

*Proof.* In the $K \to \infty$ regime,

$$p_{t+1}(x) = p_t(x)\left(q \frac{e^{r_1(x)}}{\mu_1(t)} + (1 - q) \frac{e^{r_2(x)}}{\mu_2(t)}\right), \qquad \mu_i(t) = \mathbb{E}_{p_t}[e^{r_i}]. \tag{19}$$

For $x \notin \mathcal{S}_\varepsilon$, the gap assumption gives $r_i(x) \leq U_i - \delta_{c,\varepsilon}$, hence $e^{r_i(x)} \leq e^{U_i - \delta_{c,\varepsilon}}$. Also $\mu_i(t) \geq e^{L_i}$ since $r_i \geq L_i$ everywhere. Therefore, for all $x \notin \mathcal{S}_\varepsilon$,

$$q \frac{e^{r_1(x)}}{\mu_1(t)} + (1 - q) \frac{e^{r_2(x)}}{\mu_2(t)} \leq e^{-\delta_{c,\varepsilon}}\left(q e^{U_1 - L_1} + (1 - q) e^{U_2 - L_2}\right) =: \rho_\varepsilon. \tag{20}$$

By (18), $\rho_\varepsilon < 1$. Integrating over $\mathcal{X} \setminus \mathcal{S}_\varepsilon$ yields

$$m_{t+1} = \int_{\mathcal{X} \setminus \mathcal{S}_\varepsilon} p_{t+1} \leq \rho_\varepsilon \int_{\mathcal{X} \setminus \mathcal{S}_\varepsilon} p_t = \rho_\varepsilon m_t. \tag{21}$$

Iterating gives $m_t \leq \rho_\varepsilon^t m_0 \to 0$, hence $a_t + b_t = 1 - m_t \to 1$. $\square$

**Lemma B.5** (Mass partitioning with leakage bounds). *Work in the $K \to \infty$ regime of Lemma B.2. Assume bounded rewards $r_i \in [L_i, U_i]$. Assume the following basin-separation conditions for some $\varepsilon > 0$ and gaps $\Delta_1, \Delta_2 > 2\varepsilon$:*

$$x \in S_{1,\varepsilon} \implies r_1(x) \geq U_1 - \varepsilon \text{ and } r_2(x) \leq U_2 - \Delta_2 + \varepsilon, \tag{22}$$
$$x \in S_{2,\varepsilon} \implies r_2(x) \geq U_2 - \varepsilon \text{ and } r_1(x) \leq U_1 - \Delta_1 + \varepsilon. \tag{23}$$

*Define leakage factors*

$$\kappa_1 := e^{-(\Delta_1 - 2\varepsilon)}, \qquad \kappa_2 := e^{-(\Delta_2 - 2\varepsilon)}. \tag{24}$$

*Then for every t,*

$$a_{t+1} = q \frac{\int_{S_{1,\varepsilon}} e^{r_1(x)} p_t(x) \, dx}{\int_{\mathcal{X}} e^{r_1(x)} p_t(x) \, dx} + (1 - q) \frac{\int_{S_{1,\varepsilon}} e^{r_2(x)} p_t(x) \, dx}{\int_{\mathcal{X}} e^{r_2(x)} p_t(x) \, dx}, \tag{25}$$

and the two terms admit the bounds

$$q \frac{a_t}{a_t + \kappa_1 b_t + m_t e^{-(\delta_c - \varepsilon)}} \leq q \frac{\int_{S_{1,\varepsilon}} e^{r_1} p_t}{\int_{\mathcal{X}} e^{r_1} p_t} \leq q, \tag{26}$$

$$0 \leq (1 - q) \frac{\int_{S_{1,\varepsilon}} e^{r_2} p_t}{\int_{\mathcal{X}} e^{r_2} p_t} \leq (1 - q) \frac{\kappa_2 a_t}{b_t + \kappa_2 a_t}, \tag{27}$$

where $\delta_c > 0$ is any constant such that $x \notin \mathcal{S}_\varepsilon \Rightarrow r_i(x) \leq U_i - \delta_c$ (as in Lemma B.4, with $\mathcal{S}_\varepsilon$ in place of $S_1 \cup S_2$).

Consequently, if $m_t \to 0$ and $(a_t)$ converges to a limit $a_\infty \in (0, 1)$, then necessarily

$$\frac{q - \kappa_1}{1 - \kappa_1} \leq a_\infty \leq \frac{q}{1 - \kappa_2}, \tag{28}$$

provided $q > \kappa_1$ and $1 - q > \kappa_2$ so that the interval is nontrivial and lies in $(0, 1)$. Moreover, as $\Delta_1, \Delta_2 \to \infty$ (equivalently $\kappa_1, \kappa_2 \to 0$), the interval collapses to $a_\infty = q$.

*Proof.* In the $K \to \infty$ regime,

$$p_{t+1}(x) = p_t(x) \Big( q \frac{e^{r_1(x)}}{\mu_1(t)} + (1 - q) \frac{e^{r_2(x)}}{\mu_2(t)} \Big), \qquad \mu_i(t) = \int_{\mathcal{X}} e^{r_i} p_t. \tag{29}$$

Integrating over $S_{1,\varepsilon}$ gives (25).

For the $r_1$-fraction, by basin separation, $e^{r_1} \geq e^{U_1 - \varepsilon}$ on $S_{1,\varepsilon}$, $e^{r_1} \leq e^{U_1 - \Delta_1 + \varepsilon} = \kappa_1 e^{U_1 - \varepsilon}$ on $S_{2,\varepsilon}$, and $e^{r_1} \leq e^{U_1 - \delta_c} = e^{-(\delta_c - \varepsilon)} e^{U_1 - \varepsilon}$ on $\mathcal{X} \setminus \mathcal{S}_\varepsilon$. Hence

$$\int_{S_{1,\varepsilon}} e^{r_1} p_t \geq a_t e^{U_1 - \varepsilon}, \qquad \mu_1(t) \leq e^{U_1 - \varepsilon} \big( a_t + \kappa_1 b_t + m_t e^{-(\delta_c - \varepsilon)} \big), \tag{30}$$

which yields the lower bound in (26). The upper bound $\leq q$ is immediate since the fraction is $\leq 1$.

For the $r_2$-fraction, on $S_{1,\varepsilon}$ we have $e^{r_2} \leq e^{U_2 - \Delta_2 + \varepsilon} = \kappa_2 e^{U_2 - \varepsilon}$, so

$$\int_{S_{1,\varepsilon}} e^{r_2} p_t \leq \kappa_2 a_t e^{U_2 - \varepsilon}. \tag{31}$$

Also $\mu_2(t) = \int_{S_{1,\varepsilon}} e^{r_2} p_t + \int_{S_{2,\varepsilon}} e^{r_2} p_t + \int_{\mathcal{X} \setminus \mathcal{S}_\varepsilon} e^{r_2} p_t \geq \int_{S_{2,\varepsilon}} e^{r_2} p_t \geq b_t e^{U_2 - \varepsilon}$, and in fact $\mu_2(t) \geq b_t e^{U_2 - \varepsilon} + \int_{S_{1,\varepsilon}} e^{r_2} p_t$. Therefore,

$$\frac{\int_{S_{1,\varepsilon}} e^{r_2} p_t}{\mu_2(t)} \leq \frac{\kappa_2 a_t e^{U_2 - \varepsilon}}{b_t e^{U_2 - \varepsilon} + \kappa_2 a_t e^{U_2 - \varepsilon}} = \frac{\kappa_2 a_t}{b_t + \kappa_2 a_t}, \tag{32}$$

which gives (27).

For the limit interval, assume $m_t \to 0$ and $a_t \to a_\infty \in (0, 1)$, hence $b_t \to 1 - a_\infty$. From (26) and $a_{t+1} \geq q \frac{a_t}{a_t + \kappa_1 b_t + o(1)}$, taking limits yields

$$a_\infty \geq q \frac{a_\infty}{a_\infty + \kappa_1 (1 - a_\infty)} \quad \Longrightarrow \quad a_\infty \geq \frac{q - \kappa_1}{1 - \kappa_1}. \tag{33}$$

Applying the analogous bound to $b_{t+1}$ on $S_{2,\varepsilon}$ gives $b_\infty \geq \frac{(1-q)-\kappa_2}{1-\kappa_2}$ and hence $a_\infty \leq \frac{q}{1-\kappa_2}$. Finally, $\kappa_1, \kappa_2 \to 0$ forces the interval to shrink to $\{q\}$. $\qquad \square$

Uniform non-collapse under bounded leakage

**Lemma B.6** (Uniform non-collapse under bounded leakage). *Fix $\varepsilon > 0$ and work in the large-$K$ regime. Assume the decay conclusion of Lemma B.4. Define leakage factors*

$$\kappa_1 := \exp\big(-(\Delta_1 - 2\varepsilon)\big), \qquad \kappa_2 := \exp\big(-(\Delta_2 - 2\varepsilon)\big),$$

*and suppose $q \in (\kappa_1, 1 - \kappa_2)$. Then there exist constants $c_\varepsilon > 0$ and $t_0$ such that for all $t \geq t_0$,*

$$\max\left\{0, \frac{q - \kappa_1}{1 - \kappa_1}\right\} - c_\varepsilon m_t \leq a_t \leq \min\left\{1, \frac{q}{1 - \kappa_2}\right\} + c_\varepsilon m_t.$$

*In particular, there exists $\eta > 0$ such that for all $t \geq t_0$, $\eta \leq a_t \leq 1 - \eta$.*

*Proof.* Let $A := S_{1,\varepsilon}$, $B := S_{2,\varepsilon}$, and $O := \mathcal{X} \setminus (A \cup B)$, with $a_t = p_t(A)$, $b_t = p_t(B)$, $m_t = p_t(O)$. Work in the large-$K$ regime and write the update as

$$p_{t+1} = q\, p_t^{(1)} + (1 - q)\, p_t^{(2)}, \qquad p_t^{(i)}(dx) = \frac{e^{r_i(x)}}{Z_i(t)} p_t(dx). \tag{34}$$

*Lower bound.* Dropping the nonnegative term gives $a_{t+1} \geq q\, p_t^{(1)}(A)$. By the same leakage ratio bound used in Lemma B.5, there is $C_\varepsilon > 0$ such that

$$p_t^{(1)}(A) \geq \frac{a_t}{a_t + \kappa_1 b_t + C_\varepsilon m_t} \geq \frac{a_t}{a_t + \kappa_1(1 - a_t) + C_\varepsilon m_t}. \tag{35}$$

Define $g(a) := q \frac{a}{a + \kappa_1(1 - a)}$ and note its unique fixed point on $(0, 1)$ is $L = \frac{q - \kappa_1}{1 - \kappa_1} > 0$ since $q > \kappa_1$.

*Upper bound.* Similarly $b_{t+1} \geq (1 - q)p_t^{(2)}(B)$ and Lemma B.5 gives

$$p_t^{(2)}(B) \geq \frac{b_t}{b_t + \kappa_2 a_t + C_\varepsilon m_t}. \tag{36}$$

Using $a_{t+1} = 1 - b_{t+1} - m_{t+1}$ and Lemma B.4 so that $m_{t+1} \leq \rho_\varepsilon m_t$, we obtain

$$a_{t+1} \leq h(a_t) + C_\varepsilon m_t, \qquad h(a) := 1 - (1 - q)\frac{1 - a}{(1 - a) + \kappa_2 a}. \tag{37}$$

whose unique fixed point on $(0, 1)$ is $U = \frac{q}{1 - \kappa_2} < 1$ since $q < 1 - \kappa_2$.

Since $m_t \to 0$, pick $t_0$ so that $C_\varepsilon m_t \leq \frac{1}{2}\min\{L, 1 - U\}$ for all $t \geq t_0$. Monotonicity of $g, h$ and their one-sided drift toward $L, U$ imply by induction that for all $t \geq t_0$,

$$L - C_\varepsilon m_t \leq a_t \leq U + C_\varepsilon m_t,$$

which is the stated bound (with clipping if desired). Setting $\eta := \frac{1}{2}\min\{L, 1 - U\}$ gives $\eta \leq a_t \leq 1 - \eta$ for $t \geq t_0$. $\square$

**Corollary B.1** (Nondegenerate limit points). *Under the assumptions of Lemma B.6, let $t_k \to \infty$ and suppose $p_{t_k} \Rightarrow p_\infty$. Assume additionally that $S_{1,\varepsilon}$ is a continuity set for $p_\infty$, i.e. $p_\infty(\partial S_{1,\varepsilon}) = 0$ (for example, if $r_1$ is continuous and $\varepsilon$ is chosen so that the level set $\{r_1 = r_1^* - \varepsilon\}$ has $p_\infty$-measure zero). Then*

$$p_\infty(S_{1,\varepsilon}) \in [\eta, 1 - \eta],$$

*with the same $\eta$ as in Lemma B.6. Hence no such limit point collapses onto a single basin.*

*Proof.* Lemma B.6 gives $\eta \leq a_t \leq 1 - \eta$ for all $t \geq t_0$. Along any subsequence $t_k \to \infty$, we therefore have $\eta \leq \liminf_k a_{t_k} \leq \limsup_k a_{t_k} \leq 1 - \eta$. If $S_{1,\varepsilon}$ is a continuity set for $p_\infty$ and $p_{t_k} \Rightarrow p_\infty$, then $p_{t_k}(S_{1,\varepsilon}) \to p_\infty(S_{1,\varepsilon})$. Since $p_{t_k}(S_{1,\varepsilon}) = a_{t_k}$, it follows that $p_\infty(S_{1,\varepsilon}) \in [\eta, 1 - \eta]$. $\square$

**Theorem B.2** (Two-basin subsequential limit (general case))**.** *Assume the conditions of Lemma B.5 and assume $m_t \to 0$. Consider any subsequence $t_k \to \infty$ along which $p_{t_k} \Rightarrow p_\infty$ weakly. Then $p_\infty$ is supported on $\mathcal{S}_\varepsilon$ and admits the decomposition*

$$p_\infty = a_\infty\, p_{\infty,1} + (1 - a_\infty)\, p_{\infty,2}, \tag{38}$$

*where*

$$a_\infty := p_\infty(S_{1,\varepsilon}), \qquad p_{\infty,1} := p_\infty(\cdot \mid S_{1,\varepsilon}), \qquad p_{\infty,2} := p_\infty(\cdot \mid S_{2,\varepsilon}). \tag{39}$$

*Moreover, if $a_t \to a_\infty \in (0,1)$, then $a_\infty$ satisfies the leakage-controlled interval bound*

$$\frac{q - \kappa_1}{1 - \kappa_1} \;\le\; a_\infty \;\le\; \frac{q}{1 - \kappa_2}, \tag{40}$$

*with $\kappa_1, \kappa_2$ as in Lemma B.5. In particular, as $\Delta_1, \Delta_2 \to \infty$ (so $\kappa_1, \kappa_2 \to 0$), the basin mass converges to $q$.*

*Proof.* Write for each $t$ the decomposition

$$p_t = a_t\, p_t(\cdot \mid S_{1,\varepsilon}) + b_t\, p_t(\cdot \mid S_{2,\varepsilon}) + m_t\, p_t(\cdot \mid \mathcal{X} \setminus \mathcal{S}_\varepsilon), \qquad a_t + b_t + m_t = 1.$$

Since $m_t \to 0$, for any weak limit point $p_\infty$ along a subsequence $p_{t_k} \Rightarrow p_\infty$ we have

$$p_\infty(\mathcal{X} \setminus \mathcal{S}_\varepsilon) \le \liminf_{k \to \infty} p_{t_k}(\mathcal{X} \setminus \mathcal{S}_\varepsilon) = \liminf_{k \to \infty} m_{t_k} = 0, \tag{41}$$

so $p_\infty$ is supported on $\mathcal{S}_\varepsilon$. Define $a_\infty := p_\infty(S_{1,\varepsilon})$ and the conditionals $p_{\infty,1} := p_\infty(\cdot \mid S_{1,\varepsilon}), p_{\infty,2} := p_\infty(\cdot \mid S_{2,\varepsilon})$. Then the identity

$$p_\infty = a_\infty\, p_{\infty,1} + (1 - a_\infty)\, p_{\infty,2} \tag{42}$$

holds by disjoint support and conditioning. If in addition $a_t \to a_\infty \in (0,1)$, the leakage interval bound follows directly from Lemma B.5. $\qquad\square$

**Proposition B.3** (Plateau-basin case: invariant conditionals and explicit mixture form)**.** *Fix $\varepsilon > 0$ and suppose $S_{1,\varepsilon} \cap S_{2,\varepsilon} = \emptyset$ and*

$$p_0(S_{1,\varepsilon}) > 0, \qquad p_0(S_{2,\varepsilon}) > 0.$$

*Work in the $K \to \infty$ regime of Lemma B.2. Assume the rewards are piecewise constant on the two basins: there exist constants $U_1, U_2 \in \mathbb{R}$ and gaps $\Delta_1, \Delta_2 > 0$ such that*

$$r_1(x) = U_1, \;\; r_2(x) = U_2 - \Delta_2 \quad \text{for all } x \in S_{1,\varepsilon}, \tag{43}$$
$$r_1(x) = U_1 - \Delta_1, \;\; r_2(x) = U_2 \quad \text{for all } x \in S_{2,\varepsilon}. \tag{44}$$

*Assume also the decay condition of Lemma B.4 holds for $\mathcal{S}_\varepsilon$, so that*

$$m_t := p_t(\mathcal{X} \setminus \mathcal{S}_\varepsilon) \to 0.$$

*Then for all $t \ge 0$, the basin-conditional distributions are invariant:*

$$p_t(\cdot \mid S_{1,\varepsilon}) \equiv p_0(\cdot \mid S_{1,\varepsilon}) =: p_{S_{1,\varepsilon}}, \qquad p_t(\cdot \mid S_{2,\varepsilon}) \equiv p_0(\cdot \mid S_{2,\varepsilon}) =: p_{S_{2,\varepsilon}}. \tag{45}$$

*Consequently, along any subsequence $t_k \to \infty$ such that $a_{t_k} \to a_\infty$ where*

$$a_t := p_t(S_{1,\varepsilon}), \qquad b_t := p_t(S_{2,\varepsilon}),$$

*we have the weak limit*

$$p_{t_k} \Rightarrow p_\infty = a_\infty\, p_{S_{1,\varepsilon}} + (1 - a_\infty)\, p_{S_{2,\varepsilon}}. \tag{46}$$

*Moreover, $a_\infty$ must lie in the leakage-controlled interval of Lemma B.5 with $\kappa_1 = e^{-\Delta_1}$ and $\kappa_2 = e^{-\Delta_2}$ (up to the $\varepsilon$-adjustment if you use it there). In particular, as $\Delta_1, \Delta_2 \to \infty$, any such limit weight satisfies $a_\infty \to q$, and*

$$p_\infty \to q\, p_{S_{1,\varepsilon}} + (1 - q)\, p_{S_{2,\varepsilon}}.$$

*Proof.* In the $K \to \infty$ regime,

$$p_{t+1}(x) = p_t(x)\Big(q\,\frac{e^{r_1(x)}}{\mu_1(t)} + (1-q)\,\frac{e^{r_2(x)}}{\mu_2(t)}\Big).$$

Under the plateau assumptions, the multiplier is constant on each basin: there exist scalars $M_1(t), M_2(t) > 0$ such that $p_{t+1}(x) = M_1(t)p_t(x)$ for all $x \in S_{1,\varepsilon}$ and $p_{t+1}(x) = M_2(t)p_t(x)$ for all $x \in S_{2,\varepsilon}$. Therefore, for $x \in S_{1,\varepsilon}$,

$$p_{t+1}(x \mid S_{1,\varepsilon}) = \frac{p_{t+1}(x)}{\int_{S_{1,\varepsilon}} p_{t+1}} = \frac{M_1(t)p_t(x)}{M_1(t)\int_{S_{1,\varepsilon}} p_t} = p_t(x \mid S_{1,\varepsilon}), \tag{47}$$

and similarly on $S_{2,\varepsilon}$. This proves conditional invariance (45).

Now decompose

$$p_t = a_t\,p_{S_{1,\varepsilon}} + (1 - a_t - m_t)\,p_{S_{2,\varepsilon}} + m_t\,p_t(\cdot \mid \mathcal{X} \setminus \mathcal{S}_\varepsilon), \tag{48}$$

using conditional invariance, and $m_t \to 0$ by Lemma B.4. Along any subsequence with $a_{t_k} \to a_\infty$ the last term vanishes in total mass, hence

$$p_{t_k} \Rightarrow a_\infty\,p_{S_{1,\varepsilon}} + (1 - a_\infty)\,p_{S_{2,\varepsilon}}.$$

Finally, Lemma B.5 applies with $\kappa_1 = e^{-\Delta_1}$ and $\kappa_2 = e^{-\Delta_2}$ (up to your $\varepsilon$-adjustment), forcing $a_\infty \to q$ as $\Delta_1, \Delta_2 \to \infty$ and yielding the stated limit. $\qquad\square$

**Theorem B.4** (Limit of expected rewards on two-basin subsequential limits)**.** *Assume the conditions of Theorem B.2 (Two-basin subsequential limit) for a fixed $\varepsilon > 0$, so that along some subsequence $t_k \to \infty$ we have $p_{t_k} \Rightarrow p_\infty$ and*

$$p_\infty = a_\infty\,p_{\infty,1} + (1 - a_\infty)\,p_{\infty,2}, \qquad p_{\infty,1} = p_\infty(\cdot \mid S_{1,\varepsilon}), \quad p_{\infty,2} = p_\infty(\cdot \mid S_{2,\varepsilon}).$$

*Assume in addition that $r_1$ and $r_2$ are bounded and continuous on $\mathcal{X}$. Then for each $i \in \{1, 2\}$,*

$$\mathbb{E}_{p_{t_k}}[r_i(x)] \longrightarrow \mathbb{E}_{p_\infty}[r_i(x)] = a_\infty\,\mathbb{E}_{p_{\infty,1}}[r_i(x)] + (1 - a_\infty)\,\mathbb{E}_{p_{\infty,2}}[r_i(x)]. \tag{49}$$

*Moreover, by definition of the $\varepsilon$-optimal regions,*

$$\mathbb{E}_{p_{\infty,1}}[r_1(x)] \in [r_1^* - \varepsilon,\, r_1^*], \qquad \mathbb{E}_{p_{\infty,2}}[r_2(x)] \in [r_2^* - \varepsilon,\, r_2^*]. \tag{50}$$

*If the basin-separation conditions of Lemma B.5 hold (with gaps $\Delta_1, \Delta_2$), then also*

$$\mathbb{E}_{p_{\infty,2}}[r_1(x)] \le r_1^* - \Delta_1 + \varepsilon, \qquad \mathbb{E}_{p_{\infty,1}}[r_2(x)] \le r_2^* - \Delta_2 + \varepsilon. \tag{51}$$

*Proof.* Since $r_i$ is bounded and continuous and $p_{t_k} \Rightarrow p_\infty$, we have $\mathbb{E}_{p_{t_k}}[r_i] \to \mathbb{E}_{p_\infty}[r_i]$ by weak convergence. With $p_\infty = a_\infty p_{\infty,1} + (1 - a_\infty)p_{\infty,2}$, linearity of expectation gives

$$\mathbb{E}_{p_\infty}[r_i] = a_\infty\mathbb{E}_{p_{\infty,1}}[r_i] + (1 - a_\infty)\mathbb{E}_{p_{\infty,2}}[r_i], \tag{52}$$

which is (49). The bounds (50) follow from $p_{\infty,1}$ being supported on $S_{1,\varepsilon}$ and $p_{\infty,2}$ on $S_{2,\varepsilon}$. The additional bounds (51) follow directly from the separation inequalities holding pointwise on the opposite basin. $\qquad\square$

**Corollary B.5** (Expected reward limits in the hard-separation plateau case)**.** *Under the assumptions of Proposition B.3, we have*

$$p_t \Rightarrow q\,p_{S_1} + (1 - q)\,p_{S_2}.$$

*If in addition $r_1$ and $r_2$ are bounded and continuous, then*

$$\mathbb{E}_{p_t}[r_1(x)] \longrightarrow q\,\mathbb{E}_{p_{S_1}}[r_1(x)] + (1 - q)\,\mathbb{E}_{p_{S_2}}[r_1(x)] = q\,r_1^* + (1 - q)\,\mathbb{E}_{p_{S_2}}[r_1(x)], \tag{53}$$

$$\mathbb{E}_{p_t}[r_2(x)] \longrightarrow q\,\mathbb{E}_{p_{S_1}}[r_2(x)] + (1 - q)\,\mathbb{E}_{p_{S_2}}[r_2(x)] = q\,\mathbb{E}_{p_{S_1}}[r_2(x)] + (1 - q)\,r_2^*. \tag{54}$$

*Proof.* Proposition B.3 gives $p_t \Rightarrow q\,p_{S_1} + (1 - q)\,p_{S_2}$. Since $r_i$ is bounded and continuous, $\mathbb{E}_{p_t}[r_i] \to \mathbb{E}_{qp_{S_1}+(1-q)p_{S_2}}[r_i]$. Linearity of expectation yields the mixture formula. In the hard-separation plateau case, $r_1 \equiv r_1^*$ on $S_1$ and $r_2 \equiv r_2^*$ on $S_2$, giving (53)–(54). $\qquad\square$

**Theorem B.6** (Inter-mode variance preservation under $\varepsilon$-basin limits). *Fix $\varepsilon > 0$ and suppose the $\varepsilon$-optimal regions $S_{1,\varepsilon}, S_{2,\varepsilon}$ are disjoint. Assume $r_1, r_2$ are bounded and continuous. Let $t_k \to \infty$ be a subsequence such that $p_{t_k} \Rightarrow p_\infty$ and*

$$p_\infty = a_\infty\, p_{\infty,1} + (1 - a_\infty)\, p_{\infty,2}, \qquad p_{\infty,1} = p_\infty(\cdot \mid S_{1,\varepsilon}), \ \ p_{\infty,2} = p_\infty(\cdot \mid S_{2,\varepsilon}),$$

*with $a_\infty \in (0,1)$ (so both conditionals are well-defined). Then for each $i \in \{1,2\}$,*

$$\lim_{k \to \infty} Var_{p_{t_k}}[r_i(x)] = Var_{p_\infty}[r_i(x)] = a_\infty Var_{p_{\infty,1}}[r_i(x)] + (1 - a_\infty)Var_{p_{\infty,2}}[r_i(x)] + a_\infty(1 - a_\infty)\big(\mu_{i,1} - \mu_{i,2}\big)^2, \tag{55}$$

*where $\mu_{i,1} := \mathbb{E}_{p_{\infty,1}}[r_i(x)]$ and $\mu_{i,2} := \mathbb{E}_{p_{\infty,2}}[r_i(x)]$.*

*In particular, if the basin-separation bounds hold in the form*

$$r_1(x) \le r_1^* - \Delta_1 + \varepsilon \text{ for all } x \in S_{2,\varepsilon}, \qquad r_2(x) \le r_2^* - \Delta_2 + \varepsilon \text{ for all } x \in S_{1,\varepsilon}, \tag{56}$$

*then*

$$Var_{p_\infty}[r_1(x)] \ \ge \ a_\infty(1 - a_\infty)\big(\Delta_1 - 2\varepsilon\big)_+^2, \qquad Var_{p_\infty}[r_2(x)] \ \ge \ a_\infty(1 - a_\infty)\big(\Delta_2 - 2\varepsilon\big)_+^2, \tag{57}$$

*where $(z)_+ := \max\{z, 0\}$. Hence if $\Delta_1 > 2\varepsilon$ and $\Delta_2 > 2\varepsilon$, both limiting variances are strictly positive.*

*Proof.* Because $r_i$ is bounded and continuous and $p_{t_k} \Rightarrow p_\infty$, we have $\mathbb{E}_{p_{t_k}}[r_i] \to \mathbb{E}_{p_\infty}[r_i]$ and $\mathbb{E}_{p_{t_k}}[r_i^2] \to \mathbb{E}_{p_\infty}[r_i^2]$ (Portmanteau theorem). Therefore

$$Var_{p_{t_k}}[r_i] = \mathbb{E}_{p_{t_k}}[r_i^2] - \mathbb{E}_{p_{t_k}}[r_i]^2 \ \longrightarrow \ \mathbb{E}_{p_\infty}[r_i^2] - \mathbb{E}_{p_\infty}[r_i]^2 = Var_{p_\infty}[r_i]. \tag{58}$$

For the identity (55), write $p_\infty = a_\infty p_{\infty,1} + (1 - a_\infty)p_{\infty,2}$ and denote $\mu := \mathbb{E}_{p_\infty}[r_i] = a_\infty \mu_{i,1} + (1 - a_\infty)\mu_{i,2}$. Then

$$\begin{aligned}
Var_{p_\infty}[r_i] &= \mathbb{E}_{p_\infty}\big[(r_i - \mu)^2\big] \\
&= a_\infty \mathbb{E}_{p_{\infty,1}}\big[(r_i - \mu)^2\big] + (1 - a_\infty)\mathbb{E}_{p_{\infty,2}}\big[(r_i - \mu)^2\big] \\
&= a_\infty Var_{p_{\infty,1}}[r_i] + (1 - a_\infty)Var_{p_{\infty,2}}[r_i] + a_\infty(1 - a_\infty)(\mu_{i,1} - \mu_{i,2})^2,
\end{aligned}$$

which is (55).

For the lower bound, note that by definition of $S_{1,\varepsilon}$ we have $\mu_{1,1} \ge r_1^* - \varepsilon$, and by (56) we have $\mu_{1,2} \le r_1^* - \Delta_1 + \varepsilon$. Thus $(\mu_{1,1} - \mu_{1,2}) \ge \Delta_1 - 2\varepsilon$. Plugging this into (55) and dropping the nonnegative within-basin variance terms yields (57) for $r_1$. The $r_2$ bound is identical. $\qquad\square$

**Theorem B.7** (Nash bargaining characterization on the two-basin mixture family). *Let $P_1, P_2$ be two fixed distributions supported on $S_{1,\varepsilon}$ and $S_{2,\varepsilon}$ respectively, and define the mixture family*

$$p_\alpha := \alpha P_1 + (1 - \alpha)P_2, \qquad \alpha \in [0,1].$$

*Let utilities be $u_i(p) := \mathbb{E}_p[r_i(x)]$, and define the disagreement points*

$$d_1 := u_1(P_2), \qquad d_2 := u_2(P_1).$$

*Assume the gains are strictly positive:*

$$u_1(P_1) > u_1(P_2), \qquad u_2(P_2) > u_2(P_1).$$

*Then the weighted Nash product*

$$\mathcal{N}(\alpha) := \big(u_1(p_\alpha) - d_1\big)^q \big(u_2(p_\alpha) - d_2\big)^{1-q} \tag{59}$$

*is uniquely maximized over $\alpha \in [0,1]$ at $\alpha^\star = q$. Equivalently, $p_q$ is the unique weighted Nash bargaining solution within the line segment $\{p_\alpha : \alpha \in [0,1]\}$.*

*Proof.* By linearity of expectation,

$$u_1(p_\alpha) = \alpha u_1(P_1) + (1-\alpha)u_1(P_2) = d_1 + \alpha\Delta_1, \qquad \Delta_1 := u_1(P_1) - u_1(P_2) > 0, \tag{60}$$

and similarly

$$u_2(p_\alpha) = \alpha u_2(P_1) + (1-\alpha)u_2(P_2) = d_2 + (1-\alpha)\Delta_2, \qquad \Delta_2 := u_2(P_2) - u_2(P_1) > 0. \tag{61}$$

Hence

$$\mathcal{N}(\alpha) = (\alpha\Delta_1)^q\big((1-\alpha)\Delta_2\big)^{1-q} = \Delta_1^q \Delta_2^{1-q} \, \alpha^q (1-\alpha)^{1-q}. \tag{62}$$

The constant prefactor does not affect the maximizer, so it suffices to maximize $f(\alpha) := \alpha^q(1-\alpha)^{1-q}$ on $[0,1]$. For $\alpha \in (0,1)$,

$$\frac{d}{d\alpha} \log f(\alpha) = \frac{q}{\alpha} - \frac{1-q}{1-\alpha}.$$

Setting this to zero yields $\alpha = q$, and strict concavity of $\log f$ on $(0,1)$ gives uniqueness. $\square$

**Corollary B.8** (Nash bargaining interpretation of the hard-separation plateau limit)**.** *Under Proposition B.3, let $P_1 := p_{S_{1,\varepsilon}}$ and $P_2 := p_{S_{2,\varepsilon}}$. If $u_1(P_1) > u_1(P_2)$ and $u_2(P_2) > u_2(P_1)$, then the limiting distribution*

$$p_\infty = q \, p_{S_{1,\varepsilon}} + (1-q) \, p_{S_{2,\varepsilon}}$$

*is exactly the unique weighted Nash bargaining solution over mixtures $\{p_\alpha\}$ with disagreement points $d_1 = u_1(P_2)$ and $d_2 = u_2(P_1)$.*

*Proof.* This is Theorem B.7 applied to $P_1 = p_{S_{1,\varepsilon}}$ and $P_2 = p_{S_{2,\varepsilon}}$, together with Proposition B.3, which identifies $p_\infty$ with $p_q$. $\square$

### B.4. Extension to Multi Reward Functions

We generalize the two-reward setting to a finite family $\{r_1, \ldots, r_M\}$ with sampling probabilities $q_i > 0$ and $\sum_{i=1}^M q_i = 1$. Define the $\varepsilon$-optimal regions

$$S_{i,\varepsilon} := \{x \in \mathcal{X} : \ r_i(x) \geq r_i^* - \varepsilon\}, \qquad \mathcal{S}_\varepsilon := \bigcup_{i=1}^M S_{i,\varepsilon}, \qquad m_t := p_t(\mathcal{X} \setminus \mathcal{S}_\varepsilon).$$

For clarity we state the multi-basin results under the disjointness assumption $S_{i,\varepsilon} \cap S_{j,\varepsilon} = \emptyset$ for $i \neq j$. If the $S_{i,\varepsilon}$ overlap, one can replace them by any measurable partition $\{\widetilde{S}_{i,\varepsilon}\}_{i=1}^M$ with $\widetilde{S}_{i,\varepsilon} \subseteq S_{i,\varepsilon}$ and $\bigcup_i \widetilde{S}_{i,\varepsilon} = \mathcal{S}_\varepsilon$; all arguments below apply verbatim to the partitioned sets.

### B.5. Update and large-$K$ limit

**Lemma B.7** (Pluralistic curation update for $M$ rewards)**.** *At step $t$, draw a reward $R \in \{r_1, \ldots, r_M\}$ with $\Pr(R = r_i) = q_i$. Then the finite-$K$ update satisfies*

$$p_{t+1}(x) = p_t(x) \sum_{i=1}^M q_i \, H_{p_t}^{K,r_i}(x),$$

*where*

$$H_{p_t}^{K,r_i}(x) := \mathbb{E}_{y_{1:K-1} \sim p_t}\left[ \frac{Ke^{r_i(x)}}{e^{r_i(x)} + \sum_{j=1}^{K-1} e^{r_i(y_j)}} \right].$$

*Proof.* The proof is identical to Lemma B.1, replacing the two-point mixture $q(\cdot) + (1-q)(\cdot)$ by the $M$-point mixture $\sum_{i=1}^M q_i(\cdot)$. $\square$

**Lemma B.8** (Large-$K$ exponential tilts for $M$ rewards). *Assume each $r_i$ is bounded and measurable. In the $K \to \infty$ regime,*

$$H_{p_t}^{K,r_i}(x) \to \frac{e^{r_i(x)}}{\mu_i(t)}, \qquad \mu_i(t) := \mathbb{E}_{p_t}[e^{r_i(x)}],$$

*and therefore the update becomes*

$$p_{t+1}(x) = p_t(x) \sum_{i=1}^{M} q_i \frac{e^{r_i(x)}}{\mu_i(t)}.$$

*Proof.* Apply Lemma B.2 to each fixed $r_i$ and substitute into Lemma B.7. $\square$

**B.6. Mass concentrates on $\cup_i S_{i,\varepsilon}$**

**Lemma B.9** (Decay outside $\mathcal{S}_\varepsilon$ for $M$ rewards). *Fix $\varepsilon > 0$ and work in the $K \to \infty$ regime of Lemma B.8. Assume bounded rewards $r_i \in [L_i, U_i]$ and an outside-gap: there exists $\delta_{c,\varepsilon} > 0$ such that for all $x \notin \mathcal{S}_\varepsilon$ and all $i$,*

$$r_i(x) \le U_i - \delta_{c,\varepsilon}.$$

*If*

$$\delta_{c,\varepsilon} > \log\Big( \sum_{i=1}^{M} q_i e^{U_i - L_i} \Big),$$

*then there exists $\rho_\varepsilon \in (0,1)$ such that*

$$m_{t+1} \le \rho_\varepsilon m_t \quad \text{for all } t, \qquad \text{hence} \qquad m_t \to 0.$$

*Proof.* For $x \notin \mathcal{S}_\varepsilon$, the gap gives $e^{r_i(x)} \le e^{U_i - \delta_{c,\varepsilon}}$. Also $\mu_i(t) = \mathbb{E}_{p_t}[e^{r_i}] \ge e^{L_i}$, hence

$$\frac{e^{r_i(x)}}{\mu_i(t)} \le e^{U_i - L_i - \delta_{c,\varepsilon}}. \tag{63}$$

Therefore the per-step multiplier satisfies

$$\sum_{i=1}^{M} q_i \frac{e^{r_i(x)}}{\mu_i(t)} \le e^{-\delta_{c,\varepsilon}} \sum_{i=1}^{M} q_i e^{U_i - L_i} =: \rho_\varepsilon. \tag{64}$$

By the stated condition, $\rho_\varepsilon < 1$. Integrating the update over $\mathcal{X} \setminus \mathcal{S}_\varepsilon$ yields $m_{t+1} \le \rho_\varepsilon m_t$. $\square$

**B.7. Leakage-controlled weights and the vanishing-leakage limit**

For disjoint basins, define basin masses

$$a_{i,t} := p_t(S_{i,\varepsilon}), \qquad \sum_{i=1}^{M} a_{i,t} = 1 - m_t.$$

**Assumption B.7** (Cross-basin separation for $M$ rewards). *Fix $\varepsilon > 0$ and assume the sets $S_{i,\varepsilon}$ are disjoint. For each ordered pair $i \ne j$, assume there exists a gap $\Delta_{i \leftarrow j} > 2\varepsilon$ such that*

$$x \in S_{j,\varepsilon} \implies r_i(x) \le r_i^* - \Delta_{i \leftarrow j} + \varepsilon.$$

*Define leakage factors*

$$\kappa_{i \leftarrow j} := \exp\big( -(\Delta_{i \leftarrow j} - 2\varepsilon) \big), \qquad \kappa_{\max} := \max_{i \ne j} \kappa_{i \leftarrow j}.$$

*Also assume an outside bound: there is $\delta_c > 0$ such that $x \notin \mathcal{S}_\varepsilon \Rightarrow r_i(x) \le r_i^* - \delta_c$ for all $i$.*

**Lemma B.10** ( tilt bound on basin masses)**.** *Under Assumption B.7 and in the $K \to \infty$ regime, for each $i$ there exists $C_\varepsilon > 0$ such that for all $t$,*

$$p_t^{(i)}(S_{i,\varepsilon}) \geq \frac{a_{i,t}}{a_{i,t} + \sum_{j\neq i} \kappa_{i\leftarrow j} a_{j,t} + C_\varepsilon m_t}, \qquad p_t^{(i)}(S_{j,\varepsilon}) \leq \frac{\kappa_{i\leftarrow j} a_{j,t}}{a_{i,t} + \kappa_{i\leftarrow j} a_{j,t}} \quad (j \neq i), \tag{65}$$

*where $p_t^{(i)}(dx) := \frac{e^{r_i(x)}}{\mu_i(t)} p_t(dx)$.*

*Proof.* Fix $i$. On $S_{i,\varepsilon}$, $r_i \geq r_i^* - \varepsilon$. On $S_{j,\varepsilon}$, $j \neq i$, Assumption B.7 gives $r_i \leq r_i^* - \Delta_{i\leftarrow j} + \varepsilon$, hence $e^{r_i} \leq e^{r_i^*-\varepsilon} \kappa_{i\leftarrow j}$. On $\mathcal{X} \setminus \mathcal{S}_\varepsilon$, the outside bound gives $e^{r_i} \leq e^{r_i^*-\varepsilon} e^{-(\delta_c-\varepsilon)}$. Factoring out $e^{r_i^*-\varepsilon}$ from the numerator and denominator of $p_t^{(i)}(S_{i,\varepsilon}) = \frac{\int_{S_{i,\varepsilon}} e^{r_i} p_t}{\int_{\mathcal{X}} e^{r_i} p_t}$ yields the first inequality for a suitable constant $C_\varepsilon$.

For $j \neq i$, similarly $\int_{S_{j,\varepsilon}} e^{r_i} p_t \leq \kappa_{i\leftarrow j} a_{j,t} e^{r_i^*-\varepsilon}$ and $\int_{S_{i,\varepsilon}} e^{r_i} p_t \geq a_{i,t} e^{r_i^*-\varepsilon}$, so

$$p_t^{(i)}(S_{j,\varepsilon}) = \frac{\int_{S_{j,\varepsilon}} e^{r_i} p_t}{\int_{\mathcal{X}} e^{r_i} p_t} \leq \frac{\int_{S_{j,\varepsilon}} e^{r_i} p_t}{\int_{S_{i,\varepsilon}} e^{r_i} p_t + \int_{S_{j,\varepsilon}} e^{r_i} p_t} \leq \frac{\kappa_{i\leftarrow j} a_{j,t}}{a_{i,t} + \kappa_{i\leftarrow j} a_{j,t}}. \tag{66}$$

$\square$

**Lemma B.11** (Limit weights are $O(\kappa_{\max})$-close to $(q_i)$)**.** *Under Assumption B.7, suppose $m_t \to 0$ and there exists $\eta > 0$ and $t_0$ such that $a_{i,t} \geq \eta$ for all $i$ and all $t \geq t_0$. Then there is a constant $C(\eta) > 0$ such that for all $t \geq t_0$ and all $i$,*

$$\left| a_{i,t+1} - q_i \right| \leq C(\eta)\, \kappa_{\max} + C(\eta)\, m_t.$$

*Consequently, along any subsequence $t_k \to \infty$ with $p_{t_k} \Rightarrow p_\infty$ and $m_{t_k} \to 0$, the limit weights $a_{i,\infty} := p_\infty(S_{i,\varepsilon})$ satisfy*

$$\left| a_{i,\infty} - q_i \right| \leq C(\eta)\, \kappa_{\max}, \qquad \text{so if } \kappa_{\max} \to 0 \text{ then } a_{i,\infty} \to q_i \;\; \forall i.$$

*Proof.* Write the update in tilt form:

$$a_{i,t+1} = p_{t+1}(S_{i,\varepsilon}) = \sum_{\ell=1}^{M} q_\ell\, p_t^{(\ell)}(S_{i,\varepsilon}).$$

*Lower bound.* by Lemma B.10,

$$p_t^{(i)}(S_{i,\varepsilon}) \geq \frac{a_{i,t}}{a_{i,t} + \sum_{j\neq i} \kappa_{i\leftarrow j} a_{j,t} + C_\varepsilon m_t} \geq \frac{a_{i,t}}{a_{i,t} + \kappa_{\max}(1 - a_{i,t}) + C_\varepsilon m_t}.$$

Using $a_{i,t} \geq \eta$ and $m_t \leq 1$, the identity $\frac{1}{1+u} \geq 1 - u$ gives

$$p_t^{(i)}(S_{i,\varepsilon}) \geq 1 - \frac{\kappa_{\max}}{\eta} - \frac{C_\varepsilon}{\eta} m_t.$$

Hence

$$a_{i,t+1} \geq q_i\left(1 - \frac{\kappa_{\max}}{\eta} - \frac{C_\varepsilon}{\eta} m_t\right).$$

*Upper bound.* for $\ell \neq i$, Lemma B.10 gives

$$p_t^{(\ell)}(S_{i,\varepsilon}) \leq \frac{\kappa_{\ell\leftarrow i} a_{i,t}}{a_{\ell,t} + \kappa_{\ell\leftarrow i} a_{i,t}} \leq \frac{\kappa_{\max} a_{i,t}}{a_{\ell,t}} \leq \frac{\kappa_{\max}}{\eta}.$$

Also $p_t^{(i)}(S_{i,\varepsilon}) \leq 1$. Therefore

$$a_{i,t+1} \leq q_i + \sum_{\ell \neq i} q_\ell \frac{\kappa_{\max}}{\eta} = q_i + \frac{(1 - q_i)}{\eta} \kappa_{\max} \leq q_i + \frac{1}{\eta} \kappa_{\max}.$$

Combining the two bounds yields the claimed inequality with $C(\eta)$ absorbing constants.

For the subsequential limit statement, take $t = t_k$ and pass to the limit using $m_{t_k} \to 0$. $\square$

**Discussion.** Lemma B.9 proves concentration on $\bigcup_i S_{i,\varepsilon}$ under the same outside-gap mechanism as the two-reward case. Lemma B.11 shows that, once each basin retains nontrivial mass, cross-basin leakage controls the deviation of the limiting weights from $(q_i)$, and in the vanishing-leakage regime the limit weights converge to $(q_i)$. In practice, the nontrivial-mass condition is ensured by the same fixed-point comparison argument used in the two-basin uniform non-collapse lemma, applied basin-by-basin using the $r_i$-tilt and the fact that $q_i > 0$.