# OpenReview forum: "Curated Synthetic Data Doesn’t Have to Collapse: A Theoretical Study of Generative Retraining with Pluralistic Preferences"
_ICML.cc/2026/Conference — ICML 2026 regular_

### Official Review · Reviewer_FoWN · 2026-02-17

**Soundness:** 2
**Presentation:** 3
**Significance:** 2
**Originality:** 3
**Overall Recommendation:** 4
**Confidence:** 3

**Summary:**

This paper focuses on recursively retraining generative models using curated synthetic outputs and derives conditions to prevent model collapse. Specifically, once we have multiple rewards and each of the rewards is somewhat unique compared to others, the retrained models will not collapse to overoptimize a single reward, but will produce an output distribution with mass concentrating on different rewards. Meanwhile, the output distribution can preserve the variance of rewards.

**Compliance With Llm Reviewing Policy:**

Affirmed.

**Final Justification:**

Overall the rebuttal clarifies the problems. I will improve my score to 4.

**Key Questions For Authors:**

Could you discuss/experiment with what will happen when the assumptions are not satisfied or when the models are larger?

**Limitations:**

yes

**Strengths And Weaknesses:**

> Strengths

- Soundness: I think the paper is sound and the proposed results/claims seem to be correct and make sense in my perspective.
- Presentation: Most parts are clear and well written
- Significance: I think model collapse and training using synthetic data are important.
- Originality: The paper proposes new theory that model collapse can be prevented .

> Weaknesses

- Significance: The theoretical results seem to be not surprising. Specifically, once curation randomly alternates between two unique rewards, it is expected that the retraining loop allocates mass to both high-reward regions rather than collapsing.

- Soundness:

1. The paper incorporates a number of ideal assumptions: 3 assumptions (Assumption 2.1 (i) - (iii)) are needed to ensure rewards are sufficiently separated, and the derivation of the limiting distribution seems to be more meaningful when $\Delta$ is large.

2. Meanwhile, the MLE update seems to be ideal and overly simplified, and the empirical analysis does not include more advanced large generative models.

---

> ### Author Rebuttal · Authors · 2026-03-31
>
> We thank the reviewer for the thoughtful feedback and address the main concerns below.
>
> ---
> ### 1. On Significance
> Our contribution is not the intuition that alternating rewards may preserve multiple modes. It is a quantitative characterization of when pluralistic retraining avoids collapse, when it still collapses, and what distribution it converges to. Ferbach et al. (2024) show collapse under single-reward curation. Our results show that pluralism does not automatically prevent collapse: it succeeds only under measurable separation and leakage conditions.
>
> Specifically, we show:
> - (i) exponential concentration onto the union of optimal basins (no compromise mode),
> - (ii) a leakage-controlled limiting mixture whose weight converges to \(q\) as separation grows (Lemma 3.4, Table 2),
> - (iii) a **phase transition** where pluralistic retraining still collapses below a separation threshold (Figure 3, E2),
> - (iv) a characterization of the limit as the solution to a **weighted Nash bargaining problem** with weight exactly \(q\) (Theorem 3.7).
>
> ---
> ### 2. On the assumptions
> The assumptions are **sufficient to obtain guarantees**, not necessary. The appendix studies these violations and shows the same qualitative picture.
>
> - **Bounded rewards (Assumption 2.1(i)).**
>   Used only to ensure the tilt-and-renormalize update is well-defined. All previous and new experiments use **unbounded rewards** (GMM rewards, classifier logits, GPT-2 length rewards) with no clipping, and we observe the same qualitative behavior. This is a theoretical regularity condition.
> - **Disjoint basins (Assumption 2.1(ii)).**
> E2 and Appendix C.9 directly probes this (Figure 3). As separation decreases, the model transitions smoothly from a clean mixture (large separation) to collapse (overlapping basins). The GPT-2 experiment (E4, Figure 5) shows the same phenomenon with $d = |T_A - T_B|$.
> - **Cross-reward leakage gaps (Assumption 2.1(iii)).**
>   Quantitatively validated: empirical basin mass tracks the theoretical leakage bounds (Appendix Fig. 5, Table 2), and outside-basin mass decays exponentially as predicted (Appendix Fig. 6, Table 3). The CIFAR-10 experiment further confirms this: low cross-class confusion yields preserved diversity.
>
> ---
> ### 3. Finite-\(K\) regime
> While the theory is derived in the large-\(K\) limit, results extend to finite K via Remark 3.1 with $O(\sqrt(log K/K))$ bounds. **Appendix Figure 4 directly validates this**: for $K\ge256$, the alignment and diversity trajectories are indistinguishable from the $K\to \infty$ regime. The CIFAR-10 experiment uses finite K-BT selection (from 50K candidates) and it follows the same trend as predicted by the theory.
>
> ---
> ### 4. Realism: new rebuttal experiments
> To address the realism concern, we added two new experiments that are closer to current RLHF practice and with larger generative models. Figures are available in this anonymized Google Drive:
>
> - R1: https://drive.google.com/file/d/1wAxUva-Oae5JQrAQcGq6uMjEIzUJaJmT
> - R2: https://drive.google.com/file/d/1uU7iKGBFgBKkTJR2CSHXs3_rNG4no93B
>
> **(1) Learned reward models (vision).**
> We replace the classifier-style rewards with three off-the-shelf neural reward models: PickScore, ImageReward, and LAION Aesthetic Score. Across seven curation strategies, single-reward retraining gives the worst long-run quality and diversity, pairwise mixtures are intermediate, and the uniform 3-way mixture performs best, with FID near 44 versus 60 to 72 for single-reward runs and the highest feature-space variance.
>
> Moreover, a reward’s variance is preserved mainly when that reward participates in the curation mixture, which matches the leakage mechanism behind Lemma 3.4. This shows the effect is not tied to class-conditional rewards.
>
> **(2) RLHF-style iterative pipeline (Qwen-2.5-1.5B).**
> We also ran a finite-sample iterative post-training pipeline on Qwen-2.5-1.5B, using generated outputs, BT curation under competing helpfulness (ArmoRM-Llama3-8B-v0.1) and safety (Llama-Guard-3-1B) rewards, and then fine-tuning via SFT for 30 rounds.
>
> Here, the exact-MLE abstraction is removed entirely, yet the same qualitative pattern remains: optimizing only helpfulness or only safety over-specializes the model, while balanced pluralistic curation maintains both rewards at competitive levels, preserves lexical diversity (Distinct-2: 0.60 versus 0.47 to 0.49), and substantially reduces held-out perplexity (about 6 versus 8 to 9). This shows that the non-collapse mechanism survives in a realistic iterative alignment pipeline on a modern pretrained LLM.
>
> ---
> Taken together, the theory and the new rebuttal experiments address the main concern. The effect is not tied to a toy setup: with learned vision rewards and in an iterative RLHF-style LLM pipeline, pluralistic curation again preserves diversity and improves the alignment tradeoff relative to single-reward retraining. We hope this makes the contribution and practical relevance clearer.

---

> > ### Author Rebuttal · Reviewer_FoWN · 2026-03-31
> >
> > Thanks for the clarification. I will improve my score to 4.

---

### Official Review · Reviewer_o8gx · 2026-03-10

**Soundness:** 3
**Presentation:** 2
**Significance:** 2
**Originality:** 3
**Overall Recommendation:** 4
**Confidence:** 4

**Summary:**

This paper theoretically studies whether Curated Synthetic Data can mitigate mode collapse and help model converge to a stable distribution with high reward. The key technique is using reward as a weighting term for iterative update. The authors verify the effectiveness of the theoretical results on GMM, CIFAR10 dataset, and text generation.

**Compliance With Llm Reviewing Policy:**

Affirmed.

**Final Justification:**

I read the responses and am satisfied.

**Key Questions For Authors:**

In the experiments on CIFAR10 dataset, the reward function is too naïve. It is a class conditional probability which will not used for RLHF. Could the author consider using real reward function like Pickscore, Imagereward, and Aesthetic Score.
In the experiments on text generation, we care not only the entropy of the generated text, but the perplexity. Could the authors provide the perplexity in the main text?

**Limitations:**

yes

**Strengths And Weaknesses:**

Strengths:
The theoretical setup and conclusion are important and novel.
Besides theoretical results, the authors provide experiments on diverse modality.

Weaknesses:
Main concern: The paper lacks the description of related work for mitigating mode collapse of synthetic data in the main paper. Without this comparison, it is hard to claim the contribution of this paper. Also, no comparison with related work in experiments, and the fid on CIFAR10 is far from the state-of-the-art.

The paper does not present well. The reader might have several questions go through the paper.
In Lemma 3.1, the probability q is introduced without further explanation. It seems to be a very important hyperparameter that affects each iteration and all of the theoretical results. The reader has no idea how q is chosen can benefit training. Besides the ablation study of q summarized in Table 1, are there any high-level takeaway or conclusions?
In Figure 4, what is the ratio 0.0 to 0.5? Is it q?
Change Opacity for the top plot of Figure 3 since the purple line almost disappear in the last three subplots.

---

> ### Author Rebuttal · Authors · 2026-03-31
>
> We thank the reviewer. We directly addressed the two main requests:
> - (i) replacing classifier rewards with real learned reward models, and
> - (ii) reporting perplexity in the text setting.
>
> ---
> ### 1. CIFAR-10: real reward functions
> We ran new experiments replacing classifier rewards with three learned reward models suggested by the reviewer: PickScore (P), ImageReward (I), and Aesthetic Score (A). We evaluate:
> - **Single-reward baselines:** at each iteration, curation is performed using only one reward (P, I, or A), corresponding to the standard single-objective retraining setup.
> - **Pairwise mixtures:** at each iteration, one of two rewards is selected with equal probability (\(q = 0.5\)) and used for curation (P+I, I+A, A+P).
> - **3-way mixture:** at each iteration, one of the three rewards is selected uniformly at random (\(q = 1/3\)) for curation.
> All settings are run for 25 retraining iterations on the same OT-CFM flow, and evaluating all single-reward baselines, pairwise mixtures, and a 3-way mixture over 25 iterations (Figure R1).
>
> Single-reward FID degrades to 60–72, pairwise mixtures stabilize at 52–58, and the 3-way mixture holds near 44. Feature variance collapses under single rewards (0.15–0.25) but is sustained under pluralism (0.38).
>
> Furthermore, a reward’s variance is preserved only if it participates in the mixture.  For example, P+I collapses Aesthetic variance to ~0.05 (matching single-reward), while mixtures including Aesthetic preserve it. This matches the cross-basin leakage mechanism (Lemma 3.4) using black-box neural rewards.
>
> ---
> ### 2. Text experiment: perplexity (RLHF-style pipeline)
> We run 30 rounds of iterative fine-tuning on Qwen-2.5-1.5B using BT-curated outputs under competing helpfulness (ArmoRM-Llama3-8B-v0.1) and safety (Llama-Guard-3-1B) rewards (Figure R2). This mirrors practical RLHF pipelines (generation → selection → fine-tuning).
>
> Single-reward curation over-optimizes one objective, while pluralistic curation maintains both. Perplexity rises to 8–9 under single rewards but stays near 6 under pluralism (33\% reduction), and diversity (Distinct-2) is preserved (0.60 vs. 0.47–0.49).
>
> ---
> ### 3. CIFAR-10 FID
> We follow Ferbach et al. (2024) to isolate the curation effect. Two points on the absolute FID numbers:
> - (i) These retraining experiments are computationally expensive, 25 rounds of OT-CFM retraining over 50k samples per round, so we made some cuts to keep them tractable, which inflates FID relative to SOTA generation pipelines.
> - (ii) There is no real data in this loop. Models retrain purely on their own synthetic outputs. FID here reflects self-consuming collapse dynamics, and not generation quality.
>
> Our claim is about preventing mode collapse, not achieving SOTA sample quality. What matters is the relative FID: pluralistic curation drives it from 100 (single reward) to 35.9 (five preferences) within the exact same self-consuming setup.
>
> ---
> ### 4. Related work and baselines
> Related work is in Appendix A due to page limits; we will move it to the main paper in the extra page of the camera-ready version.
>
> On experimental comparisons: There is no prior method that studies pluralistic curation in recursive retraining. The appropriate baseline is therefore the single-reward case (q=0), which corresponds to Ferbach et al. and is already reported in Table 1.
>
> Our goal is not to propose a new method for improving diversity, but to provide a theoretical analysis of multi-preference curation in the recursive retraining regime. Methods that target diversity at the model or architectural level are therefore orthogonal, and comparing to such methods would confound the effect we aim to isolate.
>
> ---
> ### 5. Parameter $q$
> We agree that $q$ requires clearer explanation. We will revise Section 2.1 to make this explicit.
> $q$ is the probability of selecting each reward during curation. In the two-reward case, $r_1$​ is used with probability $q$ and $r_2$​ with probability $1-q$. The theory shows:
> - (i) the limiting mixture weight approaches $q$,
> - (ii) diversity is maximized near balanced $q$,
> - (iii) the limit corresponds to a weighted Nash bargaining solution with weight $q$.
>
> $q$ is not a hyperparameter to be tuned, but a **design choice encoding the relative importance of preferences** in the retraining loop. It controls how often each preference is applied during retraining, and this same weighting is what the model settles on in the long run.
>
> ---
> ### 6. Presentation
> We will clarify $q$ in Section 2.1, relabel Figure 4, and fix Figure 3 opacity.
>
> ---
> These additions address the requests for realism, evaluation, and presentation. We hope this clarifies the contribution and supports a more positive assessment.
>
> Figures are available in this anonymized Google Drive:
>
> - R1: https://drive.google.com/file/d/1wAxUva-Oae5JQrAQcGq6uMjEIzUJaJmT
> - R2: https://drive.google.com/file/d/1uU7iKGBFgBKkTJR2CSHXs3_rNG4no93B

---

> > ### Author Rebuttal · Reviewer_o8gx · 2026-04-01
> >
> > I've raised my score.

---

### Official Review · Reviewer_SXwi · 2026-03-11

**Soundness:** 3
**Presentation:** 3
**Significance:** 3
**Originality:** 3
**Overall Recommendation:** 4
**Confidence:** 3

**Summary:**

This paper looks at an extension of the setting where we repeatedly train a generative model, perform curation/selection, then retrain the model and repeat this for a certain number of times. Several prior works have shown that in the case where the curation is based on a single reward function, there can be collapse. In this work, the authors look at the setting where we have more than one reward. Under several "separation" assumptions, they show that collapse does not necesarily happen: the multiple reward functions can preserve diversity and yield a stable mixture distribution rather than single-mode collapse. The paper also contains experiments with synthetic data (gaussian mixture), image and text.

**Compliance With Llm Reviewing Policy:**

Affirmed.

**Final Justification:**

The rebuttal addressed my points, I keep my positive score.

**Key Questions For Authors:**

See strengths and weaknesses section for comments

**Limitations:**

Some limitations appear through the setup and assumptions. The theory is proved only under certain conditions and relies on assumptions such as nontrivial initialization on both basins, disjoint near-optimal basins, etc. The main theory is also in a large-K setup with exact maximum likelihood regime.

**Strengths And Weaknesses:**

**Soundness:** The main contribution of this paper is theoretical, and the results seem to be correct (but I did not check all the proofs). That said, there are some places where extra assumptions are needed. For example, in the proof to Theorem B.2 in the appendix, to get equation 41, a sufficient assumption is to assume that X \ S_epsilon is open, which is obtainable if the reward functions are continuous (this is a minor point). A question I have is about the main assumptions needed to obtain the results (exact MLE retraining, bounded rewards, disjoint/near-separated basins, and in some places plateau or hard-separation structure) and if they are realistic. It would be informative to have metrics to quantify to what extend they hold in real world setups (not just based on proxies).

**Presentation:** The paper is pretty well written and easy to follow. I noted though that some assumptions needed to prove a few theoretical results only appear in the appendix and not in the main text. For example in Lemma 3.3, there is an assumption which appears in Equation 18 only in the appendix.

**Significance:** I believe it is high, given that there has been a big line of works showing collapse in the single reward setting, which is not the most realistic one in my opinion. That said, the demonstrated practical impact is still a bit narrow, with experiments limited to controlled rewards and modest proxies. But I think this is okay given that the main contribution is theoretical.

**Originality:** The originality lies in the multiple-reward curation setup and the theory behind it showing that diversity can be preserved. It would be nevertheless interesting to contrast this work with others that consider muliples rewards, such as in Falahati et al. (which is cited by the authors) and Zhao et al [1] whose setup also considers heterogenity in rewards (with an additive noise). The more general recursive synthetic retraining framework is not new (as the authors cite Ferbach et al.).

[1] Zhao, H., Fu, J., & Pham, T. (2025). Convergence and Stability Analysis of Self-Consuming Generative Models with Heterogeneous Human Curation. _arXiv preprint arXiv:2511.09002_.

---

> ### Author Rebuttal · Authors · 2026-03-31
>
> We thank the reviewer for the careful reading and address the points below.
>
> ---
>
> ### 1. $X \setminus S_\epsilon$ needs to be open
>
> The reviewer is correct. A sufficient condition for Eq. (41) is that $X \setminus S_\epsilon$ be open, which holds if the reward functions are continuous. Each $S_{i,\epsilon} = \{x \in X : r_i(x) \ge r_i^\star - \epsilon\}$ is a superlevel set of a continuous function, so $S_\epsilon = \bigcup_{i=1}^M S_{i,\epsilon}$ is closed (finite union), and hence its complement $X \setminus S_\epsilon$ is open. We will add this continuity assumption explicitly to Theorem B.2. All rewards used in our experiments (including PickScore, ImageReward, and Aesthetic Score) are continuous functions (neural networks), so the condition holds.
>
> ---
>
> ### 2. Practical impact: new experiments
>
> We added two new experiments addressing practical relevance.
>
> **CIFAR-10 with neural rewards (Fig. R1).**
> Using PickScore, ImageReward, and Aesthetic Score, we evaluate single-reward, pairwise, and 3-way mixtures. Single-reward FID degrades to 60–72, pairwise stabilizes at 52–58, and 3-way holds near 44. Feature variance collapses under single rewards but is sustained under pluralism.
>
> **Text with RLHF-style retraining (Fig. R2).**
> We run 30 rounds of RLHF-style retraining on **Qwen-2.5-1.5B** with BT-curated under competing helpfulness (ArmoRM-Llama3-8B-v0.1) and safety (Llama-Guard-3-1B) rewards. Single-reward curation degrades one objective, while pluralistic curation maintains both. Perplexity increases to 8–9 under single rewards but stays near 6 under pluralism, and diversity (Distinct-2) is preserved (≈0.60 vs 0.47–0.49).
>
> Figures are available in this anonymized Google Drive:
>
> - R1: https://drive.google.com/file/d/1wAxUva-Oae5JQrAQcGq6uMjEIzUJaJmT
>
> - R2: https://drive.google.com/file/d/1uU7iKGBFgBKkTJR2CSHXs3_rNG4no93B
>
> ---
> ### 3. Realism of assumptions
>
>  The assumptions are used to obtain theoretical guarantees, and we complement them with both quantitative analysis in the paper and new empirical validation.
>
> **Bounded rewards.**
> In practice, rewards are often not strictly bounded, but we observe the same qualitative behavior with unbounded neural rewards (e.g., PickScore, ImageReward). This assumption is not restrictive.
>
> **Exact MLE retraining.**
> This is an analytical idealization, as in Ferbach et al. (2024). Finite-K deviations are controlled (Remark 3.1; Appendix B). Importantly, our new Qwen experiment uses actual SFT on BT-curated outputs (not exact MLE), and the same qualitative behavior holds. Thus, the empirical behavior does not depend on exact MLE retraining.
>
> **Disjoint/near-separated basins.** Figure 3 is exactly the "metrics to quantify" this: it shows a phase transition in final reward variance as a function of mode separation, directly measuring the boundary of the regime where the theory applies. For real rewards, the basin geometry is unknown a priori, but our new CIFAR experiment (Fig. R1) provides a direct empirical test. The result is that a reward's variance is preserved under pluralistic curation only when that reward participates in the mixture.
>
> Specifically, PickScore + ImageReward (P+I) collapses Aesthetic variance as badly as single-reward curation (0.05), while I+A and A+P sustain it (0.20). This cross-basin selectivity is the quantitative signature of Lemma 3.4, now validated with black-box rewards whose basin structure is not known to us in advance. This constitutes direct empirical evidence that the near-separated basin structure holds for these rewards.
>
> **Plateau / hard-separation structure.**
> This structure is only needed for the closed-form results (Proposition 3.4, Corollary 3.8). The main non-collapse results (Lemma 3.5, Theorem 3.3) hold under Assumption 2.1 alone, without plateau or hard-separation.
>
> ---
>
> ### 4. Assumptions in appendix
>
> The reviewer is correct. We will surface all proof-only assumptions in the main text and distinguish those needed for non-collapse from those needed for the explicit limit.
>
> ---
> ### 5. Comparison with Zhao et al. (2025)
> We thank the reviewer for pointing to this paper. Zhao et al. models heterogeneity as additive noise around a single reward function where annotators share a common preference but differ by a stochastic perturbation. Our setup is fundamentally different: we study distinct, competing reward functions with potentially disjoint basins, where no single underlying preference exists. Their results characterize stability properties of a unimodal limit, not the mixture dynamics. We will add an explicit comparison in the related work section, clarifying this distinction.
>
> ---
>
>
> These clarifications and experiments directly address the concerns about assumptions, realism, and empirical validation. We hope this supports a more positive assessment.

---

> > ### Author Rebuttal · Reviewer_SXwi · 2026-04-03
> >
> > Thank you for your response, I will keep my positive evaluation.

---

### Official Review · Reviewer_7CVV · 2026-03-12

**Soundness:** 3
**Presentation:** 3
**Significance:** 3
**Originality:** 3
**Overall Recommendation:** 4
**Confidence:** 4

**Summary:**

Recursive retraining of GenAI models can lead to representation collapse, where synthetic outputs curated by a fixed reward signal concentrate on a narrow set of over-optimized responses. This paper shows that such collapses can be mitigated by curating synthetic data using multiple reward functions reflecting heterogeneous preferences. The authors analyze the dynamics of recursive training under this setting and prove that the model converges to a stable, diverse distribution over high-reward outputs, which can be characterized as a weighted Nash bargaining solution that aggregates competing values.

**Compliance With Llm Reviewing Policy:**

Affirmed.

**Key Questions For Authors:**

1. the Cross-reward leakage gaps assumption requires there is no overlap between two basins. But if they do overlap, it is impossible to guarantee $\Delta>2\epsilon$. I believe this is a very common scenario in practice and how it affects the validity of the theoretical results?
2. Seems like the closed-form discrete dynamics in Eq. (5) serve as a starting point for the theoretical discussions. Can we generalize this equation to incorporate more realistic factors? E.g., what if the sampling comes from more than two reward structures, or the reward structure does not even correspond to a underlying BT model. Can you derive something similar to Eq. (5) and extend the results?
3. What if the preference $r$ is not given but implicitly embedded in a pairwise comparison dataset (e.g., for offline preference optimization methods like DPO, the model trainer does not need to recover the reward function $r$)? Can your theory shed any light on the resulting iterative traininng process?

**Limitations:**

Yes

**Strengths And Weaknesses:**

Strengths:
- this paper studies a timely and relevant problem to ICML, the high-level message is encouraging to the ML community
- the presentation is good and the paper is overall technically strong. Assumptions and theorem statements are well presented

Weaknesses:
- The theoretical framework is a bit stylistic and I would like to see how standard practice of nowday's iterative training or alignment procedure for foundation models can be formulated as a special case of the introduced retraining dynamics.
- The experiment can be strengthened by testing your theory on real RLHF pipelines. Although the authors claim that "Our findings align with current practices in RLHF", the experiments do not reflect that.
- See some issues I raised in the question section.

---

> ### Author Rebuttal · Authors · 2026-03-31
>
> We thank the reviewer for the positive assessment. We address each point below.
>
> ---
> ### 1. RLHF as a special case of our framework
>
> Standard RLHF operates as follows: generate K candidate outputs from the current model, have a reward model based on human comparisons select the preferred one, and retrain. The BT model is precisely the model underlying pairwise human preference comparisons and it is how reward models are trained in RLHF. Equation (5) is the dynamics of iterative RLHF, not an approximation.
>
> The single-reward case of our framework is Ferbach et al. (2024). Our extension models what happens when different preference groups (or different objectives) govern selection at different rounds, which is the multi-objective RLHF, where helpfulness, harmlessness, and honesty objectives compete.
>
> ---
> ### 2. Real RLHF experiments
>
> We added two new RLHF-style experiments, one with learned vision reward models and one with text Qwen retraining. Both show the same pattern: single-reward retraining degrades diversity, while pluralistic curation degrades much less. For full setup and results, please see our response to **Reviewer FoWN section 4**.
>
> - R1: https://drive.google.com/file/d/1wAxUva-Oae5JQrAQcGq6uMjEIzUJaJmT
> - R2: https://drive.google.com/file/d/1uU7iKGBFgBKkTJR2CSHXs3_rNG4no93B
>
> ---
> ### 3. Behavior under overlapping reward basins
> We believe the assumption may have been misread. The disjointness condition $S_{1,\varepsilon} \cap S_{2,\varepsilon} = \varnothing$
> is on the $\varepsilon$-optimal basins for a fixed $\varepsilon$, not on the full supports of the two rewards. Once $r_1^* \neq r_2^*$, one can always choose $\varepsilon$ small enough to make the basins disjoint. Our theory applies in this regime.
>
> Also, the theory captures near-overlap continuously. When the basins are close or partially overlapping ($\Delta_1$, $\Delta_2$ small), push the leakage terms $\kappa_i = \exp(-(\Delta_i - 2\varepsilon))$  towards 1, widening the interval on $a_\infty$ in Lemma 3.4. Partial overlap does not break the analysis; it weakens the separation bounds gradually.
> At the extreme, if $\Delta_1 = \Delta_2 = 0$, then the two basins are identical. In that case, the two rewards are functionally equivalent, so there is no nontrivial diversity to preserve and that is the degenerate single-reward case, not a failure of the theory. Figure 3 confirms all of this: final reward variance degrades smoothly as optima approach each other, with a phase transition near distance 2.
>
> ---
> ### 4. Extension beyond BT and two rewards
> **More than two rewards.** Already covered. Section 3 gives the explicit M-reward extension with weights $\{q_i\}_{i=1}^M$, and Table 1 / Figure 4 report results for 1 to 5 preferences. In addition, our new experiment (R1) evaluates a 3-way mixture over three preferences.
>
> **Non-BT selection.** This is an important question. The key structural property used in the proofs is that BT selection induces an exponential tilt of the form $p_{t+1}(x) \propto p_t(x) \cdot \left(\text{weighted sum of } \exp(r_i(x))\right).$  Two points worth noting:
>
> - (i) Plackett-Luce models (ranking-based selection used in some RLHF variants) induces the same exponential tilt up to normalization, and our results extend to that setting without modification.
> - (ii) For any monotone selection mechanism, the mass-decay argument of Lemma 3.3 holds, outside-basin mass contracts because those samples are consistently disfavored. The contraction rate $\rho_\varepsilon$ changes, but the non-collapse result does not depend on the exponential form. If selection is not reward-monotone (adversarial or random curation), our argument breaks and this regime lies outside our framework and remains an open question.
>
> ---
> ### 5. DPO
> In DPO, there is no explicit reward function, and preference is encoded through a pairwise comparison, and the optimal policy implicitly defines a reward of the form  $\beta \log \frac{\pi(x)}{\pi_{\mathrm{ref}}(x)} + \beta \log Z.$
>
> In iterative DPO, the reference policy $\pi_{\mathrm{ref}}$ is itself updated across rounds, which induces a non-stationary reward. Our current theory assumes fixed reward functions $r_1, r_2$, so it does not directly apply to this setting.
>
> Two connections are important:
>
> - (i) DPO is derived under the BT assumption, where the pairwise comparison probability satisfies $P(x_w > x_l) = \sigma(r(x_w) - r(x_l)).$ This means the dataset implicitly encodes a BT reward. If such a reward is recovered, for example via a learned reward model trained, then our analysis applies. This corresponds to the standard two-stage RLHF pipeline (reward modeling followed by policy optimization), including its iterative variants.
> - (ii) In the stationary-reference regime ($\pi_{\mathrm{ref}}$ is fixed, only $\pi$ updated), the induced DPO reward remains constant across rounds, and the connection is direct. The main obstacle is therefore the non-stationarity of iterative DPO, which we leave as future work.

---

> > ### Author Rebuttal · Reviewer_7CVV · 2026-04-04
> >
> > Thank you for your response, I keep my positive evaluation.

---

### Decision · Program_Chairs · 2026-04-30

**Decision:**

Accept (regular)

**Comment:**

This paper studies whether curated synthetic data can mitigate mode collapse. The paper shows theoretically that this is possible using pluralistic curation: a generative model retraining process that selects synthetic data by randomly alternating between multiple reward functions.

Overall, there is substantial agreement among the reviewers. Initial concerns regarding the validity of the theoretical assumptions (such as overlapping reward basins), missing related works, and the real-world practical consequences were comprehensively addressed during the discussion period. The authors provided compelling clarifications and new empirical results (including RLHF-like iterative retraining on larger models), leaving only a consensus on the theory's soundness and relevance. **I recommend the paper for acceptance.**